# Neural manifolds for odor-driven innate and acquired appetitive preferences

Rishabh Chandak [1] & Baranidharan Raman [1] ✉

Sensory stimuli evoke spiking neural responses that innately or after learning drive suitable behavioral outputs. How are these spiking activities intrinsically patterned to encode for innate preferences, and could the neural response organization impose constraints on learning? We examined this issue in the locust olfactory system. Using a diverse odor panel, we found that ensemble activities both during ('ON response') and after stimulus presentations ('OFF response') could be linearly mapped onto overall appetitive preference indices. Although diverse, ON and OFF response patterns generated by innately appetitive odorants (higher palp-opening responses) were still limited to a low-dimensional subspace (a 'neural manifold'). Similarly, innately non-appetitive odorants evoked responses that were separable yet confined to another neural manifold. Notably, only odorants that evoked neural response excursions in the appetitive manifold could be associated with gustatory reward. In sum, these results provide insights into how encoding for innate preferences can also impact associative learning.

In many organisms, the olfactory system serves as the primary sensory modality that guides a plethora of behaviors, such as foraging for food, finding mates, and evading predators. The genetic makeup of these organisms determines the innate preference, or valence, associated with different olfactory stimuli[1–6]. Consequently, neural responses evoked by these stimuli have to be patterned to drive motor neurons to perform appropriate behaviors (i.e., move towards or away) that are key for survival. Given the importance of rapid and robust decision-making[7–10], we wondered how information regarding the valence of a chemical cue is encoded in the olfactory system[2,4,11–17]. Particularly, we examined whether and how neural responses are spatiotemporally structured to represent odor valence in the early locust olfactory system.

In insects, odor stimuli are detected by olfactory sensory neurons in the antenna that transduce chemical cues to electrical signals and relay them to the antennal lobe. A network of cholinergic projection neurons (PNs, excitatory) and GABAergic local neurons (inhibitory) in the antennal lobe fire in unique spatiotemporal combinations to encode for stimulus identity[18–24]. The PN responses are patterned over space and time to encode for different odorants encountered by the insects and relay this information to higher centers responsible for learning, memory, and overall behavioral preferences[25–27]. The odor-evoked PN response patterns are elaborate and continue well after the stimulus is terminated.

Since the behavioral responses initiated by an odorant are often rapid[8,9,28], the relevance of neural activity that occurs well after the stimulus onset remains to be understood. However, what has been reported is that the behavioral responses elicited by an odorant last the duration of the stimulus exposure[28,29]. Further, an emerging perspective from the fly[30] and worm chemotaxis system[31] is that the responses even after termination of a stimulus should be behaviorally relevant. During chemotaxis, the ON responses trigger a surge or upwind run behavior, and the OFF responses bring about the complementary casting or local search behavior. In addition to driving different innate behaviors, in the fly gustatory system[32], both the ON and the OFF responses evoked by bitter stimuli have been shown to alter synaptic plasticity in the mushroom body, albeit in an opposing fashion.

What is the relevance of the odor-evoked ON and OFF responses in driving innate and acquired odor-driven behavioral preferences? We examined this issue in the locust olfactory system. Our results show that both ON and OFF responses of odorants that evoked similar behavioral preferences could be grouped into separate clusters. Therefore, ensemble neural responses during both these time

[1]Department of Biomedical Engineering, Washington University in St. Louis, St. Louis, MO 63130, USA. ✉e-mail: barani@wustl.edu

windows could be used to predict the behavioral outcomes. Intriguingly, both ON and OFF responses could impact learning associations between an odorant and a gustatory reward but in disparate ways. In sum, our results reveal how spatiotemporally structured neural responses could be mapped onto innate and acquired olfactory preferences.

## Results

### Innate appetitive preferences of locusts to an odor panel

We began by assaying the innate appetitive preferences of starved locusts to a large, diverse panel of odorants (1% v/v unless stated otherwise). Each odor in the panel was presented to every locust once using a pseudorandomized order. The palp-opening responses (POR) evoked by all odorants in the panel were recorded (Fig. 1a, b). We used a binary metric to quantify whether each locust responded to an odor by opening its palps (a score of 1 to indicate a palp-opening response (white-colored boxes), and a score of 0 to indicate no response (gray-colored boxes)). For visualization, the odors were sorted based on the number of PORs they elicited across locusts.

We converted these results to a preference index for each odor (see "Methods"). As can be seen from Fig. 1c, we obtained a broad range of preferences for the odor panel. Hexanol (at 10% v/v; leftmost odorant; $x$ axis), a green-leaf volatile, had the highest preference, whereas linalool (rightmost odorant; $x$ axis), a pesticide, had the lowest preference. We categorized odorants as being appetitive, neutral, and unappetitive (one-sided binomial test comparison; neutral and unappetitive odors are jointly referred to as "non-appetitive"). Prior studies have found that preferences for certain odorants can vary between males and females of the same species[1,33,34]. To examine this possibility, we also compared behavioral responses between male and female locusts ($n = 13$ for each gender, Supplementary Fig. 1a). While appetitive preferences for certain odorants did vary between males and females in our dataset (e.g., hexanal and garlic), these differences were not significant ($t$ test, $P > 0.1$ for all odors).

Is there a simple stimulus feature that could account for these diverse appetitive preferences? Since the odorants were diluted to the same concentration (1% v/v) and delivered identically (except hexanol which was alone delivered at 10%, 1%, and 0.1% dilutions), the vapor pressure of the chemicals directly determined how much of each stimulus was delivered. We wondered then if locusts were simply behaving more frequently for more volatile odors (higher vapor pressure). However, as can be seen in Fig. 1d (and Supplementary Fig. 1b), a regression between the vapor pressure of the stimuli against the behavioral responses poorly explained the observed POR trend.

Another potential confound that could impact the observed trends could arise from fatigue/loss of motivation which could potentially diminish the locust PORs in the later trials of the experiment. To eliminate this possibility, we plotted the observed number of PORs as a function of the trial number (Fig. 1e). As can be noted, our results indicate that locust performance remains robust and even slightly increased as the experiment progressed ($R^2 = 0.23$; Supplementary Fig. 1c). In addition, we performed Monte Carlo simulations to verify that population-level responses were not biased by a handful of individuals. Our results confirmed that this is indeed the case and the results converged when any random subset of eighteen or more locusts was used to calculate behavioral preference indices for different odorants (Fig. 1f). Finally, we conducted an independent set of experiments to confirm whether the preferences of locusts to a given odorant remains consistent across repetitions. Our results indicate that the locusts' PORs remained consistent even when the same odorant was encountered in a recurring fashion (Supplementary Fig. 1d). These results, combined with the pseudorandom presentation of odorants, indicate that the behavioral preferences obtained are a strong indicator of the innate appetitive preference of the locusts, and the sample size used was sufficient to get a stable readout.

### Individual projection neuron responses to appetitive and non-appetitive odorants

Next, we sought to understand the neural basis of this behavioral readout. To examine this, we recorded odor-evoked responses from projection neurons (PNs) in the locust antennal lobe (Fig. 2a). We stimulated the antenna with the same odor panel used in the behavioral experiments. The stimulus dynamics of each odorant were quantified using a photoionization detector (PID) and the mean voltage responses for all odors are shown in Fig. 2b (left panel; see "Methods"). The right panel shows the peak PID response for each odorant arranged in order of innate appetitive preferences (cues that evoked the highest behavioral responses are on the left and the lowest are on the right).

We presented each odorant for ten repetitions in a pseudo-randomized order. A total of 89 PNs (pooled across 26 locusts of both sexes; ~10% of the total number of PNs in a single antennal lobe) were recorded using this approach and used for all subsequent analyses. Consistent with prior data, we found that odor-evoked responses had two prominent epochs: an ON response that occurred during the 4 s when the stimulus was presented, and an OFF response that occurred during a 4 s window immediately following stimulus termination. We found a PN that had an ON response for most of the odorants (Fig. 2c, PN A), whereas many PNs responded to a subset of odorants either with an ON response or an OFF response. A small fraction of neurons were OFF-responders to a few appetitive odors but switched to ON responses for some of the non-appetitive odorants (Fig. 2c; PN B; 8/89 PNs with similar tuning). Complementing these responses, we also found a small fraction of PNs that was ON-responsive to all five appetitive odorants but was OFF-responsive to one or more unappetitive odorants (Fig. 2c, PN C; 11/89 PNs with similar tuning). On average, odorants with higher valence elicited stronger ON and OFF responses across more PNs than those with lower valence, while inhibition increased as the odorants became less appetitive (Fig. 2d; see "Methods").

We computed the correlation between the individual PN responses to different odorants with the overall behavioral preferences of the same panel (Fig. 2e). Notably, we found a small subset of neurons that had either a strong positive or negative correlation with the POR responses observed. Furthermore, our results indicate that such correlations could be found when either the ON or OFF responses were used. Although, it would be worth noting that different subsets of PNs had a high correlation with appetitive preference during the ON and the OFF periods.

How selective are individual PN responses? To answer this, we computed a tuning curve for each PN during both the odor ON and OFF periods (Fig. 2f). We found that most PNs responded to at least two odorants or more during the ON period (84/89 PNs) and a small fraction of neurons (11/89 PNs) responded to ten or more odorants (Fig. 2f, bar plots along the $y$ axis). The odor-evoked responses were more selective during the OFF period, with 70/89 PNs responding to two or more odors and only three PNs responding to more than ten odorants. In sum, these results indicate that individual PNs responded to the odor panel with great diversity.

### Ensemble projection neuron responses to appetitive and non-appetitive odorants

Next, we examined how odor-evoked responses vary at an ensemble level. To visualize the ensemble neural responses and how they change as a function of time, we used a linear dimensionality reduction technique (Principal Component Analysis, PCA; see "Methods"). PCA neural response trajectories for the ON period are shown for all odorants (Fig. 3a; 4 s of odor presentation). Consistent with prior findings[22,35–37], our data also reveal that each odorant produced a distinct looped response trajectory. Interestingly, we observed that neural response trajectories evoked by odorants that were labeled as innately appetitive in the behavioral assay evolved in a similar direction (blue trajectories). This indicates that the

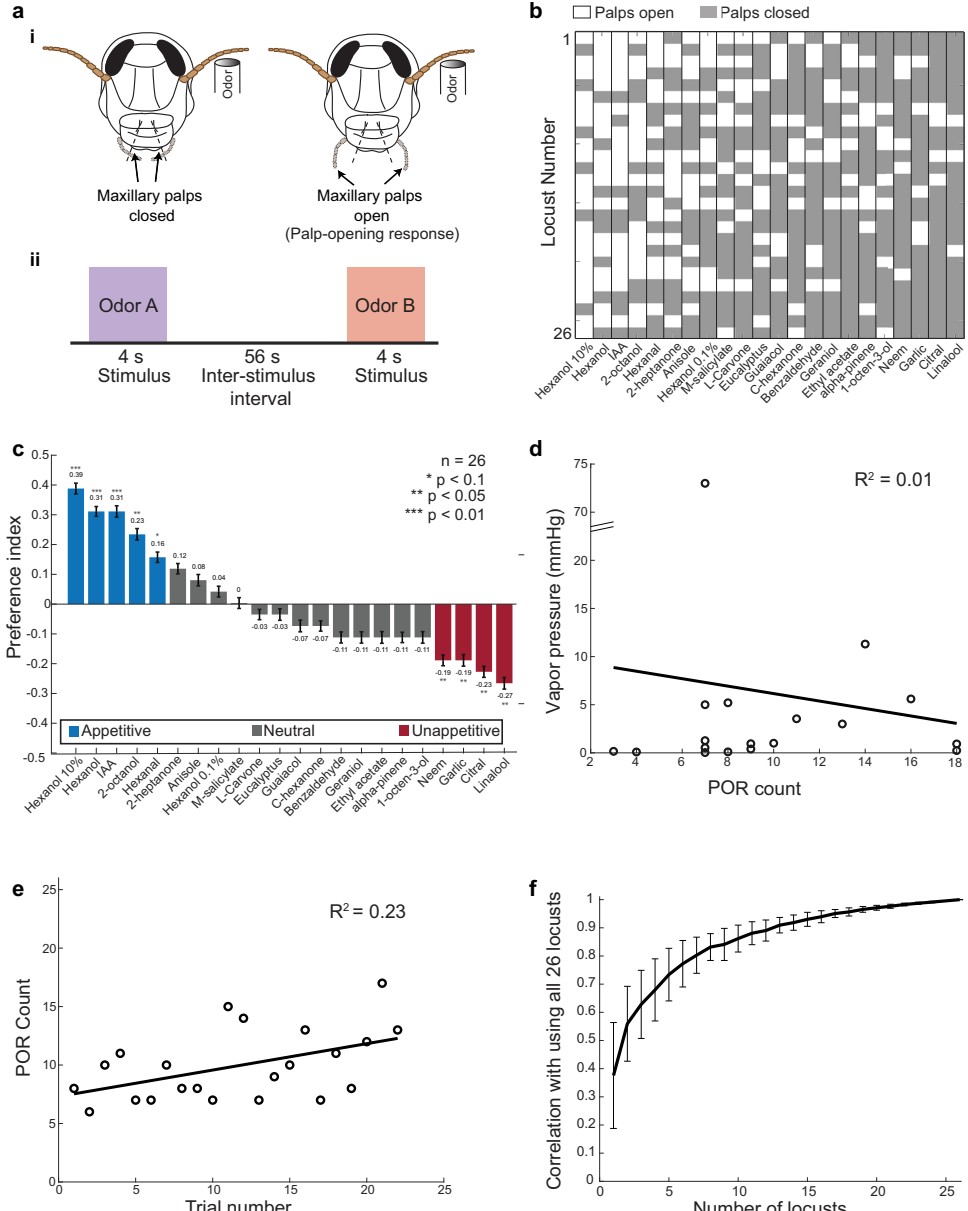

**Fig. 1 | Innate appetitive preferences of locusts to a diverse odor panel. a** (i) A schematic showing a palp-opening response (POR). A successful POR was defined as an opening of the maxillary palps beyond the facial ridges shown on the locust. (ii) Odors were delivered in a pseudorandomized order onto the locust antenna. The stimulus delivery was 4 s in duration, and the inter-stimulus-interval was set to 56 s. All source data are provided as a Source Data file. **b** Innate preferences of 26 locusts for the 22 odorants tested are shown. Each row shows the POR responses of a locust to the odor panel. White boxes indicate successful PORs to odorants and gray boxes indicate no PORs. Note that odorants are sorted based on the number of PORs elicited across locusts (highest–leftmost to lowest–rightmost). **c** Preference indices were calculated for all odors tested and are shown as a bar plot (*n* = 26 locusts). Blue bars indicate odors classified as appetitive, gray bars indicate neutral odors and red bars indicate unappetitive odors. Locusts with a significant deviation from the median response (one-sided binomial test, *P* < 0.1, were classified as either being appetitive or unappetitive; *\*P* < 0.1, \*\**P* < 0.05, \*\*\**P* < 0.01). Error bars indicate s.e.m. **d** Regression analysis of odor vapor pressure versus the number of PORs generated (across all 26 locusts) is shown. Only odorants with available vapor pressure data were considered (18/22 odors at 1% v/v concentration). The best-fit linear regression line is shown. **e** Regression analysis of POR counts versus the trial number is shown. Each circle indicates the number of locusts with successful PORs in that particular trial. The best-fit linear regression line is shown. **f** Results from Monte Carlo simulations are shown (see "Methods"). Preference indices obtained by using a random subset of locusts of a particular size (i.e., any *n*-locusts-out-of-26) were correlated with overall results obtained using all 26 locusts. The mean ± s.e.m. correlation values across 100 random simulations are shown for each sample size. An *R*² value above 0.95 was obtained for simulations with *n* > 18 locusts.

combination of PNs excited by these odors had overlap and hence the PN ensemble vectors were near one another in the state space. Similarly, the trajectories for odors labeled as unappetitive also evolved in a similar direction (red trajectories) and occupied a different region of the state space. Note that the sets of red and blue trajectories did not overlap, indicating that odors within different groups (appetitive and unappetitive) were being encoded by relatively distinct subsets of PNs.

We confirmed these dimensionality reduction results with a high-dimensional clustering analysis (Fig. 3b). We found that the spiking profiles for odors that belonged to the same group (appetitive or unappetitive) were similar, and hence clustered within the same branch when visualized using a dendrogram. These results support our interpretation that different subsets of PNs in the antennal lobe are activated in a manner that is representative of the innate appetitiveness of the stimulus.

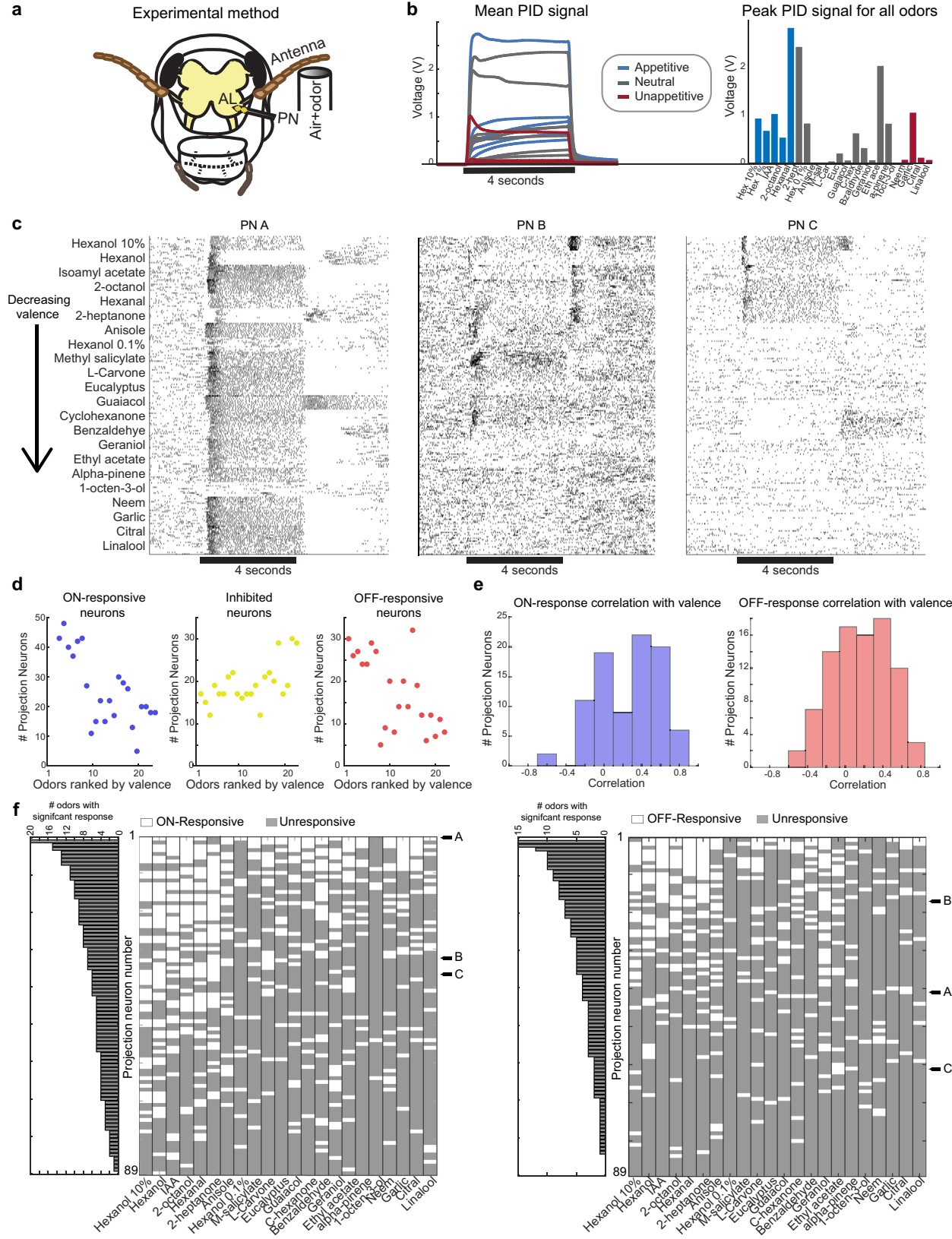

## Predicting behavioral preferences from odor-evoked neural responses

How well do the neural responses map onto the behavioral preferences for different odorants? To examine this, we used linear regression to predict the probability of generating a POR given the ensemble PN activity elicited by that odorant. (Fig. 4a). Note that for

these predictions, we used the normalized behavioral responses for each odor (see "Methods"), which could also be interpreted as the probability of a palp-opening response to a given odorant (across locusts). The regression weights were trained using all but one odorant and used to predict the probability of POR for the left-out odorant (i.e., a leave-one-odorant-out-cross-validation approach; 22 different linear

**Fig. 2 | Individual PN responses to appetitive and non-appetitive odorants. a** A schematic of the experimental setup is shown (see "Methods"). All source data are provided as a Source Data file. **b** Mean voltage signals (left panel) and peak values (right panel) acquired from a photoionization detector are shown for all odorants in the panel. Same coloring/ordering convention as in Fig. 1c. **c** Representative PN responses to all 22 odorants are shown. Each tick indicates an action potential, each row corresponds to one trial, and ten trial blocks are shown for each odorant. The black bar indicates the four seconds odor presentation window. **d** Left panel: The number of PNs that were activated during the odor presentation window (ON responsive) is plotted for all odorants in the panel. Middle and right panels: Similar plots but showing the number of PNs that were inhibited during odor presentation, and the number of PNs activated after odor termination (OFF-responsive) are shown. In all three scatter plots, the odorants are arranged along the x axis based on

their appetitive valence. **e** Left panel: The distribution of correlation values between each PN's response to the odor panel (a 22-dimensional vector; mean ON responses) with the overall appetitive preference for the same set of odorants is shown. Right panel: Similar plot as the left panel, but the OFF-period PN activity (4 s immediately following odor termination) was now correlated with the overall odor valences. **f** Left: Responses of individual PNs to all 22 odors during the ON period are shown. Each row corresponds to a single PN, and the odorants (columns) were organized from the highest valence to the lowest (left to right). PNs were classified as ON responsive (white box) or unresponsive (gray box). The bar plot on the left indicates the number of odorants that activated each PN. PNs are sorted such that those that responded to most odorants are at the top (i.e., least selective). Note that individual PNs whose rasters are shown in (**c**) are identified. Right: Similar plot as the left panel but characterizing OFF responses across all 89 PNs to the odor panel.

regression models were used). We found that this simple approach yielded robust predictions for all odorants (Fig. 4b, c).

Note that we made predictions using the mean ensemble PN activity during 4 s of odor exposure (i.e., an "ON-regressor"), and using 4 s of odor-evoked activity after the termination of the odorant (i.e., an "OFF-regressor"). Both the regressors performed relatively well with the ON-regressor performance being better than the OFF-regressor. Further, the performance of the linear regression approach with shuffled prediction probabilities for different odorants (i.e., "shuffled control" for both ON and OFF cases) predicted values around the mean POR probability for all odorants (Fig. 4b, c; mean = -0.4), and was significantly inferior compared to the ON- and OFF- regression approaches. The poor performance of the shuffled control approach compared to the ON- and OFF- regressors suggest that the spiking activity across PNs is indeed organized to enable mapping between neural and behavioral response spaces.

How consistent were the different regression models? Our results indicate that the weights assigned to each PN remained stable irrespective of the odor that was left out to train the regression model (Fig. 4d). This consistency of the assigned weights across regressors indicates that no particular odorant disproportionately influenced the regression model used to transform neural responses into POR probabilities. In addition, Monte Carlo simulations (see "Methods") revealed that both the ON- and OFF- regressors' performance improved as the number of PNs used in the analyses was increased (Supplementary Fig. 2a).

We wondered whether the same set of PNs contributed during both ON and OFF periods to predict the preference index for different odorants. To understand this, we calculated the correlation coefficient between the weights assigned by both these regression approaches (Fig. 4e). Our results indicate that there was only a weak correlation between weights assigned by the ON- and OFF- regressors. To further corroborate this conclusion, we examined the ability of the ON-regressor to predict the behavioral PORs when using OFF-period data (i.e., using temporal epoch that was not used to obtain ON-regressor weights; Supplementary Fig. 2b). Our results indicate that this approach resulted in very poor prediction results ($R^2$ of 0.291). Even poorer results were obtained when OFF-regressor was used to predict behavioral PORs using odor-evoked responses during stimulus presentation (Supplementary Fig. 2c). Taken together, these results indicate that information regarding the overall appetitive preference is distributed across different sets of PNs during the ON vs OFF epochs. In sum, we conclude that the ensemble neural responses during odor presentations and after their termination are odor-specific and contain information about the overall innate behavioral response generated by that odorant.

**Innate versus acquired appetitive preferences for odorants**

Next, we wondered if innate appetitive preferences for odorants and the neural responses they evoke can inform regarding other behavioral dimensions such as learning and memory. To examine this, we used an

appetitive-conditioning assay (Fig. 5a). Locusts were starved for 24 h and pre-screened for innate responses to the odorants used in the assay. Only those that did not have innate responses were used for the appetitive-conditioning experiments (see "Methods").

We trained locusts with four chemically and behaviorally diverse odorants as conditioned stimuli in an "ON-training paradigm" (Fig. 5a and Supplementary Fig. 3). Following training, we examined the ability of the trained locusts to respond to the conditioned stimulus in an unrewarded test phase. Opening of maxillary palps (palp-opening response) was regarded as a readout of successful stimulus recognition. We found that locusts trained with hexanol or isoamyl acetate as a conditioned stimulus robustly responded to the presentation of these odorants in the test trials. However, we found that locusts trained with citral and benzaldehyde showed no palp-opening response during the testing phase (Fig. 5b, c). This observation eliminates the possibility that the POR responses we observed could arise from non-specific behavioral facilitation and that no odor-reward associations were learned.

Next, we examined whether locusts could be conditioned when the reward was delayed until half a second after the termination of the conditioned stimulus (i.e., "OFF-conditioning paradigm (0.5 s delay)"; note that similar conditioning protocol but with longer delays between odor termination and presentation of electric shock have been examined in flies[38]). For this set of experiments, we only used hexanol and benzaldehyde as the conditioned stimuli (Fig. 5d). Once again, our results indicated that only locusts trained with hexanol robustly responded with PORs to the trained odorant in the testing phase. However, the POR dynamics observed in OFF-paradigm trained locusts were noticeably different from those we noted in the ON-training paradigm case. In the ON-training case, we found that locust PORs began immediately after the onset of the CS, lasted the duration of the stimulus, and the palps began to close following the termination of the stimulus. The peak of the PORs always occurred during the CS presentations. In contrast, for the OFF-training case, locust PORs were significantly slower (Supplementary Fig. 4), and the peak of the PORs in many locusts occurred after the termination of the stimulus.

In sum, these results indicate that only some odorants can successfully be associated with the food reward. Furthermore, both presentations during and after the termination of the stimulus can lead to the odor-reward association but the behavioral response dynamics are significantly different between the two cases.

**A linear model predicts behavioral response dynamics and cross-learning**

How important is the timing of the reward in learning odor-reward associations? We found that even for those odorants that resulted in a successful association between the odor and the reward, the timing of the reward during training was important. When the reward was presented 4 s after the termination of the odorant we found that no stimulus-reward associations were learned (Fig. 6a; "OFF-Conditioning (4 s delay)"). These results confirm that learning did happen in the

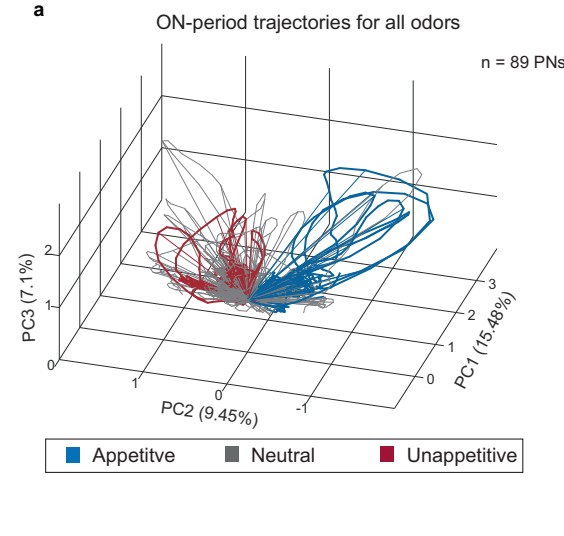

**a** ON-period trajectories for all odors

n = 89 PNs

PC3 (7.1%)
PC1 (15.48%)
PC2 (9.45%)

■ Appetitve   ■ Neutral   ■ Unappetitive

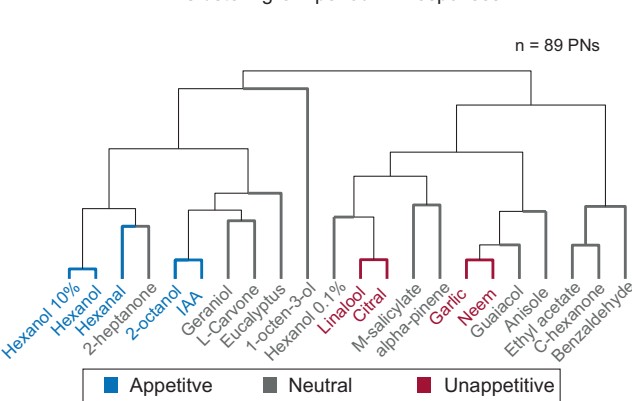

**b** Clustering ON-period PN responses

n = 89 PNs

Hexanol 10%, Hexanol, Hexanal, 2-heptanone, 2-octanol, IAA, Geraniol, L-Carvone, Eucalyptus, 1-octen-3-ol, Hexanol 0.1%, Linalool, Citral, M-salicylate, alpha-pinene, Garlic, Neem, Guaiacol, Anisole, Ethyl acetate, C-hexanone, Benzaldehyde

■ Appetitve   ■ Neutral   ■ Unappetitive

**Fig. 3 | Ensemble PN responses for appetitive and non-appetitive odorants.**
**a** Visualization of the ensemble (*n* = 89) PN responses to the odor panel after Principal Component Analysis (PCA) dimensionality reduction are shown (see "Methods"). 4 s of ON responses for all 22 odorants were used for this analysis, and the data were projected onto the first three principal components that captured the highest variance (~30% captured along the three axes shown). Neural response trajectories evoked by innately appetitive odors are colored in blue, neutral odors response trajectories are indicated in gray, and unappetitive odors responses are shown as red trajectories. Note that the ensemble neural response trajectories cluster based on overall appetitive valence. **b** Dendrogram showing the overall hierarchical organization of 89-dimensional PN ON responses. Odorants are again colored based on the corresponding behavioral preferences (blue indicates appetitive odors, gray indicates neutral odors, and red indicates unappetitive odors). Appetitive odors cluster along the left branch, while unappetitive odors cluster on the right branch. It is worth noting that these results are similar to the overall arrangement of responses shown after dimensionality reduction in (**a**). All source data are provided as a Source Data file.

showed PORs to hexanol and isoamyl acetate (Fig. 6b). For the OFF-training paradigm, we found that learning/cross-learning was observed only in those locusts that received rewards within 2 s of the termination of the conditioned stimulus. Interestingly, a large fraction of locusts (~60%) that received reward immediately after the termination of benzaldehyde (0.5 s after cessation) again paradoxically responded to hexanol and isoamyl acetate (Fig. 6c, d).

How predictable are these behavioral response dynamics and memory cross-talks given the neural responses evoked by these four odorants? To understand this, we set up determining the neural-behavioral transformation as a regression problem with sparsity constraints. For each training paradigm, the goal was to predict the POR responses to all four odorants examined given the time-varying ensemble neural responses evoked by each odorant. Six such regression problems were set up, one for each training paradigm used in our study. We found that POR responses to all four odorants could be predicted reliably for all cases (red curves, Fig. 7a). We found that a linear mapping could indeed be found where the POR dynamics predicted from the neural responses were in good agreement with those observed in behavioral experiments (Fig. 7a; black (actual) vs. red (predicted); Fig. 7b). Notably, the regression weights assigned to different PNs to predict the POR for each training paradigm were highly similar (Fig. 8a, b). This result indicated that the mapping between neural responses and the PORs is highly consistent since the main trend observed in all cases were PORs to positive valence odorants (hex and iaa) and a lack of response to those with negative valence (citral and bzald). Consistent with this interpretation, we found that those PNs that received the most positive weights in the linear regression responded strongly to both positive valence odorants and had little to no responses to exposures of benzaldehyde and citral (Fig. 8c). On the other hand, PNs that responded strongly to the negative valence odorants and had transient responses at the onset and offset of both positive valence odorants received the most negative weights. More importantly, the negatively weighted PNs showed stronger spiking activities to the non-appetitive odorants, which allowed the suppression of POR responses (Fig. 8c; gray traces taller than black traces for benzaldehyde and citral).

We also compared the firing activity of individual neurons with the weights they were assigned in the regression analysis (Fig. 8d, e). Consistent with the PSTH shown (Fig. 8c, black traces), we found that positively weighted PNs had stronger responses to the appetitive odors (hexanol and IAA) relative to non-appetitive odors (benzaldehyde and citral). Furthermore, the negatively weighted PNs had stronger responses for non-appetitive odorants (consistent with Fig. 8c, gray traces).

In sum, these results indicate that the behavioral responses' strength and dynamics evoked by different odorants could be predicted from time-varying ensemble neural responses observed in the antennal lobe and that a robust linear mapping involving ~50% of the total neurons was sufficient to transform neural activity into POR output.

### A spatiotemporal coding logic for encoding appetitive odor preferences

Are the neural responses to appetitive and non-appetitive odorants organized in an interpretable fashion to explain the diverse set of neural and behavioral observations? To understand this, we visualized the ensemble neural activity trajectories of different odorants during both the ON and OFF periods. As can be observed, the odor-evoked ensemble responses were organized into four well-defined subspaces/clusters: appetitive ON, appetitive OFF, non-appetitive ON, and non-appetitive OFF (Fig. 9a, b, the non-appetitive cluster includes odorants with both neutral and negative valences). Note that the different directions in this coding space indicate different combinations of PN

other paradigms (Fig. 5) and that both the identity of the odorant and the timing of the reward constrain what can be acquired through this Pavlovian conditioning approach.

Next, we wondered how locusts conditioned with a particular odorant (i.e., "the training odor") respond when tested using other untrained odorants i.e., how olfactory learning generalizes between odorants. Our results indicate that locusts trained with hexanol responded robustly to presentations of isoamyl acetate (another odorant with a positive valence; Fig. 6b). Exposures to citral and benzaldehyde evoked no responses in hexanol-trained locusts. Surprisingly, locusts trained with citral and benzaldehyde showed no responses to the trained odorant, but a significant fraction of them

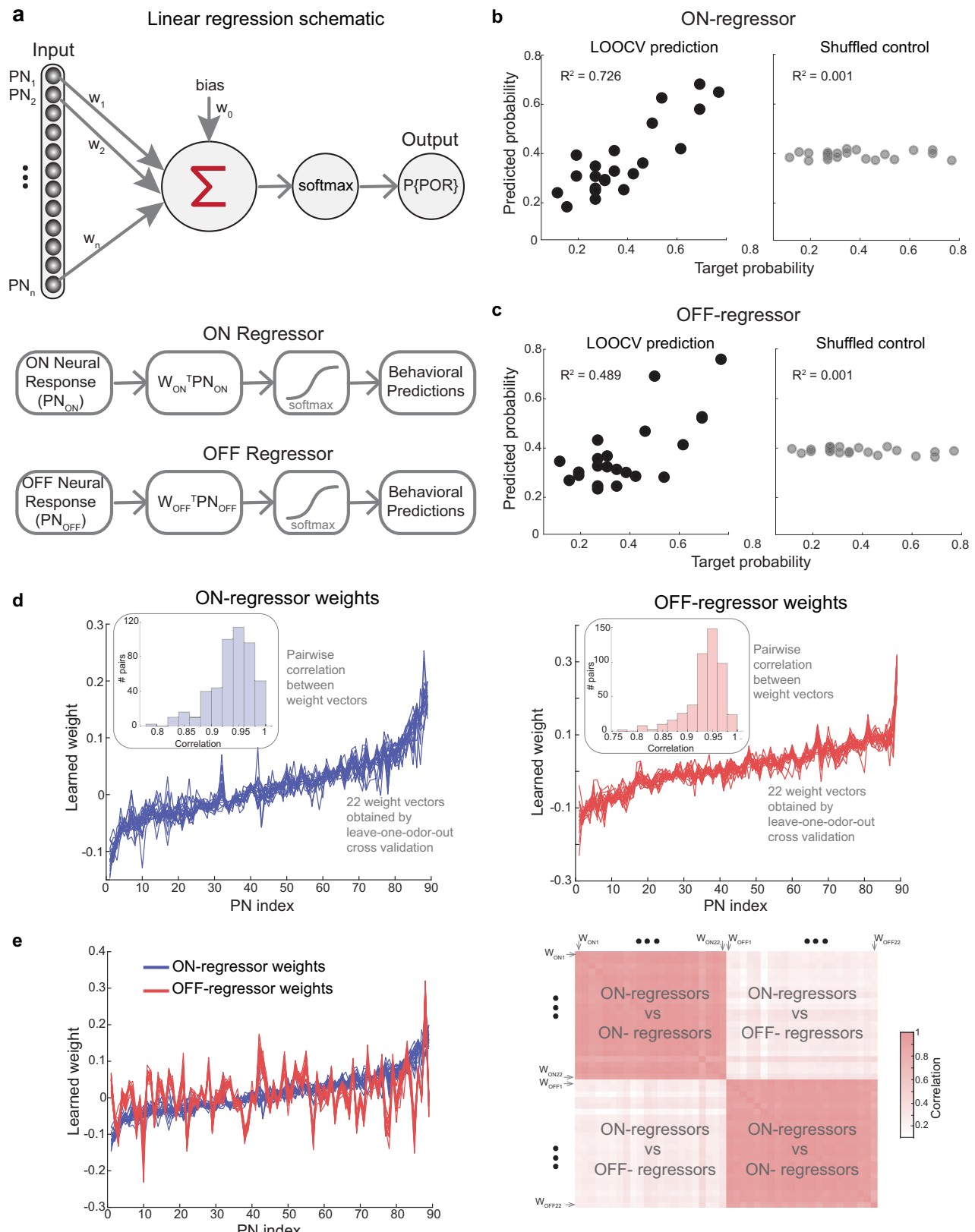

responses, and nearby regions indicate pattern-matched neural responses. Therefore, these results indicate that while the neural activities during appetitive odorant exposures varied from one odorant to another (Fig. 9a, b−cluster 1), they were still constrained to exploit only a limited combination of PN responses and therefore restricted to a particular subspace/region in this coding space.

Extending this logic, these results also indicate that ensemble activities after the termination of appetitive odorants (Fig. 9a, b−cluster 2), during exposures to non-appetitive odorants (Fig. 9a, b−cluster 3), and after cessation of the non-appetitive stimuli (Fig. 9a, b−cluster 4) all employed restricted combinations of ensemble neural responses that were different from each other.

**Fig. 4 | Neural response patterns robustly predict innate behavioral preferences for odorants. a** Schematic of the linear regression approach is shown (see "Methods") All source data are provided as a Source Data file. **b** Left: Predictions from the ON-regressor versus the actual probabilities obtained from the behavioral assay for all odorants in the panel are shown. Overall, the $R^2$ value between the predicted value and the actual behavioral response was high ($R^2 = 0.726$). Right: Similar plot but for the shuffled control is shown. Here, the behavioral POR probabilities were randomized, and a regression model was fit similar to learning the unshuffled case. Note that the predictions are centered around the mean valence of -0.4. **c** Similar plots as panel b, but using models trained on the OFF-period responses are shown. The OFF-regressors performed poorer than the ON-regression models but were still well above shuffled control performance levels. **d** Left: The ON-period linear regression model was validated by training 22 different models, leaving 1-of-the-22-odors out each time for validation. The weights obtained for each PN are shown for all 22 models trained using this leave-one-odor-

out-cross-validation approach. The weights assigned to 89 PNs were sorted (i.e., lowest to highest) based on the model used to predict POR responses to hexanol. The inset shows the distribution of pairwise correlations between each weight vector obtained for predicting POR for different odorants. Right: Similar plot as left panel, but for the 22 OFF-regressors are shown. **e** Left: Blue curves indicate weight vectors obtained from the ON-period regressors as shown in panel d. Red traces show weights learned by the OFF-period regressors but sorted using the same indices as the ON-period vectors. As can be seen, the blue and red curves are uncorrelated. Right: Correlation analysis quantifying the similarities in weights assigned to PNs by the ON- and the OFF- regressors. Weights learned by the PNs are highly correlated within the ON-period and OFF-periods (darker colors along the diagonal blocks). However, the weights assigned to each PN are different between the ON- and OFF-regressors, and hence the off-diagonal blocks have lower correlations (lighter colors).

Notably, the variance in neural responses evoked by appetitive odorants primarily spanned a low-dimensional space (i.e., a "neural manifold") that contained clusters 1 and 2. Only odorants that evoked neural responses limited to this manifold could be associated with food rewards (therefore referred to as the "reinforceable odor manifold"; Fig. 9a). Presenting the reward during activation of either neural cluster 1 or cluster 2 led to learning. However, the behavioral response dynamics significantly varied depending on whether the reward overlapped with cluster 1 or cluster 2 (Supplementary Fig. 4). In contrast, the variance in neural responses evoked by non-appetitive odorants spanned a different manifold that contained clusters 3 and 4. Presenting reward during the activation of either of these response clusters did not result in successful conditioned stimulus-reward associations (therefore referred to as the "non-reinforceable odor manifold"). Although, we note that non-specific facilitation of PORs to other odorants (hex and iaa) was observed.

To quantify these low-dimensional patterns observed in the PCA space, we computed the similarity between odor-response vectors obtained using all 89 PNs. For each odor, we obtained an 89-dimensional vector to capture the mean response during the ON period and calculated the angle between all such vectors for all odors (Fig. 9c and Supplementary Fig. 5). Note that a smaller angle (in degrees) represents the greater similarity between two vectors/odors. For each odor, we computed 21 angles (22 odors, ignoring self-comparison; refer Supplementary Fig. 5b, c) and grouped them based on comparison with either appetitive or non-appetitive odors. We then subtracted the average angle of the innately appetitive group from the non-appetitive group to obtain a single similarity angle for each odor. A net positive angle indicates that the odor's responses were more similar to the appetitive group while negative angles denote better pattern-match with non-appetitive odors. Figure 9c shows this net angular similarity value for each odorant in our panel. The odors are sorted by valence, and the bars are colored to denote the probability of innate PORs for the odorant. Overall, these results are quite similar to those obtained from the manifold analyses (clusters 1 and 2), indicating that high-dimensional neural responses agree with the low-dimensional approximations. A similar result was also obtained when using the OFF-period responses to perform this analysis (Fig. 9d; similar to clusters 3 and 4; Supplementary Fig. 5b).

Remarkably, our results indicate that similar odor-evoked response manifolds were also observed when neural responses were monitored in behaving preparations (Supplementary Fig. 6). In sum, these results reveal an organizational logic for patterning spatio-temporal ensemble neural responses to mediate both innate and acquired odor-driven appetitive preferences.

## A Hebbian neural network for sensory-to-behavior mapping

Finally, to gain mechanistic insights regarding how conditioning odorants with reward increased PORs for only some odorants (hex and

iaa) but not others (bza and cit), we developed a computational model (Fig. 10a). In this model, the input neuron responses (obtained directly from the antennal lobe projection neuron responses we recorded experimentally) feed-forward onto two downstream neurons that had opposing functions: a "Decoding Neuron 1 (DN1)" that drives appetitive response and a "Decoding Neuron 2 (DN2)" that inhibits that same response (i.e., an "anti-neuron"). Both downstream neurons received input from the entire input ensemble. However, weights from one set of input neurons (encoding neural ensemble 1) onto the appetitive Decoding Neuron 1 alone were Hebbian plastic in this model. The rest of the network connections remained unaltered after initialization (see "Methods" for details).

Such "neuron–anti-neuron" pairs have been utilized for predicting overall motor outputs[16,29,39], and are highly consistent with the emerging view from other insect models that have shown mushroom body output neurons form segregated channels to drive opposing behaviors[40,41]. Finally, the motor output neuron that drives behavior in the model merely takes the difference in the overall activity of the "neuron-anti-neuron pair" (i.e., DN1− DN2) to determine the final behavioral output: successful POR only if appetitive Decoding Neuron 1's activity was stronger than the suppressive Decoding Neuron 2's response. In order to replicate the results from our conditioning experiments, the model would require two criteria to be met: a) reinforceable odorants (hex and iaa) should strongly activate encoding neural ensemble 1 that makes plastic connections with Decoding Neuron 1, b) Non-reinforceable odorants (bza and cit) should evoke strong neural activity in ensemble 2, and at least some weak activity in the encoding neural ensemble 1 (Fig. 10b).

Consistent with our hypothesis, we found that the neural responses we recorded for the four odorants used in the conditioning experiments did meet the above expectations regarding how the appetitive and non-appetitive odorants activated the neural ensemble (Fig. 10c). As can be noted, some neurons (at the top of PN activity vectors) were activated more by hexanol and isoamyl acetate while responding less to benzaldehyde and citral. On the other hand, only a smaller subset of neurons that were strongly activated by benzaldehyde (near the bottom of PN activity vectors) also responded to hexanol and isoamyl acetate. Therefore, the antennal lobe activity that drives the responses in downstream neurons is consistent with the schematized inputs shown in Fig. 10b.

Next, to simulate behavioral conditioning, we updated the weights between the encoding input ensembles and Decoding Neuron 1 alone using a simple Hebbian update rule (see "Methods"). Note that the response threshold for Decoding Neuron 1 was set high to prevent false positive PORs before learning. On the other hand, the response threshold for Decoding Neuron 2 was set low as the overall response strength for non-appetitive odorants was weaker and allowed robust suppression of PORs to all odorants. Our results indicate that irrespective of whether the hexanol or benzaldehyde

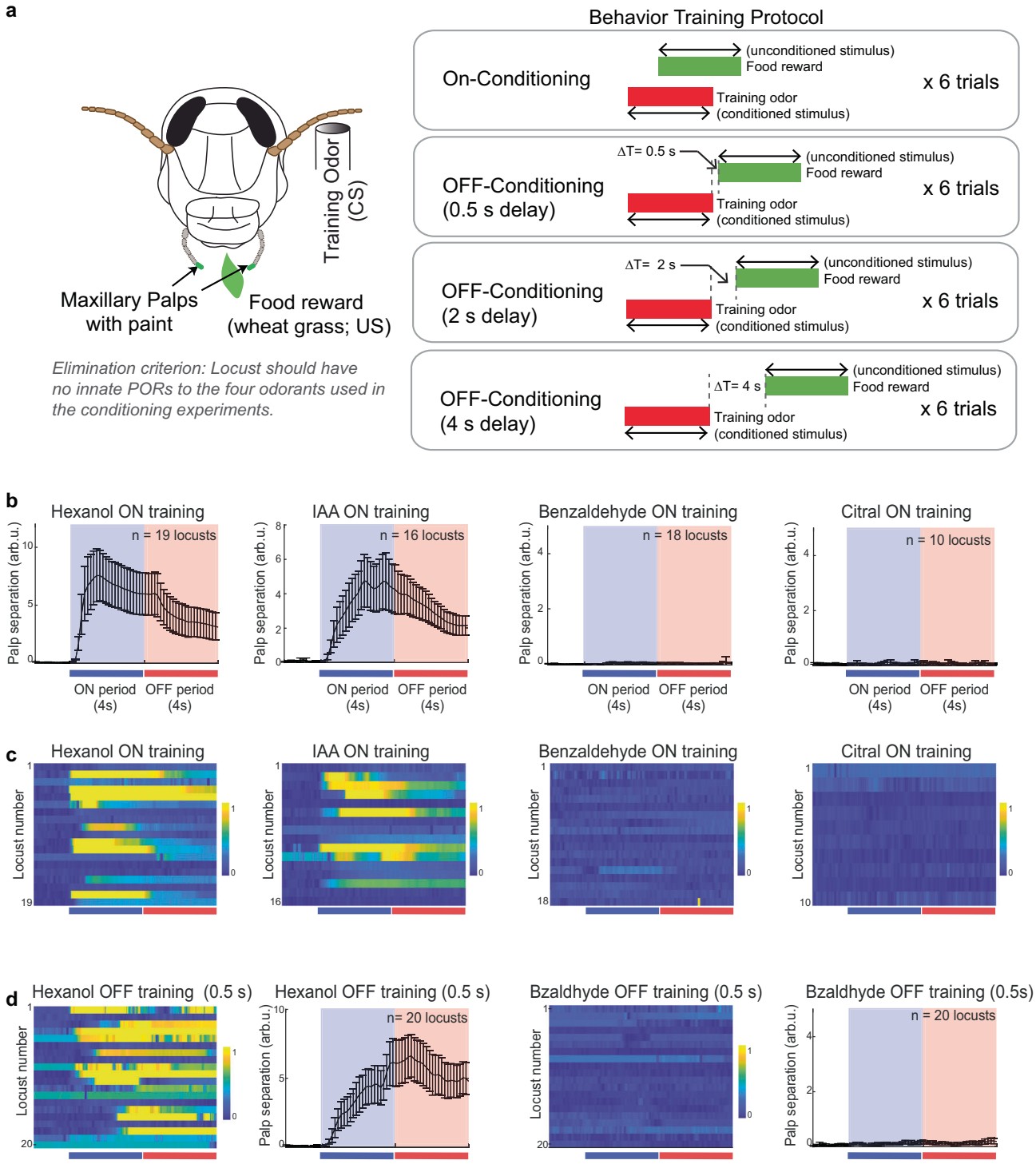

**Fig. 5 | Only innately appetitive odorants can be reinforced using classical conditioning. a** A schematic showing the training protocol followed for both ON- and OFF- classical appetitive-conditioning assays (see "Methods" for details). Following the training phase, locusts were then tested for palp-opening responses (PORs) in an unrewarded phase. All source data are provided as a Source Data file. **b** Results from ON-conditioning using four different odors are shown. The mean POR response of locusts during the unrewarded testing phase is shown in each plot. The testing odor was the same as the training odor, as indicated on each plot. Colored bars indicate 4 s of odor presentation and 4 s immediately following odor termination. Error bars indicate s.e.m., and the number of locusts that had significant PORs for each conditioning odorant are shown in parentheses. As can be seen, locusts trained with hexanol and isoamyl acetate were able to produce POR responses in the test phase, while benzaldehyde and citral training yielded no responses. Note that different sets of locusts were trained/tested for each odorant. **c** POR traces for the four sets of locusts trained with hexanol, isoamyl acetate, benzaldehyde, or citral are shown. The PORs shown were recorded during the testing phase. Each row corresponds to the response observed in one locust. The responses were normalized to range between [0, 1] for each locust (see "Methods"; blue = 0 and yellow = 1). **d** Similar traces as shown in panels b and c but for OFF-conditioning using hexanol or benzaldehyde are shown. Hexanol-OFF training produced significant PORs in 12/20 locusts, whereas benzaldehyde-OFF training yielded no significant responses. Note that the PORs for hexanol-OFF training were delayed and persisted well into the OFF period (compared to hexanol-ON trained responses shown above).

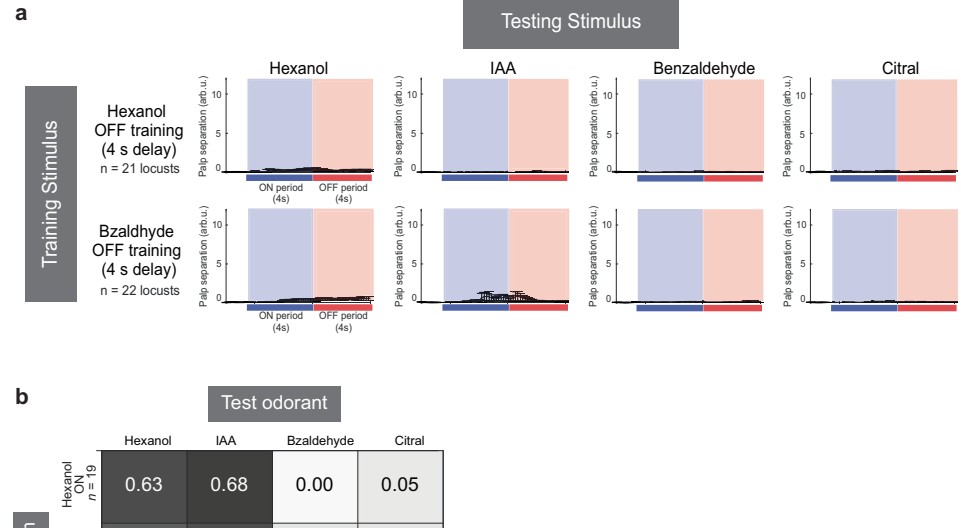

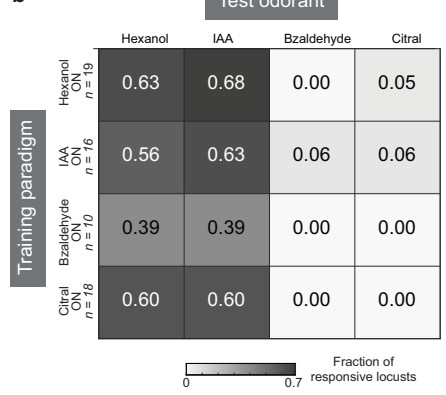

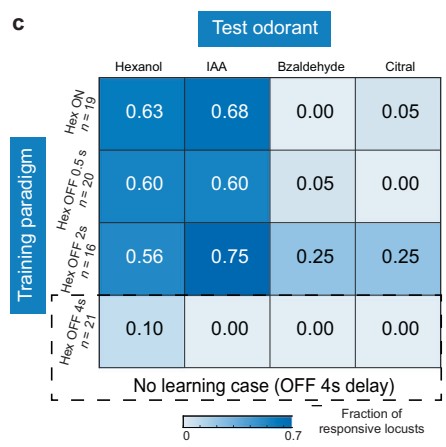

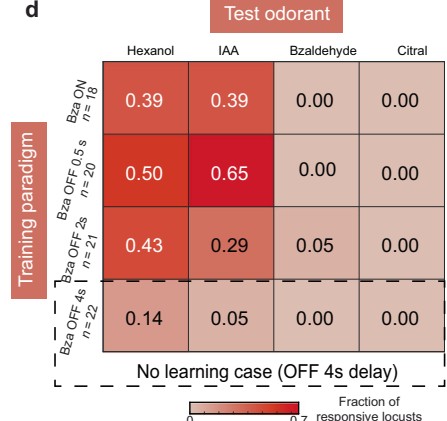

**Fig. 6 | Quantifying learned POR responses across different training paradigms.** **a** POR responses observed in locusts that were conditioned by presenting the reward 4 s after the termination of the odorant (i.e., OFF-conditioning locusts with 4 s delay). Mean responses of conditioned locusts are plotted, and error bars indicate s.e.m. As can be noted, none of the locusts learned to respond with a POR to any of the odorants presented during the test phase. This result indicates that the timing of the reward is an important variable and that only some stimulus-reward

associations could be learned. All source data are provided as a Source Data file. **b** Heatmap showing the fraction of locusts that produced significant PORs to the four test odorants (x axis; rows) for ON training with four different odorants (y axis; columns). **c** Similar plot as (**b**) but for locusts trained with hexanol using ON- and OFF training (0.5 s, 2 s, 4 s gaps) paradigms. **d** Similar plot as (**b**) but for locusts trained with benzaldehyde using ON- and OFF-training (0.5 s, 2 s, 4 s gaps) paradigms.

ensemble responses were used for updating network weights, it always resulted in an increased input to the Decoding Neuron 1 (Fig. 10d vs. e). Therefore, Decoding Neuron 1 had a transient output after onset for hexanol and isoamyl acetate irrespective of the odor used for reward pairing (Fig. 10d, e, arrowheads). The stronger response of Decoding Neuron 2, that was only primarily activated by benzaldehyde and citral ensured that there was no POR output to these odorants even after Hebbian modification of network weights. Thus, this simple neural network with a "neuron–anti-neuron pair" and selective Hebbian connections was sufficient to replicate results from our conditioning experiments.

## Discussion

In this study, we examined the neural correlates of innate and acquired olfactory preferences. Our results indicate that while the neural responses evoked by an odorant are patterned over combinations of neurons activated and over time, the ensemble neural responses are still constrained by the overall behavioral relevance of the chemical cue. Odorants that have a positive appetitive preference, or valence, evoked ensemble neural responses that overlapped during odor presentations (i.e., ON responses) and after their terminations (i.e., OFF responses). Similarly, odorants with a neutral or negative appetitive preference evoked spiking activities that formed similar ON and OFF

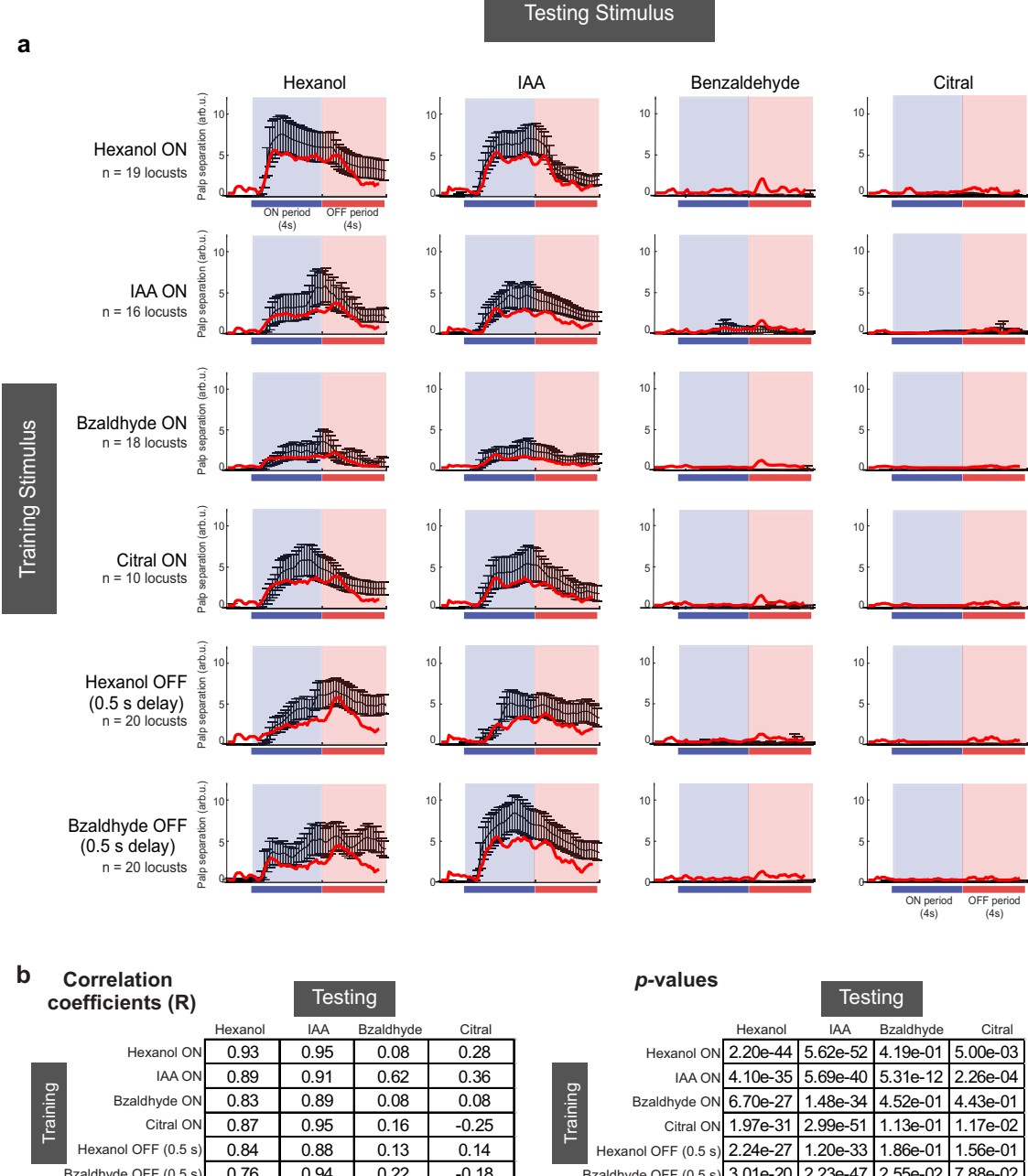

**Fig. 7 | Predictable behavioral response dynamics, cross-learning, and generalization between trained odors. a** Summary of observed and predicted POR responses for six different training conditions are shown: row 1—ON-trained with hexanol, row 2—ON-trained with isoamyl acetate, row 3—ON-trained with benzaldehyde, row 4—ON-trained with citral, row 5—OFF-trained (0.5 s gap) for hexanol and row 6—OFF-trained with benzaldehyde. The number of locusts tested in each training paradigm is shown on the left. Responses of the trained locusts were examined for all four odorants during the unrewarded testing phase. The mean POR for each odorant is shown in black and error bars indicate s.e.m. Colored bars indicate odor ON and OFF periods. Red traces on each plot show PORs produced by a linear regression model that used ensemble PN activity for the four different odorants as inputs (see "Methods"). All source data are provided as a Source Data file. **b** The two tables show the correlation between the predicted POR versus the observed behavioral response dynamics (R, top table) and significance (P value, bottom table) (red traces in (**a**)). Similar to the convention in (**a**), each row corresponds to one training paradigm and each column shows one test odor. The significance of the correlation values were determined by comparing the obtained values with a t-statistic with n−2 (158 time points) degrees of freedom.

response clusters that were distinct from the appetitive response clusters. As a direct consequence of this spatiotemporal organization of neural responses, the innate behavioral responses were entirely predictable from neural responses during either of these epochs but using distinct subsets of neurons.

Furthermore, our results indicate that delivering gustatory rewards during ON and OFF response epochs of odorants with positive appetitive valences alone resulted in successful Pavlovian conditioning. Reinforcing non-appetitive odorants did not generate successful odor-reward associations, but resulted in an increase in behavioral responses to other odorants with a positive valence. Notably, a linear model could map neural responses evoked by the odorants onto behavioral response dynamics and cross-associations learned. In sum, our results reveal a spatiotemporal coding logic that supports encoding both innate and acquired odor-driven appetitive preferences.

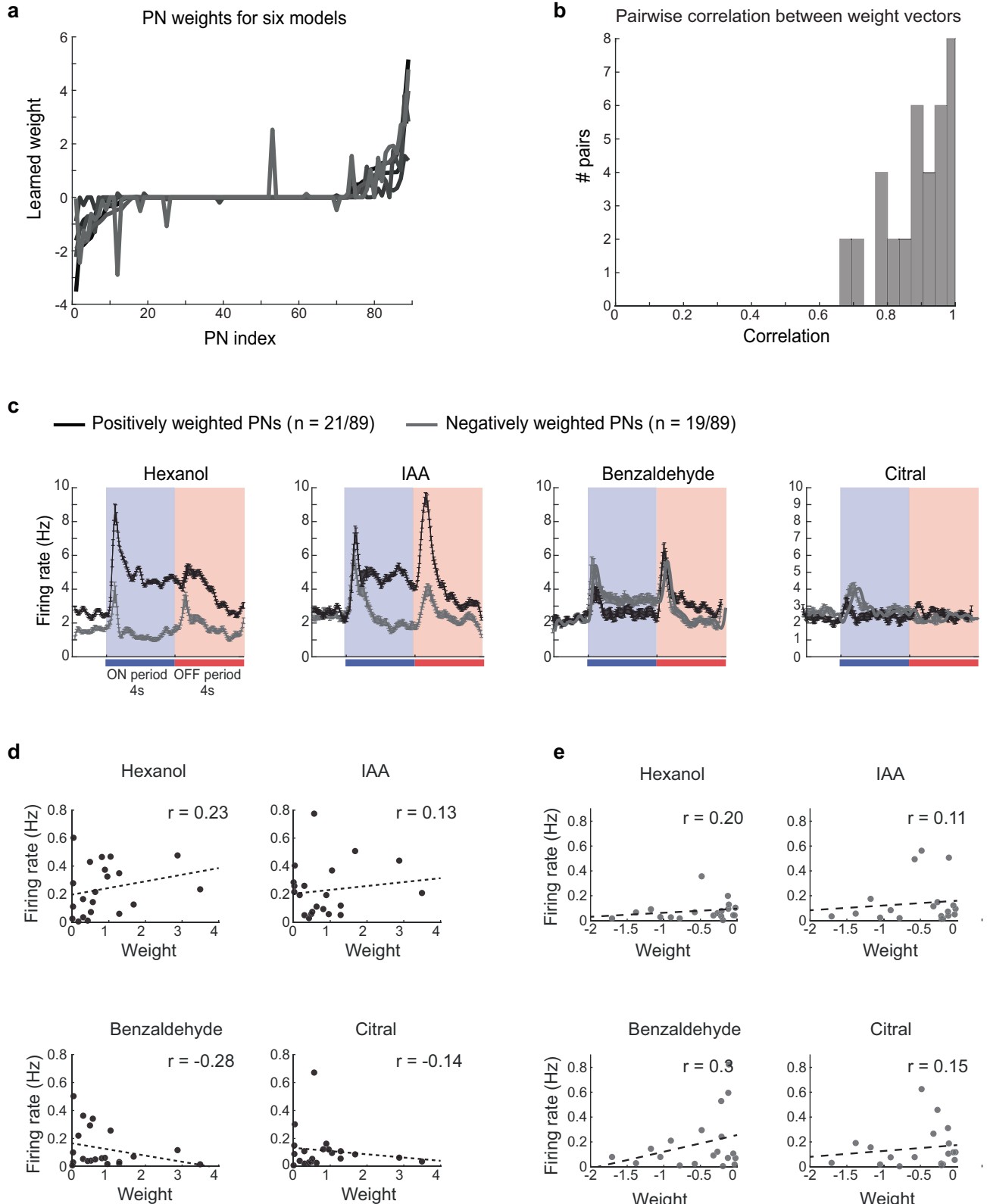

**Fig. 8 | Dissecting the regression model to understand the contributions of individual PNs. a** Weights learned by linear regression models (in Fig. 7) used to predict mean POR responses using ensemble PN activity are shown. The weights were sorted based on the values assigned to each PN for the hexanol-ON-training model. All source data are provided as a Source Data file. **b** The distribution of pairwise correlations between different pairs of weight vectors from panel a is shown. As can be seen, the weights assigned to PNs are highly similar, given the high correlation for all pairwise comparisons. **c** Mean spiking activities of all PNs that were assigned positive (black) or negative (gray) weights are shown. In total 21 PNs were assigned positive weights, 19 PNs received negative weights <0, and the remaining 49 PNs were assigned a weight of 0. Error bars indicate standard deviation. **d** Relationship between the mean firing rate and model weight for PNs assigned positive weights are shown for all four odorants. The correlation coefficient for each distribution is indicated. **e** Similar plot as (**d**) but for PNs assigned negative weights.

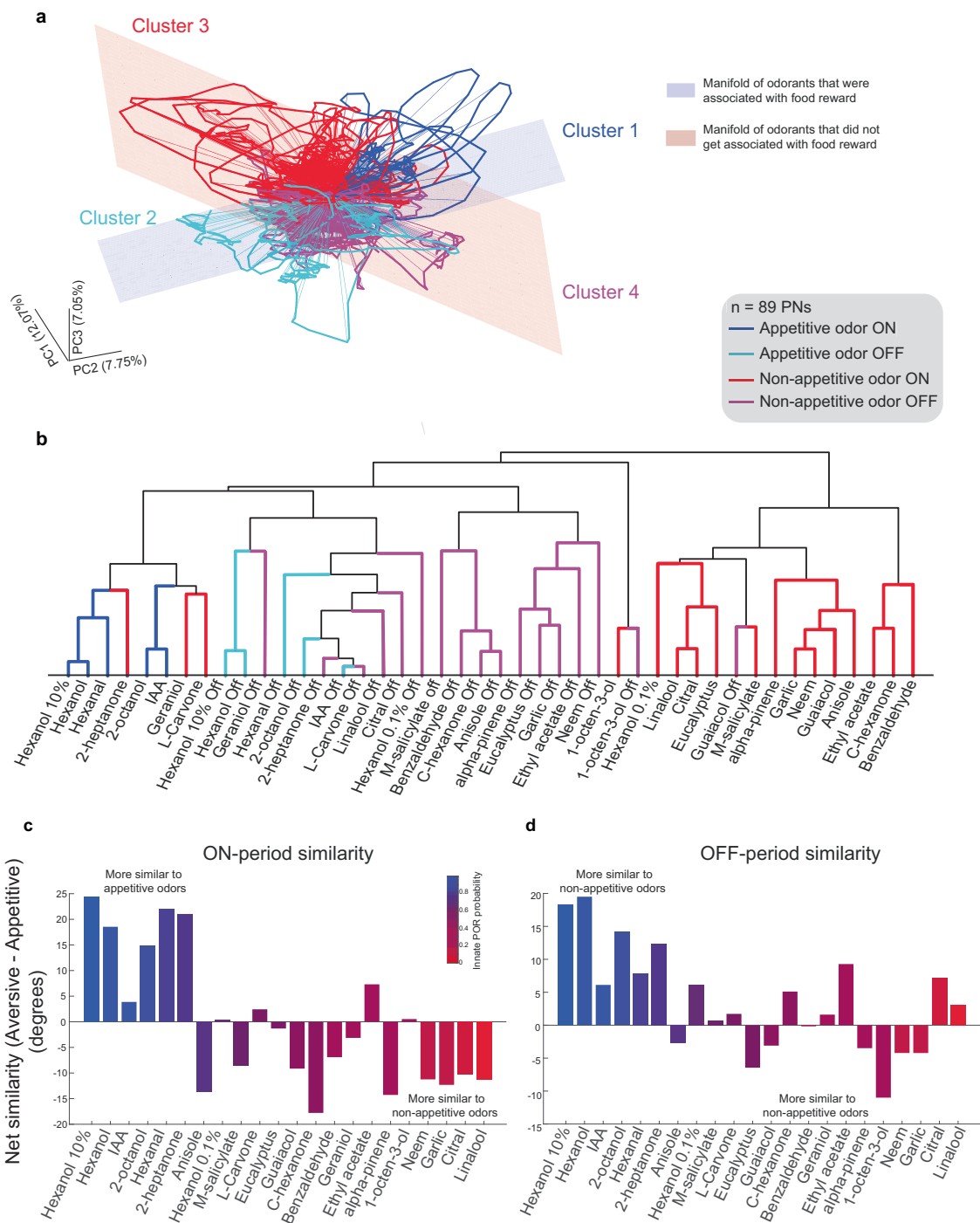

**Fig. 9 | Neural manifolds can explain innate and acquired behaviors.**
**a** Trajectories showing ensemble neural responses during both the ON- and the OFF-periods for all 22 odors are shown along the top 3 principal components (*n* = 89 PNs; see "Methods"). The trajectories were colored as follows: blue—appetitive odorants ON responses, cyan—appetitive odorants OFF responses, red—non-appetitive odorants ON responses, and magenta—non-appetitive odorants OFF responses. Variances in odor-evoked responses of appetitive odorants were not uniformly distributed but confined to a subspace and are schematically shown as using a linear plane (colored in blue and encompasses appetitive ON and OFF neural ensembles). Similarly, non-appetitive odorants ensemble responses are confined to a distinct neural manifold schematically shown in red. All source data are provided as a Source Data file. **b** Dendrogram showing the categorization of odor-evoked ON and OFF responses of all 22 odorants in the panel are shown. A correlation distance metric was used to assess the similarity between 89-dimensional PN response vectors. Coloring convention similar to (**a**). Note that the appetitive and non-appetitive odorants form

supra-clusters, each containing ON and OFF responses sub-clusters. **c** Plot showing the average similarity of an odorant to other appetitive and non-appetitive odorants. For each odor, we took the ON response across 89 PNs (i.e., 89-d vector) and computed its cosine similarity with the ON responses for all other odorants. Twenty-one such angles were obtained for each odorant (ignoring self-comparison; Supplementary Fig. 5). The angles obtained from comparison with appetitive and non-appetitive odorants were grouped, and the average for each group was taken. The difference between the average angles for each group (non-appetitive minus appetitive) is shown here as a bar plot. The odorants along the *x* axis are shown in order of decreasing innate valence (left to right), and the bars are colored to indicate the probability of innate PORs (Fig. 1). Note that a positive similarity score indicates the odor responses were more similar to appetitive odors while a negative score indicates better pattern-match with non-appetitive odorants. **d** Similar plot as (**c**) but using the OFF responses across all 89 PNs.

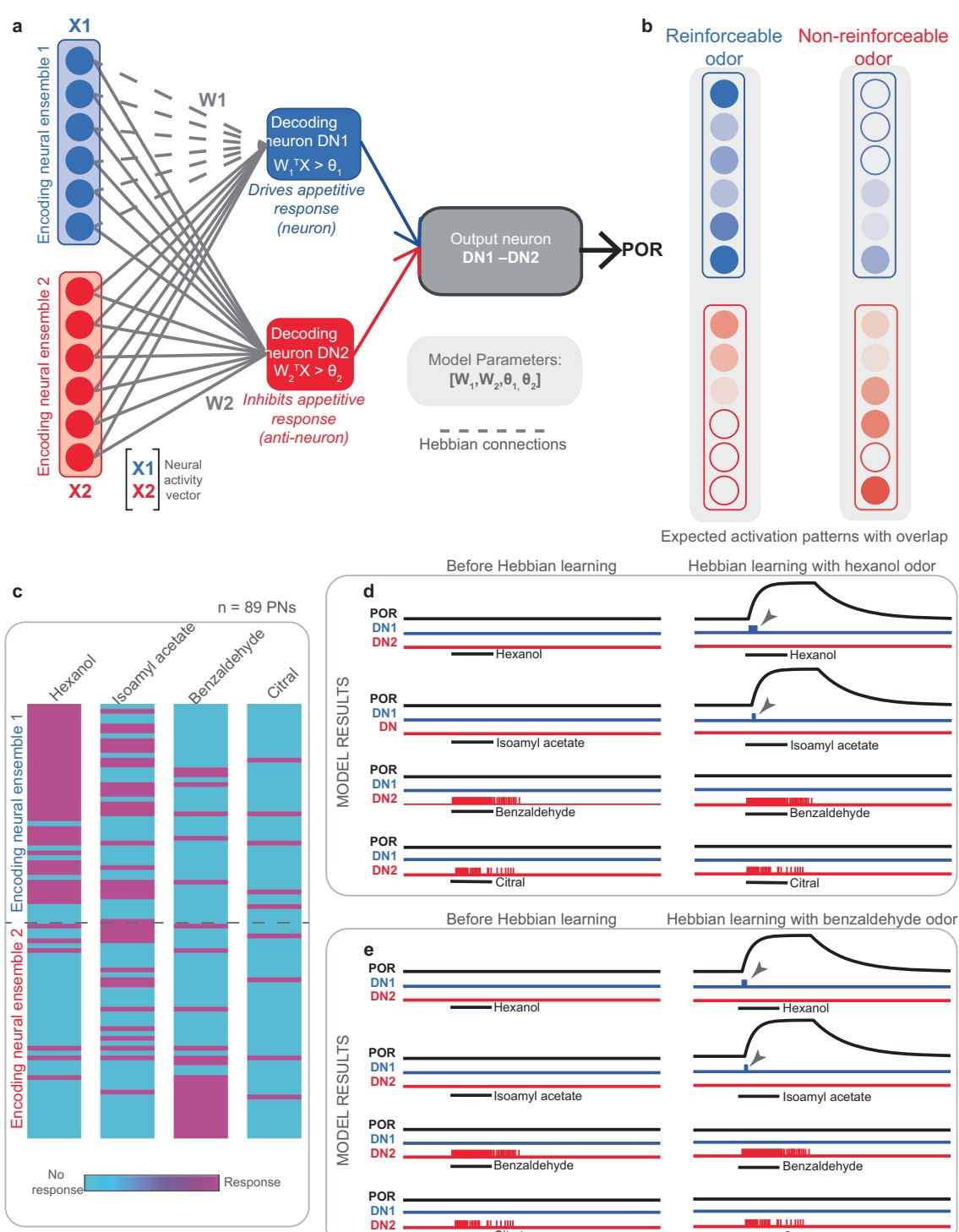

Could the observed appetitive preferences for different odorants be predicted directly from the stimulus/chemical space[34,42,43]? We found that chemical features such as those extracted by nuclear magnetic resonance spectra or infrared spectra did not have good correlations with the overall appetitive preferences for different chemicals on the odor panel (Supplementary Fig. 7). Our results indicate that chemically similar odorants evoked divergent neural responses (isoamyl acetate and ethyl acetate—both esters but opposite valences). Conversely, we found odorants that had different chemical features mapped onto similar appetitive preferences (benzaldehyde and cyclohexanone). Even features such as the vapor pressure that controls the number of molecules reaching the antenna did not have a good correlation with the overall behavioral preference. While this is not an exhaustive list of chemical features that can be extracted, these results appear to indicate that it would be difficult to find a simple linear mapping of the chemical space onto the behavioral space. Similar results have recently been reported in the mouse olfactory bulb[44]. Contrasting the non-linearity between the chemical−neural transformations, a linear mapping was indeed found between neural and behavioral spaces. These results support the idea that neural responses, even in those circuits very early in the olfactory pathway, are organized to generate appropriate behavioral outcomes rather than faithfully represent the chemical features of the odorants.

**Fig. 10 | A simple neural network with Hebbian plasticity recreates our conditioning experiment results. a** Schematic of the network used to examine the effects of associative learning on behavioral PORs. The input neurons (corresponding to PNs) are divided into two non-overlapping groups. While all input neurons connect to both the downstream decoding neurons, only the connections between encoding ensemble 1 and the decoding neuron 1 are plastic and are altered during associative conditioning using a simple Hebbian rule (see "Methods"). All source data are provided as a Source Data file. **b** *A key model assumption:* Appetitive odorants (hex and iaa) that elicited PORs after pairing with food reward should activate the encoding ensemble 1 more. Non-appetitive odorants (bza and cit) that did not elicit PORs after conditioning are expected to activate PNs in the encoding ensemble 2 more. However, the non-appetitive odorants should also activate a few neurons in the encoding ensemble 1. **c** Binary categorization of PN responses in our experimental dataset (*n* = 89 PNs) as responsive or non-responsive to a given

odorant. The PNs are ordered such that those activated by hexanol are at the top and those activated by bza are at the bottom. PNs with peak odor-evoked activity greater than the mean + 6.5 standard deviations of pre-stimulus activity were considered responsive. The same ordering was used to compare the PN response categorization for all four odorants. **d** Left panel: The activity of decoding neuron 1 (DN1) and decoding neuron 2 (DN2) along with the expected POR responses generated before any Hebbian alteration of network weights are shown for all four odorants. *Right panel:* Similar plots but now showing DN1 and DN2 along with predicted POR after Hebbian learning. Only the PN activities during *hexanol* presentations were used for altering the network connections. **e** Similar plots as in (**d**) but showing model outputs before and after learning using PN activity patterns generated by *benzaldehyde* odor. Note that both *hexanol* and *benzaldehyde* learning resulted in increased PORs to *hex* and *iaa*, and no responses to *bza* and *cit*, consistent with our experimental data.

Surprisingly, at the individual neuron level, we found that responses in a small subset of PNs had a strong correlation with the overall innate preference for different odorants (Fig. 2e; correlations >0.75 for 4/89 PNs for ON responses and 2/89 PNs for OFF responses). Such encoding of overall odor valence by individual neurons so early in the olfactory pathway has indeed been reported in other invertebrate models[2–4]. While the simplest model to predict the behavioral outcomes from the neural activity would be to just use a few of these neurons, whether such a model would be robust is unclear. Earlier studies have shown that individual projection neuron responses change unpredictably with changes in stimulus dynamics, intensity, competing cues, stimulus history, and ambient conditions[23,29,45–47]. Notably, the behavioral recognition of odorants was found to remain invariant under a battery of these perturbations[48]. Therefore, a more robust and fault-tolerant model to overcome such variations in neural responses that arise due to natural perturbations would involve a combinatorial readout of the ensemble activity as proposed in our regression analyses.

To understand the appetitive preferences of locusts to different odorants, we used the palp-opening responses that locusts use to grab food. While preferences of individual locusts to the odor panel were idiosyncratic (i.e., varied from one locust to another; see Fig. 1b), as a group they tended to have similar behavioral preferences (Fig. 1f). This simple readout provided a one-dimensional quantitative summary of the innate appetitive preferences for the different odorants used in our panel. We found that a simple linear regression was sufficient to map ensemble neural responses during both stimulus presentation and after termination onto this behavioral dimension. Therefore, we concluded that the neural responses were spatiotemporally formatted to support the generation of innate behavioral outcomes.

Prior studies have shown that the palp-opening responses to an odorant could also be learned through associative conditioning[45,49]. To understand the rules that constrain learning in this paradigm, we screened and identified locusts that did not have any innate responses. We were concerned that repeated exposures to an odorant may induce PORs in these locusts. In this scenario, the PORs observed in the testing phase may not arise from conditioning but rather from sensitization due to repeated exposures to a stimulus. However, our results indicate that when the introductions of the reward were delayed to occur well after the termination of the odorant (hexanol OFF 4 s and benzaldehyde-OFF 4 s paradigms), locusts did not show PORs and maintained their lack of responses to the conditioning odorants (Fig. 6a). We interpreted this result as an appropriate control indicating that locusts did not become sensitized to generate PORs to the conditioned stimulus and that PORs in these locusts were observed only in certain scenarios that suited associative learning.

Our conditioning experiments revealed that only two of the four odorants (hex and iaa) used resulted in a successful association between the odorant and the reward. As a result, locusts responded with PORs to the presentation of these odorants during the testing

phase. We also observed the generalization of the learned PORs to other odorants. Locusts trained with hexanol also showed responses to isoamyl acetate and vice versa (generalization to similar odors). Intriguingly, locusts trained with citral and benzaldehyde also increased PORs to hexanol and isoamyl acetate (cross-learning could also alter behavioral responses to unrelated odorants). We again found that linear mapping between neural and behavioral responses existed and captured all the important trends in our data (Fig. 7a).

We found that delaying reward such that it was delivered either during the presentation of hexanol (ON-training paradigm) or immediately after its termination (OFF-training paradigm; 0.5 s after stimulus termination) both resulted in associative learning. However, we found that the POR dynamics were different between these two training paradigms. We note that locusts in the ON-training paradigm had PORs that were significantly different from those observed in locusts trained using the OFF paradigm. Notably, such nuanced differences in POR dynamics still correlated with the neural response similarity between the test odorant and the conditioned stimulus (Supplementary Fig. 8, red PORs vs black neural response correlations). These results are consistent with the interpretation that the POR trends could be predicted from the overall neural response profile and how they change as a function of time. Taken together, these results suggest that the timing of the reward could be controlled to coincide during different phases of neural response dynamics and such manipulations result in predictable changes in behavioral responses.

Is this learning paradigm still associative learning? Generalization of learning to other untrained stimuli has indeed been reported in other model organisms such as honey bees and ants[50–52]. Such generalization or cross-learning has been noted to be asymmetric, and in some cases generate a stronger response to the untrained odorants[50,51]. Our results are consistent with these prior findings, but as we noted, pairing non-appetitive odorants such as benzaldehyde and citral did not increase PORs to these odorants but increased responses to other appetitive odorants (hex and iaa). The non-specificity of this learning effect, while surprising, raises questions regarding the stimulus features that are associated with the reward. What potential mechanism could provide a neural correlate for cross-learning observed in our conditioning experiments? Our results show that a simple neural network model with Hebbian plasticity was sufficient to map the neural activity we recorded onto the behavioral outcomes observed. Notably, the model required two key features to replicate results: overlap in neural responses between appetitive and non-appetitive odorants, and restricting plasticity to only a subset of the neural network that connected encoding neurons predominantly activated by appetitive odorants. These features nudged the system to become more sensitive to the other appetitive odorants via conditioning, but the behavioral response to non-appetitive odorants was robustly shut down by the non-malleable part of the network. Hence, we believe these results reveal an unnoted feature of associative learning in a sensory modality using a combinatorial coding scheme.

In this study, our datasets comprised neural responses evoked by a panel of diverse odorants and their innate and acquired appetitive preferences. Surprisingly, we found that there exists a theoretical framework that would allow us to integrate these observations and understand the neural underpinnings of behavior. We regarded the ensemble neural activity as a high-dimensional neural response trajectory. Each odor-evoked response trajectory consisted of two non-overlapping segments, one during odor presentation (i.e., ON response), and the other after its terminations (i.e., OFF response). Notably, we found that ON responses and OFF responses evoked by innately appetitive odorants were on or near a low-dimensional subspace or "manifold" (Fig. 9a). Similarly, we found that ON and OFF responses evoked by odorants with negative appetitive valence were on or near a separate low-dimensional manifold in the coding space (Fig. 9a).

We note that neuronal manifolds that encode for different behavioral response motifs have been reported in other model organisms[42,53–55]. In *C. elegans*, these neuronal manifolds appear to arise globally and engage several circuits throughout the entire brain. Importantly, even those neuronal circuits that are directly downstream of sensory neurons were incorporated in these brain-wide dynamics to orchestrate the innate behavioral outcomes[53]. If this is indeed a generic phenomenon, we would expect the spiking response patterns in the early olfactory circuits, such as invertebrate antennal lobe or vertebrate olfactory bulb would be organized into behaviorally relevant neural manifolds. Our results indeed reveal that this is the case at least in the locust olfactory system.

Results from our conditioning experiments indicated that delivering rewards while the odor-driven neural activities were in the "appetitive manifold" resulted in successful conditioning, whereas delivering rewards during responses excursion in the "non-appetitive manifold" did not result in the non-appetitive odorants being associated with the food reward. Interpreted differently, this result suggests that neural activity patterns on some manifolds are conducive for learning, while activity patterns outside this manifold could be harder to learn. Similar results have been reported in the context of motor control in the primate motor cortex[56]. While the motor cortex result arose from constraints imposed by the neural circuitry making certain neural activity patterns difficult to generate, here the antennal lobe network could generate neural response excursions in both learnable and non-learnable manifolds depending on the identity of the stimuli.

The topic of how attractive and aversive odorants are encoded in the antennal lobe has been explored in the fly and the honeybee olfactory systems[2,11–15]. The emerging view from these studies is predominantly that of labeled lined codes, where some glomeruli are activated by attractive odorants and a non-overlapping subset by repulsive odorants. In some cases, the ensemble neural activity across the entire olfactory network during odor presentations has also been shown to predict the overall innate behavioral responses[4,17]. Although the importance of time as a coding dimension has been well established in olfaction[37,57–61], the importance of time-varying neural activity for shaping innate behavioral preferences is not fully understood.

As our results indicate, the neural responses during an odor presentation and after its termination are highly distinct. Therefore, ensemble neural activity traces distinct neural response trajectories (as shown in Fig. 9a) during these epochs. Both ON and OFF responses were correlated with innate odor appetitive preference and therefore could be used to predict the overall behavioral outcomes (Fig. 4). Notably, our results indicate that these ON-OFF neural responses are not randomly scattered in the state space but are highly organized. Both ON and OFF responses evoked by all odorants with positive valence resided on or near a neural manifold that were distinct from the ON and OFF responses evoked by odorants with negative valence (Fig. 9).

What makes an odorant or a sensory stimulus "naturally appetitive" or "innately pleasurable" or the opposite? Once transduced into an electrical signal, it is the neural activity patterns that the downstream/higher circuits have to work with. Are there features/aspects of neural activity patterns that help determine whether the stimulus that evoked it can be behaviorally categorized as "pleasurable" or "unpleasant", or "appetitive" or "unappetitive"? This is the question that we sought to answer. Further, compared to other sensory modalities, olfactory responses are highly combinatorial and dynamic. So how are these temporally evolving patterns of neural activity organized to facilitate this mapping onto the behavioral responses?

Taking a step further back, it does make sense to not have the capacity to link a foul smell with food, or for someone with a peanut allergy, almond/aldehyde smells with food. They are potentially protective mechanisms for keeping the organisms safe. In that sense, it is a straightforward hypothesis to expect the neural coding for these non-appetitive cues to differ from the ones deemed good or edible. How do the responses to these different classes of odorants differ and how soon do they start to diverge from one another? Our results indicate that this divergence begins straight from the first neural circuit that receives the sensory input (i.e., the antennal lobe). While the odor-evoked responses are spatiotemporally patterned, they are still organized in a meaningful way to facilitate this neural-behavioral mapping. Additionally, interpreting ensemble neural response patterns this way also allowed us to understand which odorants could be associated with the gustatory reward.

## Methods

### Odor stimulation

All odorants were delivered at a 1% v/v dilution in mineral oil and placed in dark 60-ml bottles. A constant background air stream (desiccated and filtered) at 0.75 L/min was used as the carrier stream for 0.1 L/min pulses of odorants. A large vacuum funnel placed directly behind the antenna allowed for the constant clearing of the odorants delivered.

For behavioral experiments to quantify innate appetitive preferences, each odorant in the panel was presented for one trial in a pseudorandomized order (Fig. 1a). Odorants were delivered by displacing a 0.1 L/min of headspace in the odor bottles using a pneumatic picopump (WPI Inc., PV-820). Each odor pulse was 4 s and the intertrial interval was 60 s.

For electrophysiology experiments, each odorant was presented for ten trials in a pseudorandomized order. To minimize interference during the experiment, we designed and built a custom olfactometer (SMC valves, NI-DAQ controller) that was automated and triggered using MATLAB. Each odor pulse was 4 s in duration, and the inter-pulse interval was 60 s.

### Behavior experiments to characterize innate palp-opening responses

Young adult locusts (*S. americana*) of either sex were starved for 24 hours before the experiment. Locusts were immobilized within a plastic tube and their compound eyes were covered using black tape. All 20 odorants were diluted to 1% v/v. Hexanol alone was additionally diluted to 0.1% and 10% dilutions (i.e., a total of 22 odorants in the panel). Each locust was presented with all 22 odorants in a pseudorandomized order for 4 s pulses separated by 56 s inter-pulse intervals (60 s between the starts of two consecutive pulses). The experiments were recorded using a video camera (Microsoft). An LED was used to track stimulus onset/offset. The POR responses were scored offline in a blind fashion with no odorant information to remove any experimenter biases. Responses to each odorant were scored a 0 or 1 depending on if the palps remain closed or opened (Fig. 1b). A successful POR was defined as an opening of the maxillary palps beyond the facial ridges as shown on the locust schematic (Fig. 1a).

## Preference index

As noted above, locust responses to each odorant were binarized. The responses of all locusts to an odor were then summed to obtain a Total Score. A normalized score for each odorant was then calculated as follows:

$$\text{Norm\_score}_{odor} = \frac{\text{Total Score}_{odor}}{\text{Total \# locusts}} \qquad (1)$$

The preference index (Fig. 1c) was then calculated for each odorant by performing a median subtraction from the Norm_score as follows –

$$\text{Preference index}_{odor} = \text{Norm\_score}_{odor} - \text{Norm\_score}_{median} \qquad (2)$$

Norm_score$_{median}$ was obtained by calculating the median across all odorants.

## Vapor pressure analysis

Vapor pressure data for 18 odorants were obtained from an online database (The Good Scents Company)[62]. Data for neem and garlic could not be obtained, and these odors were omitted from our analyses in Fig. 1d. Regression analysis was performed between vapor pressure values and the POR Total Scores. An $R^2$ value was obtained using the "fitlm" function in MATLAB (Fig. 1d). One of the odorants in the panel (ethyl acetate) had a vapor pressure much higher than all other chemicals, and hence the weak correlations in Fig. 1d could be driven by this potential outlier. To control for this, a similar analysis was performed in Supplementary Fig. 1b, but using only seventeen odorants (i.e., excluding ethyl acetate).

## Monte Carlo simulations for evaluating behavioral stability

We performed Monte Carlo simulations on the data shown in Fig. 1b. We randomly sampled locusts ("$n$" ranging from 1 to 26) and calculated preference indices for all odors using POR scores using the selected subsets of locusts. For each $n$, we performed 100 such simulations and computed an average preference index, which was then compared with the preferences obtained using all 22 locusts. The mean correlation for each $n$ is shown in Fig. 1f. Error bars indicate the standard error of the mean (s.e.m.).

## Electrophysiology experiments

Young adult locusts of either sex were used for these experiments[63]. The legs and wings were removed, and they were immobilized on a custom platform. The head was fixed into place by a wax cup and the antennae were held in place inside a thin tube using epoxy glue. The cuticle above the brain was cut open, the air sacs covering the brain were removed, and the locusts were degutted to minimize any internal movements. A metal-wire platform was then inserted underneath the brain to lift and stabilize it. Finally, the transparent sheath covering the brain was removed after applying protease enzyme.

Locust brains prepared this way were super-fused with artificial saline buffer, and a reference electrode (Ag/Ag–Cl) was inserted into the saline. Multi-unit recordings were made from the antennal lobe projection neurons (PNs) using a 4 × 4 silicon probe (NeuroNexus) with impedance in the 200–300 kΩ range (Fig. 2a). Data were acquired at a 15 kHz sampling rate using a custom MATLAB program and filtered between 0.3 and 6 kHz using an amplifier system (Caltech) that provided a 10,000 gain.

Offline spike-sorting (IgorPro) was performed using the best four channels recorded[64]. To identify single units (PNs), the following published criteria were used: unit cluster separation >5 noise s.d., the number of spikes within 20 ms <6.5%, and spike waveform variance <6.5 noise s.d. To account for baseline drift and loss of neurons during an experiment, we only included PNs with consistent baseline spiking activity in all 220 trials (22 odors, 10 trials each). We defined a PN as being consistent if its baseline firing rate (during a 4 s period before odor presentation) in all trials was no less than 15% of the maximum baseline firing rate for that PN. A total of 89 PNs were identified using these criteria (originally acquired 131 PNs from 26 locusts).

## PID experiment

We used a fast-photoionization diode (miniPID, Aurora Scientific) to characterize the stimulus delivery dynamics of all odors used in the electrophysiology experiments. Each odor was presented for five trials and PID signals were acquired at 15 kHz using a custom MATLAB program. The mean signals for all odors are shown in Fig. 2b.

## Projection neuron response classification

We defined 4 s of odor presentation as an ON period, and the 4 s immediately following odor termination as an OFF period. PNs were classified as ON-responsive if the firing activity was 6.5 s.d. above the mean baseline (2 s preceding the stimulus) firing activity in at least five of the ten trials during the ON period. Similarly, PNs were classified as being OFF-responsive using a similar metric applied to the OFF period. PNs were classified as "Inhibited" if their firing activity did not exceed 2 s.d. of baseline in any time bin during odor presentation and the mean firing rate during the entire stimulus duration (4 s) was lower than the mean baseline activity (in at least five out of ten trials). These classifications are summarized for all odors in Fig. 2d.

## Dimensionality reduction analysis

We used Principal Component Analysis (PCA) to visualize ensemble PN activity (Figs. 3a and 9a). The spiking activity for each PN during 4 s of odor presentation was averaged across all 10 trials and binned in 50 ms non-overlapping time bins. In this manner, we obtained an 89 PN x 80 time-bin matrix for each odorant. We concatenated these data matrices obtained for each odor to obtain an 89 × 1760 data matrix (80 bins * 22 odors). We then computed a covariance matrix (89 × 89) for this data matrix.

Each 89-dimensional response vector was then projected onto the top three eigenvectors (that captured the highest variance). For visualization, the first time bin was subtracted from each odor to obtain a similar pre-stimulus baseline for all odors. The odor trajectories were smoothed using a three-point moving average low-pass filter.

## Hierarchical clustering analysis

The spiking activity of each PN during 4 s of odor presentation was summed to obtain an 89 × 1 (89 PNs) vector per odorant. Agglomerative hierarchical clustering was performed on vectors for all 22 odors using the "linkage" function in MATLAB. The odors were clustered based on a correlation distance metric, and the farthest pairwise distance between clusters was minimized. The clustering was visualized using the "dendrogram" function (Fig. 3b) after obtaining a leaf ordering using the "optimalleaforder" function.

## Linear regression to predict valence from PN activity

Mean odor-evoked activity for each PN ($n_i$) was used as the input for the linear regressor and the behavioral Norm_score for each odor was used as the output. A softmax layer was added to ensure that the final prediction was always between 0 and 1. A leave-one-out-cross-validation (LOOCV) approach was used, where the model weights were trained using data for 21 odors using gradient descent, and then the neural response for the test odorant was used to predict the behavioral POR preference index. The mean squared error cost function was minimized.

$$\text{Predicted POR} = \text{softmax}\left(\sum_{i=1}^{89} w_i * n_i + \text{bias}\right) \qquad (3)$$

Where $n_i$ is the number of spikes evoked during odor exposure in PNi, and $w_i$ is the weight assigned by the linear regressor for PN i.

As controls for the regressors, the POR preference indices of different odorants were shuffled randomly before training. We used the entire 4 s of PN activities during odor presentation for the ON-regressor, and 4 s of OFF activity immediately following odor termination for the OFF-regressor (Fig. 4 and Supplementary Fig. 2).

## Monte Carlo simulations for electrophysiology
We performed Monte Carlo simulations to gauge the performance of the linear regressors as a function of the number of PNs used for the analysis was varied. To achieve this, we randomly sub-sampled $n$ (where $n$ ranged from 1 to 89) PNs and quantified the predictive performance using mean squared error (MSE). For each $n$, we performed 1000 simulations and reported the average MSE (Supplementary Fig. 2). We performed these simulations for both the ON- and OFF-regressors.

## Behavior experiments—classical conditioning
Appetitive classical conditioning experiments were performed on young adult locusts of either sex starved for 24 h before the experiment. Locusts were immobilized within a plastic tube, their eyes were closed using black tape, and their maxillary palps were painted using a zero-volatile–organic–chemical green paint (Valspar ultra). A brief 20-min buffer period was allowed for the paint to dry and the locust to acclimatize back to baseline activity levels.

Prior to conditioning, each locust was presented with a 4 s pulse of all four odorants used in the experiment (hexanol, isoamyl acetate, benzaldehyde, and citral). If a locust had a palp-opening response to any of these odorants, it was deemed "pre-conditioned" and was discarded from the experiment. A 15-min buffer was allowed between this pre-test and the training phase.

During the training phase, locusts were presented with the training odorant diluted at 1% v/v at a rate of 0.1 L/min diluted in a constant background air stream (desiccated and filtered) of 0.75 L/min. A vacuum funnel placed behind the locust allowed for odor clearance. The odor was presented for 10 s and a food reward (wheatgrass) was presented at 5 s post-odor onset for ON-conditioning. The odor was presented for 10 s and a food reward (wheatgrass) was presented at 0.5 s, 2 s, or 4 s post-odor termination for OFF-conditioning. Six such training trials were performed with an intertrial interval of 10 min. Locusts that met the training criteria (>3 food reward acceptances out of 6) were then evaluated in the testing phase.

During the testing phase, locusts were presented with 4 s pulses of various odorants (at 1% dilution) in a pseudorandomized manner with a minimum interval of 20 min between successive tests. The palp-opening responses of the locusts were recorded using a video camera (Microsoft) at 30 fps. The odor delivery and video acquisition were synced using a custom LabView program.

Locusts were kept on a 12 h day–12 h night cycle (7 am–7 pm day). All behavioral experiments were performed between 10 am and 3 pm to ensure that the training phase coincided with the daily feeding time for the locusts. For each set of experiments, a different group of locusts was used. No locust was re-used across different data sets.

## Palp-tracking algorithm
To accurately track maxillary palp separation, we trained a UNet convolutional neural network using randomized initialization of weights in Keras and Tensorflow[65]. During the training phase, the input into this network was a single channel (green) $128 \times 128$ image cropped around the palps. The outputs were manually labeled palps (as binarized $128 \times 128$ matrices with 1's indicating palps and 0's indicating no palps). We trained the network using the Adam optimizer and binary cross-entropy loss function. We performed image augmentation using the

"imgaug" Python library and trained the network on approximately 2000 labeled frames.

Videos were input into the trained network frame-by-frame and the output was thresholded and binarized using a combination of Otsu, mean, and triangle filters from the "skimage" library. The palp distance for each frame was calculated as the distance between the centroids of the two predicted palps using the "regionprops" function.

## Responsive locusts
Locusts were considered "responsive" to a particular odor if they had a palp-opening response that was >6.5 s.d. above pre-stimulus baseline (2 s) for at least 30 time-frames (1 s) with palp separation >1.5 arb.u. (which was the noise threshold of the tracking algorithm) (Figs. 5b, d and 6a).

## Individual locust responses
For the normalized POR traces shown in Fig. 5c, d, we scaled each locust's response such that 0 corresponded to the minimum palp separation and 1 corresponded to the maximum palp separation the locust had across all test odors. Note that after each training paradigm, we tested locusts on four odors—hexanol, isoamyl acetate, benzaldehyde, and citral.

## Mapping neural responses onto palp-opening response dynamics
PN activity and POR responses (distance between palps) for hexanol, isoamyl acetate, benzaldehyde, and citral were averaged across trials and down-sampled to 10 Hz. For each odor, we used a 2 s baseline, 4 s of odor presentation, and 4 s after odor termination to obtain a 10 s vector (100 elements at 10 Hz). We then concatenated responses from all 4 odors to obtain 400-dimensional vectors. The input data was hence $89 \times 400$ (89 PNs; spiking activity at each time point) and the output was $400 \times 1$ (palp separation at each time point). A regularized model was fitted using "lasso" (sklearn in Python) with an "alpha" value of 0.01. The learned $89 \times 1$ weights were then used with the input data to generate predicted POR responses shown in red in Fig. 7a.

We trained 6 such models for each training condition shown in Fig. 7a. The weights obtained for all 6 models were sorted using the weights from the hexanol-ON model and are shown in Fig. 8a. Figure 8b shows pairwise correlations between each weight vector pair. The weights across all six models were averaged for each PN. 21/89 PNs had a weight >0 and 19 PNs had a weight <0, with the remainder of PNs assigned a weight of 0 due to regularization. The PSTHs of the PNs assigned positive and negative weights are shown for all four odors in Fig. 8c.

## Monitoring neural responses in behaving locusts
We developed a minimally invasive preparation to facilitate the monitoring of projection neuron responses in locusts while they were classically conditioned. In brief, the locusts were immobilized identically to the procedure followed for the prior classical conditioning experiments (see above). A small cut was made in their cuticle to allow access to the antennal lobe, which was stabilized using a metal-wire platform. Finally, the antennal lobe was de-sheathed to allow electrode implantation. The neural recordings were performed similarly to the previous set of electrophysiology experiments.

Before conditioning, we recorded 5 trials of responses to each of the 6 odors used (appetitive—hexanol, isoamyl acetate, 2-octanol; non-appetitive—cyclohexanone, benzaldehyde, citral). After a 15-min gap, we performed the conditioning as follows—locusts were presented with six trials of trained odor (hexanol or benzaldehyde) with overlapping presentations of a food reward (sucrose in water 1 g/10 ml concentration) similar to conditioning methods described above. To minimize the movement of the locust and conserve neural stability, we switched from solid food reward (grass) to liquid food reward (sucrose

in water) and presented it in an automated manner using a pneumatic pump (WPI Inc., PV-820). The intertrial interval was set to 3 min for the training phase. Post-training, we waited for 15 min and then repeated the presentations of all six odors for five trials each. In all blocks of neural recordings, we pseudorandomized the order of odor presentation.

The neural data acquired in these experiments could not be reliably spike-sorted using the approach mentioned above. As a result, we used an alternative approach for processing this dataset[66]. The raw data signals (acquired at 15 kHz) were de-noised using a band-pass between 300 and 6000 Hz followed by clipping of signals 5 s.d. above or below the baseline level. These were then passed through a continuous moving root-mean-squared (RMS) filter with a 20 ms window (DSP toolbox on MATLAB), down-sampled by a factor of 150, smoothed by a 10-point moving average filter, and finally down-sampled by a factor of 5 to produce a temporal resolution of 20 Hz (50 ms, similar to spike-sorted PN responses). The samples were finally baseline subtracted using the mean of 1 s baseline prior to odor presentation (two sample recordings shown in Supplementary Fig. 6a) to obtain the ΔRMS signal. For the PCA analysis shown in Supplementary Fig. 6b, we followed a similar approach as mentioned above. We used the mean of 4 s of odor presentation and 4 s of responses immediately after odor termination to obtain a 160-dimension vector for each odor (8 seconds × 20 samples per second) for each locust. We recorded from 10 locusts each for hexanol and benzaldehyde training experiments and concatenated these neural responses to obtain a final 20 locust × 160 bin response matrix for each odor during both the pre- and post-training periods.

### Hebbian neural network model

The architecture of the model matched the schematic shown in Fig. 10a. The input to the model was the spiking activity across the 89 neurons from our electrophysiology dataset. The weights were initialized as follows:

Connection onto the Decoding Neuron 1 (DN1):

$$\mathbf{W_1} = \left(\mathbf{X}\mathbf{X^T}\right)^{-1}\mathbf{X}\mathbf{Y_1} \tag{4}$$

X denotes the matrix of neural activity. Each column of **X** represents trial-averaged firing activity across 89 PNs in a 50 ms time bin. Neural responses before, during, and after the termination of all four odorants (hex, iaa, bza, and cit) were included. **Y₁** is a binary row vector with 1's only during those time bins when hex and iaa were presented.

Connection onto the Decoding Neuron 2 (DN2):

$$\mathbf{W_2} = \left(\mathbf{X}\mathbf{X^T}\right)^{-1}\mathbf{X}\mathbf{Y_2} \tag{5}$$

Where **X** is the same neural data matrix as before. **Y₂** is a binary row vector with 1's only during those time bins when bza and cit were presented.

The threshold $\theta_1$ was to be higher than any value generated by **X\*W₁**. This was done to ensure that the Decoding Neuron 1's output was zero for all odorants before learning (i.e., no generation of PORs for any input). The threshold $\theta_2$ was to be the half-max value of **X\*W₂**. This allowed bza and cit to drive the output of the Decoding Neuron 2.

The 89 neurons were divided into two groups of nearly equal sizes. Those neurons that had more responses to hex were assigned to one group (Encoding Neural Ensemble 1), and those with more responses to benzaldehyde were assigned to the second group (Encoding Neural Ensemble 2) (see Fig. 10c). Only those connections between neurons with stronger hex responses and connecting to Decoding Neuron 1 (DN1) had a plastic connection that could be modified through a simple Hebbian rule.

During Hebbian learning, only connections onto the Decoding Neuron 1 (DN1) were allowed to change based on the following update rule:

$$\text{Conditioning with hexanol}: \mathbf{W_1} = \mathbf{W_1} + \delta \cdot \left(\mathbf{X} * \mathbf{N_{hex}} * \mathbf{R}\right) \tag{6}$$

Where $\delta$ is the learning rate (set to 0.25), **N_hex** is a binary vector with 1's for neurons responding to hex and part of the Encoding Neural Ensemble 1 and 0's for all other neurons (shown in Fig. 10c; i.e., neurons in the top half of vector that responded to hexanol), and $R$ is the food reward (set to 1) only during those time bins when hexanol was presented.

$$\text{Conditioning with benzaldehyde}: \mathbf{W_1} = \mathbf{W_1} + \boldsymbol{\delta} \cdot \left(\mathbf{X} * \mathbf{N_{bza}} * \mathbf{R}\right) \tag{7}$$

Where $\delta$ is the learning rate (again set at 0.25), **N_bza** is a binary vector with 1's for neurons responding to benzaldehyde and part of the Encoding Neural Ensemble 1 and 0's for all other neurons (shown in Fig. 10c; i.e., neurons in the top half of the vector that responded to benzaldehyde), and $R$ is the food reward (set to 1) only during those time bins when benzaldehyde was presented.

### Reporting summary

Further information on research design is available in the Nature Portfolio Reporting Summary linked to this article.

## Data availability

All data presented in this paper are publicly available in Figshare (https://doi.org/10.6084/m9.figshare.22656154). Source data are provided with this paper.

## Code availability

Custom codes used to generate figures in this paper are publicly available along with the datasets in Figshare (https://doi.org/10.6084/m9.figshare.22656154).

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

## Acknowledgements

We thank members of the Raman Lab (Washington University in St. Louis) and Dr. Debajit Saha (Michigan State University) for their feedback on the manuscript. We thank Pearl Olsen for insect care. This research was supported by NSF (1453022, 1724218, 2021795) and ONR (N00014-19-1-2049, N00014-21-1-2343) grants to B.R.

## Author contributions

R.C. and B.R. conceived the study and designed the experiments/analyses. R.C. performed all the experiments and analyzed the data. B.R. developed the model. R.C. and B.R. wrote the paper. B.R. supervised all aspects of the work.

## Competing interests

The authors declare no competing interests.
