## [Peer Review File · Nature Communications]

Neural manifolds for odor-driven innate and acquired appetitive preferencesReviewers' comments:

Reviewer #1 (Remarks to the Author):

The study by Chandak and Raman investigates the structure of neural responses to odours in the locust antennal lobe and their relation to learning. The study is well conducted, and the manuscript is relatively clearly written, although the figure could be improved. I have some questions and concerns about the learning (conditioning) part of it, especially in terms of the behaviour, neural data analysis and interpretation (i.e. the causality of who drives what). Some of their first findings follow nicely from Stopfer et al Neuron 2003, whereas the conditioned responses part is novel.

MAIN COMMENTS:

1. Did the authors check that the same odour evokes the same type of POR response on the same animal? Fig 1B shows some degree of consistency across animals, and Fig 2a shows neural responses to different presentations of the same stimulus to each animal (I think), but I can't find a within-animal analysis of the POR response. The authors use a reasonable approach to account for across-locust differences (lines 143-50), but I'm not sure how they handled within-locust consistency.

2. When does the behavioural response (POR) start with respect to odour presentation? If it is soon (i.e., <4 s) after odour presentation, then having good within-locust consistency could be critical: If part of the PN neural activity depended on the behavioural outcome (not only on the stimulus itself), then the authors would have to be very careful when pooling neurons together for analysis (e.g., for Figure 3), or even when computing their tuning to different stimuli (Figure 2)

3. Some clarifications on Figure 2: Panel b: what do "leftmost" and "rightmost" refer to? Panel c: indicate that the 4 seconds are odour presentation. Do each of the PN A, PN B, and PN C panels just show the response of one example neuron from each class 10 presentations of each odourant? The text (Lines 103-108) and the caption seem at odds with each other. Panel e: I assume the authors binned the spikes to compute firing rates and used that to find correlations with how appetitive each odour was found to be in Figure 1? Panel e: It would be nice to show the two histograms overlaid on top of each other to highlight the absence of a clear relationship. Finally, do you need the PN A, B, C labels? Why not just call them things like on, on+off that can be readily recognised?

4. Figure 4d,e present an interesting analysis trying to assess whether models trained to predict the locust response using neural activity during stimulus presentation ('on') were similar to those trained using neural activity following stimulus presentation ('off'). While the analyses are quite convincing, the authors should try a simpler one: use the 'on' models to predict POR during the 'off' phase, and viceversa. In addition the authors could even compute the overlap between the models (e.g., see the metrics in Elsayed, Lara et al Nature Comm 2016 or Gallego et al Nature Comm 2018).

5. Conditioning training: Figure 5c,d: some locusts didn't learn the stimulus response associations; were they excluded from subsequent analyses? (They should in my opinion, although I'm not sure they were given how large the error bars are in all plots in c and d). Also, many 'on' locusts kept their POR after the stimulus was removed and viceversa: many 'off' locusts started their POR during the on period, rather than waiting to its end. This surely has to impact the analysis of generalisation of odourants between on and off, potentially hindering the next studies on the paper; how did the authors manage this?

6. I'm confused about Lines 208 onwards. The authors said in the previous section that only innately appetitive odourants can be conditioned, which means generating a POR. Yet, here they say that it is surprising that generalising a conditioned POR doesn't work when the odourants are non-appetitive (Lines 208-210). Wasn't that to be expected? Moreover, in Figure 6a, the authors have, e.g., a Citral ON training row: Are these the animals who never learned the conditioned response as per their Figure 5c? If that's the case, what is the rationale to include them if they were trained on that association but only one out of 20 managed to learn it?

7. Generalisation between odours: Lines 214-16: Clarify these sentences; at least I couldn't grasp how the analyses were performed. Line 224: The authors say that the weights were similar, but I gather they mean 'their amplitude distributions were similar' based on the figure they cite? Clarify and include a similar comparison to that in Figure 4 to compare models. Figure 6d is very hard to interpret without error bars and an additional neuron-by-neuron analysis (e.g., a scatter plot comparing firing rates and model weights if they want to show a dependence between these two variables), therefore I'm not fully convinced by lines 224-237.

8. Figure 7 is interesting but I think the authors should add additional analyses to substantiate their finding about relative orientation of the manifolds underlying ensembles 1, 2, 3, and 4. If they want to discuss their overlap they should compute their principal angles (Gallego et al 2018) or use another subspace overlap metric (Elsayed, Lara et al 2016). The decision tree analysis in b is nice but a better (or at least additional and very useful) way to show clustering is to compare distances between neural trajectories within and across groups of odourants+responses (the four "ensembles" in 7a).

By the way, I think the authors used the term ensemble to refer to population of neurons and here they seem to use it for clusters. These two should be disambiguated.

9. Abstract “Only odourants that evoked neural response excursions in the appetitive manifold were conducive for learning”. This is true but I don’t think the authors can’t conclude that the similarity in neural responses was key for that, since conditioning learning was only possible for naturally appetitive odourants, which I think probably happened because of their behavioural readout (POR). Therefore, one can’t conclude whether it’s the similarity in the antennal lobe neural ‘representations’ that drives this trend in learning, or that to the animal these associations are easier to learn because they are innately pleasurable (I recommend the authors to read Gómez-Marín, Ghazanfar 2019). This comment also applies to lines 348-58. I’d change this in the introduction, and would like to read the authors’ thoughts on it.

MINOR COMMENTS:

ABSTRACT

- “could the neural response organization impose constraints on learning?” -> This sounds a lot like the title of a previous paper showing this for short term learning

- “Similarly, innately non-appetitive odorants evoked responses that were separable yet confined to another neural manifold.” -> clarify the relationship between the two

INTRODUCTION

- The last paragraph of the Introduction is less clear than the rest; please rewrite. I’d also encourage the authors to include a small summary of their results

RESULTS

- In Fig 1A, do palps open or close supposed to correspond to odours A and B respectively? From odour B it seems that palp opening may precede odour presentation although I guess that isn’t the case? —I don’t work with locusts myself

- I personally find the term valence a bit confusing since it could be related to the chemical properties of the odourant molecules —why not just use appetitive as the authors often do?

FIGURES

- Some figure axes (e.g., Fig 2f, Fig 4) are too little to be read easily. The insets in Figure 4 are impossible to read on my computer.

- The authors state that odour presentation was pseudorandom: did they present the odours on the same order for all locusts, or did they change the sequence across locusts?
- Why do the authors think there's a neutral odourant (2-heptanone) on the same exact branch of the tree as almost all the appetitive odours?
- I'm guessing the authors used different locusts for on and off training, but they should clarify this in the text, as well as give additional details on the training (number of sessions, repetitions...) I also encourage them to consider adding PN responses during On and Off stimuli to the paper.
- Figure 6c: 3D bar plots are very hard to read. Use heatmaps/confusion matrices
- Figure 5 caption: "only innately appetitive odourants can be reinforced..." should be worked into the text describing those results: while the authors say that only some odourants can be conditioned, it's key that it's the innately appetitive odourants
- Lines 240- "The variance in neural responses" -> neural trajectories or trajectory described by the neural activity.

DISCUSSION:

- Lines 280-1: This sentence is very useful and could have been included in the Results (or a variation of it)
- What is the "spatiotemporal coding logic"? This is never defined and to me it just means that different odourant+response combinations are associated with activity patterns that live in different parts of neural state space
- Lines 301-12: while I agree with the overall message, the logic seems to be laid out like a bit of a straw man? (Of course it's the authors' paper and it's up to them to decide)
- Lines 374-77: I'm a bit confused about this statement since Stopfer, Laurent et al showed almost 20 years ago that the same system the authors are studying has responses organised in a "lower dimensional manifold".

REFERENCES:

Elsayed Lara et al 2016: <https://pubmed.ncbi.nlm.nih.gov/27807345/>

Gallego e al 2018: <https://www.nature.com/articles/s41467-018-06560-z>

Gómez-Marín and Ghazanfar 2019: [https://www.cell.com/neuron/fulltext/S0896-6273\(19\)30790-1](https://www.cell.com/neuron/fulltext/S0896-6273(19)30790-1)

Reviewer #2 (Remarks to the Author):

In this manuscript, Chandak et al investigated the odor responses of locust antennal lobe projection neurons (PNs), using a combination of electrophysiological recordings, multi variate statistical analysis, and behavioral experiments. They showed that locusts exhibit stereotyped behavioral responses to appetitive and aversive odors. Moreover, they also showed that the appetitive and aversive odors are encoded by distinct activity patterns. Finally, they showed that locusts can perform associate learning using appetitive odors .

The experimental quality of the work appears to be at the level of the standards of the field, and the manuscript in general is easy to read. However, this study clearly lacks the conceptual advancement that is “in my view” needed for a Nature Communications paper. The results are in general a reiteration of several previous observations that we shown in different insect species, primarily in fruitflies and honey bees, and to some extent also locusts. Moreover, I think the present study is very descriptive, and do not include any mechanistic insight into the phenomenon, or the underlying circuitry. Unfortunately I also did not see any attempt from the authors to go beyond this descriptive work, and highlight why shall we be interested in this phenomenon observed in locusts, while several other similar work in fruitflies and honeybees were published many years ago.

On a finaly note, the authors do not use the term ensembles correctly, and somehow use the term as a replacement for population activity. In my view, no ensembles of neurons are studies here, and I strongly recommend the authors to reconsider their terminology in their future revisions.

Reviewer #3 (Remarks to the Author):

This manuscript by Chandak and Raman uses the locust olfactory system to examine the relationship between olfactory projection neuron (PN) responses and an appetitive behavior (palp opening). The authors measure palp opening in response to a panel of odorants and compare these to neural responses to the same stimuli. They find that neural responses to odorants that evoke strong palp opening and those that suppress palp opening are clustered, such that palp opening can be predicted as a linear function of PN responses. They further explore plasticity of the palp opening behavior by pairing odors that either evoke or suppress palp opening with a food reward at different latencies relative to the stimulus. Surprisingly they found that pairing food with either type of odorant increased palp opening but only in response to odors that innately drive this behavior. The authors interpret their results through the lens of neural manifolds.

Overall this is an intriguing dataset that convincingly links specific patterns of PN activity to palp opening behavior. However, I have concerns about the design and interpretation of the behavioral experiments as well as the overall interpretation of the results.

Major concerns:

It appears that the palp opening behavior is analog and dynamic and can be measured quantitatively as shown in Figure 5, however the behavioral measurements shown in Figure 1 are binarized and only one measurement is made per odor per animal (!). I do not understand why the behavior cannot be measured repeatedly in the same animal (as the authors argue that it does not adapt) with the same precision shown in figure 5 to provide measures of behavioral probability, amplitude, and dynamics. As the authors wish to argue that odor OFF responses are also related to the behavior, the latency and dynamics of the behavior across odorants and animals appears to be key. (i.e. responses to odor OFF cannot be causing the behavior if they occur after the behavior and any correlation between these two would have to reflect some other correlation between OFF responses and prior neural activity).

A related point is that the quantification of learning shown in Figures 5 and 6 would be more convincing if the authors showed a direct comparison of mean behavioral dynamics in trained and untrained locusts. Here they are relying on the fact that they chose locusts that did not show an initial reaction to the appetitive odorants, however it seems difficult to completely rule out some kind of facilitation or arousal mechanism to account for the “learning” shown here.

The authors argue that pairing reward with odor offset results in different behavioral dynamics as they show in Figure 5d. However in Figure 6, it appears that IAA ON training also results in hexanol responses with a delayed peak, and benzaldehyde OFF training results in IAA responses with normal latency. These variations make it challenging to conclusively link neural dynamics to the dynamics of the behavior.

Overall it does not seem that the pairing done here produces a learned odor-reward association, rather the food pairing seems to facilitate an innate behavioral response to a fixed set of odorants. Although the authors note this in their Discussion, it seems that the most parsimonious mechanism for such plasticity would be modulation of connections downstream of the antennal lobe onto motor or premotor centers. In this view, a relatively fixed pattern of ORN activities is linked to the POR, but the strength of this connection is variable. Repeated food exposure could potentiate this connection to elicit stronger behavior in response to the same innate odorants, regardless of which odor was paired with food. The one piece of data arguing against this interpretation is the delayed OFF pairing with hexanol, but this seems like a lot of weight to place on this result in the context of all the other findings.

Finally the authors argue in Figure 7 that only odors that evoke activity in the “learning manifold” can be associated with food reward however this seems to be in conflict with their statement in Fig. 6 that locusts trained with citral and benzaldehyde showed palp opening to hexanol and isoamyl acetate. Again, the simplest interpretation is that no odor-food association is formed but rather that certain set of PN activities can evoke palp opening and this response can be potentiated by repeated food reward or potentially by food-odor pairing. A good control here might be to expose a set of locusts to repeated food reward with no odor and observe any change in the probability of response.

Given these concerns I am not convinced by the authors’ statements about a neural manifold for learning although I think they show convincingly that the POR is driven by a specific pattern of PN activity that can be described either using a regression model or by a manifold.

Minor concerns:

— its a small point but I am not sure palp opening is equal to having positive valence. Data from several labs suggests that different appetitive behaviors can be differentially driven by different patterns of odor-evoked neural activity (e.g. Jung and Bhandawat 2015).

— the linear regression model shown in Figure 4 is closely related to the vector valence coding model proposed by Badel and Kazama 2016. Both studies argue that the strength of a specific behavior can be predicted as a linear combination of PN activity where each PN has a relative valence weight either promoting or suppressing the behavior.

— I am a bit confused how the OFF activity can predict the behavior if the behavior typically starts while the odor is still going (as shown in Figure 5). Is it possible that OFF activity is correlated with earlier patterns of activity such as suppression below baseline?

— the authors propose building a model in which behavior is predicted from a relatively sparse set of PN activities but then reject this model out of hand. I think it would be interesting to explore this model quantitatively and see how many PNs are required to match the performance of the whole ensemble in predicting behavior. This might be informative about the types of activity that encode the behavior.

— Figure 6b axis labels are extremely small and hard to read

Reviewers' comments:

Reviewer #1 (Remarks to the Author):

The study by Chandak and Raman investigates the structure of neural responses to odours in the locust antennal lobe and their relation to learning. The study is well conducted, and the manuscript is relatively clearly written, although the figure could be improved. I have some questions and concerns about the learning (conditioning) part of it, especially in terms of the behaviour, neural data analysis and interpretation (i.e. the causality of who drives what). Some of their first findings follow nicely from Stopfer et al Neuron 2003, whereas the conditioned responses part is novel.

MAIN COMMENTS:

1. Did the authors check that the same odour evokes the same type of POR response on the same animal? Fig 1B shows some degree of consistency across animals, and Fig 2a shows neural responses to different presentations of the same stimulus to each animal (I think), but I can't find a within-animal analysis of the POR response. The authors use a reasonable approach to account for across-locust differences (lines 143-50), but I'm not sure how the handled within-locust consistency.

Rebuttal Figure 2: Palp-opening responses for multiple presentations of an odorant in untrained (naïve) locusts. We recorded innate palp-opening responses (POR) for six repetitions of hexanol in untrained/naïve locusts (n = 20). The responses are summarized in this figure. Each row shows responses for one locust with white boxes indicating a POR and black boxes indicating no POR.

We thank the reviewer for their suggestions and comments on the manuscript. As recommended, we performed an additional set of experiments to evaluate the consistency of POR responses to repeated presentations of the same odorant. We recorded the innate palp-opening responses (PORs) of untrained locusts to six presentations of hexanol (1% v/v). The results of this experiment are shown in Rebuttal **Figs. 2, 3**. As can be seen, locust responses to multiple presentations of the same odor are highly consistent across trials, indicating strong within-animal consistency. 12/20 locusts performed PORs in all six trials whereas 8/20 locusts performed PORs in 5/6 trials.

2. When does the behavioural response (POR) start with respect to odour presentation? If it is soon (i.e., <4 s) after odour presentation, then having good within-locust consistency could be critical: If part of the PN neural activity depended on the behavioural outcome (not only on the stimulus itself), then the authors would have to be very careful when pooling neurons together for analysis (e.g., for Figure 3), or even when computing their tuning to different stimuli (Figure 2)

We computed the latency of innate behavioral responses for the untrained locusts assayed in **Rebuttal Fig. 2**. For these locusts, we painted the palps similar to conditioning experiments (**Manuscript Fig. 5**) to allow accurate quantification of response latencies. The response latency was then measured as the time taken after odor onset for the palp separation to exceed 1 standard deviation above the average baseline palp separation. As can be seen, locusts initiate behaviors within 0.6 – 1 s for all presentations of hexanol – well below the 4 s when odor is terminated.

We are not sure whether we fully understood the reviewer's concern regarding PN responses depending on behavioral outcome, since the antennal lobe activity is thought to be driven only by antennal ORN inputs^{7,8} and not by behavioral outputs/motor output such as the palp-opening response.

a Repeated hexanol presentations to untrained locusts

b Response latency of untrained locusts

Rebuttal Figure 3: Palp-opening response dynamics in untrained locusts.

a) The mean palp-opening responses for 20 untrained locusts, presented with six trials of hexanol. Error bars indicate s.e.m.

b) **Response latency of palp-opening responses.** In order to accurately quantify response latency, we tracked the palp-opening responses of locusts for the results shown in **Rebuttal Fig. 2**. We used a similar approach as reported in the conditioning experiments (**Manuscript Figs. 5-6**). The time taken for the palp separation to exceed 1 s.d. of baseline (no odor) separation was used to compute the response latency. The mean latency across all locusts ($n=20$) for each of the six trials are shown here (error bars indicate s.e.m.). As can be seen, the locusts initiate PORs within 0.6 s to 1 s of odor onset for all trials.

3. Some clarifications on Figure 2: Panel b: what do “leftmost” and “rightmost” refer to? Panel c: indicate that the 4 seconds are odour presentation. Do each of the PN A, PN B, and PN C panels just show the response of one example neuron from each class 10 presentations of each odourant? The text (Lines 103-108) and the caption seem at odds with each other. Panel e: I assume the authors binned the spikes to compute firing rates and used that to find correlations with how appetitive each odour was found to be in Figure 1? Panel e: It would be nice to show the two histograms overlaid on top of each other to highlight the absence of a clear relationship. Finally, do you need the PN A, B, C labels? Why not just call them things like on, on+off that can be readily recognised?

We apologize for the confusion caused by the terminology. We used the terms ‘leftmost’ and ‘rightmost’ to describe results within a single panel and not across panels. We used these terms to indicate that the odors were sorted based on innate valence, with the odor placed on the left end having the highest valence and the odor at the right end having the lowest innate valence. In **Manuscript Fig. 2b** in the manuscript, hexanol 10% is plotted on the left end of the bar plot, and this is the odorant with the highest preference. Linalool 1% is plotted on the right end of the bar plot and correspond to the odorants with lowest preference.

The representative PN A, PN B, PN C responses shown in **Manuscript Fig. 2c** are spiking activity observed in individual PNs. Spiking activity before, during and after presentation of all 22 odorants are shown (10 trials per odorant). Therefore, the raster plot comprised of 220 rows (22 odorants x 10 trials). Each of these 3 PNs is one example from the categories of responses we describe in the text in lines 103-108. For example, we found 8/89 PNs that that similar responses as the sample PN identified as PN B, but are showing the responses from just 1 of these 8 PNs. These sample plots serve to show the diversity of responses that can be obtained when looking at individual PNs. However, note that they are still only classifying small fractions of the responses (8/89 PNs for response type shown in PN B, or 11/89 PNs for response type shown in PN C). The PNs were labeled as A, B, C for ease of mapping them to the subsequent plots shown in **Manuscript Fig. 2f**. We believe that labeling them in a more descriptive fashion on the figure may confuse or mislead the readers (as most of these neurons have ON response to some subset of odorants and OFF response another subset).

Yes, the reviewer is correct in how we computed the histograms in **Manuscript Fig. 2e**. For each PN, we computed the average/mean spiking response over 4 s during either the ON or OFF period for all 22 odors. This 22-dimensional vector was then correlated with the 22-dimensional vector containing innate behavioral responses for the odorants to obtain a single correlation value for each PN. This was repeated for all 89 PNs in our dataset for both the ON (blue) and OFF (red) distributions.

We overlaid the two panels from **Manuscript Fig. 2e** and the result is shown in **Rebuttal Fig. 4**. The two distributions are significantly different ($p < 0.05$, t-test), which indicates that the individual PN ON-responses (blue distribution) have a higher correlation to innate behavior than OFF-responses (red curve). However, note that this comparison is made on the overall distributions. Hence, it is possible that for some PNs, the ON-periods correlate with behavior better than the OFF-periods, and vice versa for others.

The main motivation behind **Manuscript Fig. 2e** was to indicate that the responses of individual PNs are highly variable, with a small fraction correlating strongly with innate behaviors (correlations > 0.75 for 4/89 PNs for ON responses and 2/89 PNs for OFF responses). However, even this small fraction of PNs with strong correlations cannot be expected to reliably encode innate preferences for all odors, since it has been shown that individual PNs can significantly alter their responses to odors when encountered in varying contexts or ambient conditions⁹⁻¹³.

Rebuttal Figure 4: Distributions showing the correlations of PN responses with innate behavioral preferences for all 22 odors. Distributions are identical to those reported in **Manuscript Fig. 2e**, but are overlaid for comparison purposes. (p -value = 0.0466, t-test)

4. Figure 4d,e present an interesting analysis trying to assess whether models trained to predict the locust response using neural activity during stimulus presentation ('on') were similar to those trained using neural activity following stimulus presentation ('off'). While the analyses are quite convincing, the authors should try a simpler one: use the 'on' models to predict POR during the 'off' phase, and viceversa. In addition the authors could even compute the overlap between the models (e.g., see the metrics in Elsayed, Lara et al Nature Comm 2016 or Gallego et al Nature Comm 2018).

We thank the reviewer for suggesting this additional control analysis. The results of this analysis are shown in **Rebuttal Fig. 5** below. Both these controls performed quite poorly compared to the original predictions shown in **Manuscript Fig. 4b, c**. For regressors trained using ON period, predictions made on the ON period for the left-out odorant had an R^2 of 0.726, whereas predictions made using the OFF period had an R^2 of 0.291. For regressors trained using OFF period, predictions made on the OFF period for the left-out odorant had an R^2 of 0.489, whereas predictions made using the ON period had an R^2 of 0.156.

We compared the differences between the ON and OFF regression models by analyzing the weights that were assigned to PNs in each of these approaches. We found that these two approaches assigned very distinct set of weights to PNs and quantified this using the correlation analysis shown in **Manuscript Fig. 4e** (strong correlations along the diagonal blocks and weak correlations in the non-diagonal blocks).

Rebuttal Figure 5: Additional controls for regressor specificity

a) We trained regressors to predict behavioral probability similar to the approach in **Manuscript Fig 4**. Note that for this analysis, the regressor learned weights for PNs using the ON-period odor responses but made predictions using the OFF-period odor responses for the held-out odorant. These predictions had an $R^2 = 0.291$ and were poorer than those obtained in **Fig. 4b** ($R^2 = 0.726$).

b) Similar analysis as panel a but weights were learned using OFF-period responses and predictions were made using the ON-period responses of the held-out odorant. These predictions had an $R^2 = 0.156$ and were poorer than those obtained in **Fig. 4c** ($R^2 = 0.489$).

5. Conditioning training: Figure 5c,d: some locusts didn't learn the stimulus response associations; were they excluded from subsequent analyses? (They should in my opinion, although I'm not sure they were given how large the error bars are in all plots in c and d). Also, many 'on' locusts kept their POR after the stimulus was removed and viceversa: many 'off' locusts started their POR during the on period, rather than waiting to its end. This surely has to impact the analysis of generalisation of odourants between on and off, potentially hindering the next studies on the paper; how did the authors manage this?

For all the results shown for the conditioning experiments, we tested all locusts that met our training criterion of accepting food reward in more than 50% of training trials (>3 trials out of 6 training trials). For the final results shown, we retained all locusts that met this training criteria and did not discard any locust's responses. The large error bars in the mean POR plots are driven by some locusts not performing PORs, but also in part due to locusts having different magnitudes of PORs. We did not normalize the responses from single locusts before computing the mean PORs, but have included this analysis as subplot panels in **Manuscript Fig. 5c, d** (heatmaps in blue – yellow).

The reviewer is correct in noting that there are odor-specific nuances and differences in dynamics in the generated POR responses. As we had discussed, while the learned POR responses for 0.5 s OFF-trained locusts were delayed compared to ON-trained locusts, they still initiated within the odor presentation period (avg 2-2.5 seconds to reach 50% peak for hex and iaa; **Supplementary Fig. 4**).

In order to better understand these results, we looked at how the neural similarity between the trained and tested odorant evolved as a function of time (**Rebuttal Fig. 6**, black curves). Next, we compared these results to the mean behavioral responses (red curves) produced by the test odorant for locusts conditioned with the trained odorant. Finally, we computed the similarity between these two curves using a correlation metric (R-values shown on each plot) and found them to be significant in each case (p-values on each plot). This analysis should reveal how the behavioral response dynamics (temporal and magnitude) produced during the test phase is tightly linked to the neural response evoked by the test odorant (i.e., the similarity of the neural response pattern evoked by the test odorant with the response evoked by the training stimulus).

Rebuttal Figure 6: Mapping learned behavioral responses to PN responses.

a) The mean behavioral responses for locusts trained on Hexanol ON and tested for Hexanol is shown in red (right axis). The PN response correlation (black curve, left axis) between the trained condition (Hexanol ON) and test condition (Hexanol ON and OFF) is shown in black (left axis). The correlation between the red behavioral traces and black neural response correlations are indicated in the plot (top right). As can be seen, the two curves have a significant correlation ($R^2 = 0.78$, $p\text{-value} = 8.4 \times 10^{-18}$), indicating a strong pattern match between neural response temporal dynamics and observed behavioral results.

b) Similar plot as in **panel a**, but for locusts trained for IAA ON and tested on Hexanol.

c) Similar plot as in **panel a**, but for locusts trained for Benzaldehyde OFF and tested on IAA.

d) Similar plot as in **panel a**, but for locusts trained for Benzaldehyde ON and tested on IAA.

6. I'm confused about Lines 208 onwards. The authors said in the previous section that only innately appetitive odourants can be conditioned, which means generating a POR. Yet, here they say that it is surprising that generalising a conditioned POR doesn't work when the odourants are non-appetitive (Lines 208-210). Wasn't that to be expected? Moreover, in Figure 6a, the authors have, e.g., a Citral ON training row: Are these the animals who never learned the conditioned response as per their Figure 5c? If that's the case, what is the rationale to include them if they were trained on that association but only one out of 20 managed to learn it?

For all our conditioning experiments, we pre-screened locusts for all 4 odorants – hexanol, iaa, benzaldehyde, and citral. Only locusts that did not produce innate PORs to ANY of the 4 odorants were used for the conditioning experiments. Hence, prior to conditioning, the locusts selected for this set of experiments shown in Figure 6 displayed no innate preferences to these four odorants.

We found that conditioning these locusts with no innate PORs using odorants such as hexanol and iaa resulted in learned responses to these odors, but benzaldehyde and citral still could not be trained to produce PORs. Rather unexpectedly, locusts trained using benzaldehyde and citral started to produce PORs for hexanol and iaa. Hence, we concluded that only some odorants (hex and iaa) could be as effective conditioning stimulus, and these odorants tended to generate innate PORs in other locusts. Our attempts to induce behavior for benzaldehyde and citral resulted in increased behavioral responses to hex and iaa. This was a result that we did not expect and hence, we termed it 'surprising'.

The reviewer is correct in noting that for Citral ON (Fig 6a), we are showing the same locusts as in Fig. 5b, c for Citral training. These locusts met our training criteria (as discussed above) and hence we retained them for the unrewarded test phase. The goal of these experiments was to understand whether different odorants could be conditioned to produce learned responses – and hence, we think a 'negative' result showing a lack of learned responses is still a valid outcome.

Additionally, we wanted to understand if responses learned using this approach were odor-specific. This analysis would not be possible if we tested Citral-trained locusts for Citral during the test phase, and then discarded them if they did not produce a learned response. Also note that during the test phase, the four odors we tested were presented in a pseudorandomized order for each locust.

7. Generalisation between odours: Lines 214-16: Clarify these sentences; at least I couldn't grasp how the analyses were performed. Line 224: The authors say that the weights were similar, but I gather they mean 'their amplitude distributions were similar' based on the figure they cite? Clarify and include a similar comparison to that in Figure 4 to compare models. Figure 6d is very hard to interpret without error bars and an additional neuron-by-neuron analysis (e.g., a scatter plot comparing firing rates and model weights if they want to show a dependence between these two variables), therefore I'm not fully convinced by lines 224-237.

Regarding lines 214-216, the method used for mapping neural responses onto palp-opening response dynamics is described below. It is included as part of the Methods section of the manuscript.

PN activity and POR responses (distance between palps) for hexanol, isoamyl acetate, benzaldehyde, and citral were averaged across trials and down-sampled to 10 Hz. For each odor, we used 2 s baseline, 4 s of odor presentation, and 4 s after odor termination to obtain a 10 s vector (100 elements at 10 Hz). We then concatenated responses from all 4 odors to obtain 400-dimensional vectors. The input data was hence 89x400 (89 PNs; spiking activity at each time point) and the output was 400x1 (palp-separation at each time point). A regularized model was fitted using 'lasso' (sklearn in Python) with an 'alpha' value of 0.01. The learned 89x1 weights were then used with the input data to generate predicted POR responses shown in red in **Manuscript Fig. 7a**.

We trained 6 such models for each training condition shown in **Manuscript Fig. 7a**. The weights obtained for all 6 models were sorted using the weights from the hexanol-ON model and are shown in **Manuscript Fig. 8a**. The inset plot shows pair-wise correlations between each weight vector pair. The weights across all six models were averaged for each PN. 21/89 PNs had a weight > 0 and 19 PNs had a weight < 0, with the remainder of PNs assigned a weight of 0 due to regularization. The PSTH's of the PNs assigned positive and negative weights are shown for all 4 odors in **Manuscript Fig. 8c**.

We apologize for the lack of clarity in these figures. We have re-organized **Manuscript Fig. 6, 7** from the manuscript to clarify these valid confusions and made a new stand-alone figure to analyze this linear mapping approach. **Panel b** in **Rebuttal Fig. 7** (below) shows the pairwise correlation between weight vectors learned (**panel a**) using this linear regression approach for all 6 training paradigms. Similar to the analysis of weights shown in **Manuscript Fig. 4**, we can see that the weights assigned to PNs appear to be highly similar (correlated) across these six modeling approaches.

We found that this approach resulted in 21 PNs (out of 89) receiving positive weights and 19 PNs receiving negative weights. PNs that received the most positive weights responded strongly to both positive valence odorants and had little to no responses to exposures of benzaldehyde and citral. On the other hand, PNs that received the most negative weights responded strongly to the negative valence odorants (benzaldehyde and citral) and had weak responses to both positive valence odorants.

We performed the neuron-by-neuron analysis as requested by the reviewer and this is shown in panels **Rebuttal Fig. 7 d, e**.

We also quantified the firing activity of these individual neurons as shown in **Rebuttal Fig. 7** (below). We found that positively weighted PNs had stronger responses to the appetitive odors

(hexanol and iaa) relative to non-appetitive odors (benzaldehyde and citral), consistent with the PSTH shown in **Rebuttal Fig. 7c** (black traces). Also consistent with PSTHs shown in **Rebuttal Fig. 7c** (gray traces), the negatively weighted PNs had stronger responses for non-appetitive odorants. These two distributions of firing activity (positive and negative weighted PNs) were found to be significantly distinct ($p < 0.005$, t-test).

Rebuttal Figure 7: Linear regression models to map neural responses to behavior

a) We trained 6 linear regression models with sparsity constraints to map PN responses to PORs from 6 training paradigms (**Manuscript Fig. 6**). The weights learned by these models are shown here. PN indices are sorted by the weights assigned for the hexanol-ON model.

b) The distribution of pairwise correlations between different pairs of weight vectors from **panel a** are shown. As can be seen, the weights assigned to PNs are highly similar, given the high correlation for all pairwise comparisons.

c) Peri-stimulus time histograms (PSTHs) of all PNs that were assigned positive (black) or negative (gray) weights are shown. 21 PNs were assigned positive weights, 19 PNs received negative weights < 0, and the remaining 49 PNs were assigned a weight of 0. Error bars indicate standard deviation.

d) Relationship between the mean firing rate and model weight for PNs assigned positive weights are shown for all four odorants. The correlation coefficient for each distribution is indicated.

e) Similar plot as **panel d** but for PNs assigned negative weights.

Rebuttal Figure 8: Analyzing PNs weights when mapping neural responses to behavior.

We further quantified the PN responses for positively and negatively weighted neurons shown in **Rebuttal Fig. 8**. Each circle indicates one PN, and the value along the y-axis indicates the net firing rate for appetitive odors (hexanol and iaa) minus non-appetitive odors (benzaldehyde and citral) for each PN. 21 PNs with positive weight are shown in black circles and 19 PNs with negative weights are shown in gray circles. As can be seen, PNs that received a positive weight tended to have a higher average response to appetitive odors (mean net firing rate = 1.9 Hz), whereas those that were negatively weighted had stronger responses to non-appetitive odors (mean net firing rate = -0.65 Hz). The two distributions were found to be significantly different ($p < 0.005$, t-test).

8. Figure 7 is interesting but I think the authors should add additional analyses to substantiate their finding about relative orientation of the manifolds underlying ensembles 1, 2, 3, and 4. If they want to discuss their overlap they should compute their principal angles (Gallego et al 2018) or use another subspace overlap metric (Elsayed, Lara et al 2016). The decision tree analysis in b is nice but a better (or at least additional and very useful) way to show clustering is to compare distances between neural trajectories within and across groups of odourants+responses (the four “ensembles” in 7a). By the way, I think the authors used the term ensemble to refer to population of neurons and here they seem to use it for clusters. These two should be disambiguated.

We thank the reviewer for their suggestion to clarify and strengthen these points with additional quantitative analyses. We have re-named the groups of PCA trajectories to ‘clusters’ to avoid the unfortunate confusion caused by our nomenclature. ‘Ensembles’ now exclusively refers to populations of PNs. We have added an additional analysis quantifying the neural similarities that we show in the PCA plot **Rebuttal Fig. 9**. This figure is now re-done to include these changes and analyses and is shown below.

As per the reviewer’s suggestion, we further quantified these low-dimensional patterns observed in PCA-space by computing the similarity between odor-response vectors obtained using all 89 PNs (**Rebuttal Fig. 9 c, d**). For each odor, we obtained an 89-dimensional vector to capture the mean response during the ON period and calculated the angle between all such vectors for all odors. Note that a smaller angle (in degrees) represents greater similarity between two vectors/odors. For each odor, we computed 21 angles (22 odors, ignoring self-comparison) and grouped them based on comparison with either appetitive or non-appetitive odors. We then subtracted the average angle of the innately appetitive group from the non-appetitive group to obtain a single similarity angle for each odor. A net positive angle indicates that the odor’s responses were more similar to the appetitive group while negative angles denote better pattern-match with non-appetitive odors. In panel c, we plot this net angular similarity for each odor. The odors are sorted by valence and the bars are colored to denote the probability of innate PORs for the odorant. Overall, these results are quite similar to those obtained from the manifold analyses (clusters 1 and 2), indicating that high-dimensional neural responses agree with the low-dimensional approximations. A similar result was also obtained when using the OFF-period responses to perform this analysis (panel d; similar to clusters 3 and 4).

Figure 9

Rebuttal Figure 9: Neural manifolds can explain innate and acquired behaviors

a) PCA trajectories showing ensemble neural responses during both the ON- and the OFF-periods for all 22 odors are shown along the top 3 principal components ($n = 89$ PNs; see Methods). The trajectories were colored as follows: blue – appetitive odorants ON responses, cyan – appetitive odorants OFF responses, red – non-appetitive odorants ON responses, and magenta – non-appetitive odorants OFF responses. Variances in odor-evoked responses of appetitive odorants were not uniformly distributed but confined a subspace and are shown as using a linear plane (see Methods; plane colored in blue that encompasses appetitive ON and appetitive OFF neural ensembles). Similarly, non-appetitive odorants ensemble responses are confined to a distinct neural manifold schematically shown in red.

b) Dendrogram showing the categorization of odor-evoked ON and OFF responses of all twenty-two odors in the panel are shown. A correlation distance metric was used to assess the similarity between 89-dimensional PN response vectors. Coloring convention similar to panel a. Note that the appetitive and non-appetitive odorants form supra-clusters, each containing ON and OFF responses sub-clusters.

c) Plot showing the average similarity of an odorant to other appetitive and non-appetitive odorants. For each odor, we took the ON-response across 89 PNs (i.e., 89-d vector) and computed its cosine similarity with the ON-responses for all other odorants. Twenty-one such

angles were obtained for each odorant (22 odors, ignoring self-comparison). The angles obtained from comparison with appetitive and non-appetitive odorants were grouped, and the average for each group was taken. The difference between the average angles for each group (non-appetitive minus appetitive) is shown here as a bar plot. The odorants along the x-axis are shown in order of decreasing innate valence going from left to right, and the bars are colored to indicate the probability of innate PORs (**Manuscript Fig. 1**). Note that a positive similarity score indicates the odor responses were more similar to appetitive odors while a negative score indicates better pattern-match with non-appetitive odorants. On average, the probability of PORs appears to reduce as the neural similarity with appetitive odorants diminishes.

d) Similar plot as **panel c** but using the OFF-responses across all 89 PNs.

9. Abstract “Only odourants that evoked neural response excursions in the appetitive manifold were conducive for learning”. This is true but I don’t think the authors can’t conclude that the similarity in neural responses was key for that, since conditioning learning was only possible for naturally appetitive odourants, which I think probably happened because of their behavioural readout (POR). Therefore, one can’t conclude whether it’s the similarity in the antennal lobe neural ‘representations’ that drives this trend in learning, or that to the animal these associations are easier to learn because they are innately pleasurable (I recommend the authors to read Gómez-Marín, Ghazanfar 2019). This comment also applies to lines 348-58. I’d change this in the introduction, and would like to read the authors’ thoughts on it.

We are not arguing that the neural responses in the antennal lobe is what is being reinforced or drives innate preferences directly. Our main point is that, even though the neural responses for different odorants were diverse and dynamic, their organization makes sense when viewed from the point of the behavior elicited (i.e. utility to organism). Furthermore, while such organization is to be expected at higher centers, we were surprised to find this ensemble response structure at the level of a neural circuit directly downstream to sensory neurons.

What makes an odorant or a sensory stimulus, as the reviewer says ‘naturally appetitive’ or ‘innately pleasurable’ or the opposite? Once transduced into an electrical signal, it is the neural activity patterns that the downstream/higher circuits have to work with. Are there features/aspects of neural activity patterns that help determine whether the stimulus that evoked it can be behaviorally categorized as ‘pleasurable’ or ‘unpleasant’, or ‘appetitive’ or ‘unappetitive’? This is the question that we wanted to answer. Further, compared to other sensory modalities, olfactory responses are highly dynamic. So how are these temporally evolving patterns of neural activity organized to facilitate this mapping on to the behavioral responses.

Taking a step further back, it does make sense to not have the capacity to link a foul smell with food, or if you have peanut allergy almond/aldehyde smells with food. They are protective mechanisms for keeping the organisms safe. In that sense, it is a straight-forward hypothesis to expect the neural coding for these unappetitive cues must differ from the ones deemed good for health. How do they differ and how soon do they start to diverge from one another? Our results indicate that this divergence begins straight from the first neural circuit that receives the sensory input (i.e. the antennal lobe). While the odor-evoked responses are spatiotemporally patterned they are still organized in a meaningful way to allow this neural-behavioral mapping. Additionally, we were surprised to find that looking at ensemble neural response patterns this way also allowed us to understand which odorants could be associated with the reward. That is all our point is!

Finally, the POR responses is a highly useful behavior that happens when the organism feeds on its food (usually a blade of grass). In that spirit, we have not reduced the behavior to something that has suppressed its utility to the organism or its relevance to its life. Since the behavior is dynamic, whatever implicit sensory-behavioral loop that must be there are left unperturbed. It is difficult to see this feedback loop in our neural data, as antennal lobe is mostly drive by feed-forward chemosensory signals from the antenna^{7,8}. Hence, from our point of view, we expect that we have retained the spirit of the behavior as recommended in Gómez-Marín, Ghazanfar 2019.

MINOR COMMENTS:

ABSTRACT

- “could the neural response organization impose constraints on learning?” -> This sounds a lot like the title of a previous paper showing this for short term learning

This was unintended. But some interpretations of the results are astoundingly similar even though the paper that the reviewer is referring to was on Monkey BCI and this one on insect olfaction. That manuscript has been cited. While Sadtler et al., 2014 found that re-learning on some neural manifolds were difficult than others, ours indicate that odorants that evoked neural responses in the non-appetitive manifold were not conducive for learning!

- “Similarly, innately non-appetitive odorants evoked responses that were separable yet confined to another neural manifold.” -> clarify the relationship between the two

Each non-appetitive odorant evoked separable odor-evoked neural ensemble responses. However, they had some similarity and therefore still clustered during ON and OFF periods. These clusters were distinct from the cluster of neural responses evoked by appetitive odorants. Hope this clarifies our point.

INTRODUCTION

- The last paragraph of the Introduction is less clear than the rest; please rewrite. I'd also encourage the authors to include a small summary of their results

The reviewer's point is well taken. We broke down the last para in to two to make our points clear. We also included an additional final para to summarize the salient results.

RESULTS

- In Fig 1A, do palps open or close supposed to correspond to odours A and B respectively? From odour B it seems that palp opening may precede odour presentation although I guess that isn't the case? —I don't work with locusts myself

We have updated the figure to avoid this unintended confusion.

- I personally find the term valence a bit confusing since it could be related to the chemical properties of the odourant molecules —why not just use appetitive as the authors often do?

We have never referred to the valence/chemical property of the molecule anywhere in the manuscript. Throughout the manuscript we only referred positive/negative appetitive preference as positive/negative valence (and we defined our usage of this terminology in the manuscript). This we did for brevity and for consistency with other prior studies. We would like to keep this 'jargon' to allow smooth flow of the write up.

FIGURES

- Some figure axes (e.g., Fig 2f, Fig 4) are too little to be read easily. The insets in Figure 4 are impossible to read on my computer.

We have broken out most of the insets from our previous submission into standalone figures to increase the legibility.

- The authors state that odour presentation was pseudorandom: did they present the odours on the same order for all locusts, or did they change the sequence across locusts?

Each locust was presented a different, pseudorandomized sequence of the odors.

- Why do the authors think there's a neutral odourant (2-heptanone) on the same exact branch of the tree as almost all the appetitive odours?

This is a good question, one that we cannot answer at the moment. We hypothesize that this odorant might be an appetitive odorant at lower concentrations, but will switch to being unappetitive as concentration is increased beyond what was examined in this study. This remains to be tested.

- I'm guessing the authors used different locusts for on and off training, but they should clarify this in the text, as well as give additional details on the training (number of sessions, repetitions...) I also encourage them to consider adding PN responses during On and Off stimuli to the paper.

The reviewer is correct – different locusts were used in each set of behavioral experiments reported in this study. No locust was re-used across two or more data sets. We have added this clarification to the Methods section. The **Manuscript Fig. 2F** summarizes information about PN responses during ON and OFF time periods.

- Figure 6c: 3D bar plots are very hard to read. Use heatmaps/confusion matrices

We have updated this figure to heatmaps as per the reviewer's suggestion. Refer Rebuttal Fig. 14 and Manuscript Fig. 6.

- Figure 5 caption: "only innately appetitive odourants can be reinforced..." should be worked into the text describing those results: while the authors say that only some odourants can be conditioned, it's key that it's the innately appetitive odourants

While this is true for locusts shown in **Manuscript Fig.1**, the locusts used for conditioning experiments (**Manuscript Fig. 5**) did not have any innate POR responses to any of the four odorants used. So, technically, for the set of conditioned locusts there was no innate response/preference for any of the odorants. This was the reason why we carefully worded our result section to avoid unnecessary confusion.

- Lines 240- "The variance in neural responses" -> neural trajectories or trajectory described by the neural activity.

This wording has now been clarified in the main text as per the reviewer's suggestion.

DISCUSSION:

- Lines 280-1: This sentence is very useful and could have been included in the Results (or a variation of it)

We have expanded the results section to illustrate this point better.

- What is the "spatiotemporal coding logic"? This is never defined and to me it just means that different odourant+response combinations are associated with activity patterns that live in different parts of neural state space

Our results indicate that both ON and OFF responses, though distinct for each odorant on our panel, could be used to predict the overall behavioral outcomes. Note that ON and OFF responses evoked by an odorant were the most distinct responses that were generated. In other words, the angular similarity between ON responses of an appetitive and a non-appetitive odorant was greater than that of the ON and OFF responses of the same odorant! Nevertheless, the odor-evoked spatiotemporal response patterns were not randomly scattered in the state-space. They were organized such that both ON and OFF responses of odorants with similar overall behavioral preferences resided on or near a neural manifold. Odorants that evoked diverging behavioral responses evoked neural responses with response variance being captured in distinct neural manifolds. This is the spatiotemporal coding logic for mapping neural response onto a 1-d behavioral response that we were referring to. This is now clarified in the manuscript.

- Lines 301-12: while I agree with the overall message, the logic seems to be laid out like a bit of a straw man? (Of course it's the authors' paper and it's up to them to decide)

Since robustness of neural responses is not the focus of this paper, we kept this simple and our logic clear.

- Lines 374-77: I'm a bit confused about this statement since Stopfer, Laurent et al showed almost 20 years ago that the same system the authors are studying has responses organised in a "lower dimensional manifold".

Yes, Stopfer, Laurent et al were the first to show the existence of lower dimensional neural manifolds in their seminal paper! However, these manifolds were viewed/analyzed from an ideal observer/statistical classifier viewpoint and devoid of any behavioral data! We highlighted a few efforts that combined the neural manifolds with behavioral responses.

REFERENCES:

Elsayed Lara et al 2016: <https://pubmed.ncbi.nlm.nih.gov/27807345/>

Gallego e al 2018: <https://www.nature.com/articles/s41467-018-06560-z>

Gómez-Marín and Ghazanfar 2019: [https://www.cell.com/neuron/fulltext/S0896-6273\(19\)30790-1](https://www.cell.com/neuron/fulltext/S0896-6273(19)30790-1)

We once again thank the reviewer for the excellent and in-depth feedback provided. We believe revising the paper to address these issues significantly improved our manuscript.

Reviewer #2 (Remarks to the Author):

In this manuscript, Chandak et al investigated the odor responses of locust antennal lobe projection neurons (PNs), using a combination of electrophysiological recordings, multi variate statistical analysis, and behavioral experiments. They showed that locusts exhibit stereotyped behavioral responses to appetitive and aversive odors. Moreover, they also showed that the appetitive and aversive odors are encoded by distinct activity patterns. Finally, they showed that locusts can perform associate learning using appetitive odors .

The experimental quality of the work appears to be at the level of the standards of the field, and the manuscript in general is easy to read. However, this study clearly lacks the conceptual advancement that is “in my view” needed for a Nature Communications paper. The results are in general a reiteration of several previous observations that we shown in different insect species, primarily in fruitflies and honey bees, and to some extent also locusts. Moreover, I think the present study is very descriptive, and do not include any mechanistic insight into the phenomenon, or the underlying circuitry. Unfortunately I also did not see any attempt from the authors to go beyond this descriptive work, and highlight why shall we be interested in this phenomenon observed in locusts, while several other similar work in fruitflies and honeybees were published many years ago.

On a finaly note, the authors do not use the term ensembles correctly, and somehow use the term as a replacement for population activity. In my view, no ensembles of neurons are studies here, and I strongly recommend the authors to reconsider their terminology in their future revisions.

We thank the reviewer for taking the time to review our manuscript and for the feedback provided. Looking at their comments, we feel that we did not highlight the novelty of the work sufficiently. As the reviewer correctly points out, the topic of how attractive and aversive odorants are encoded in the antennal lobe has been explored in the fly and the honeybee olfactory systems¹⁻⁶. The consensus emerging from these systems is that glomeruli (spherical neuropil structures in the antennal lobe) that are activated by attractive odorants are distinct from those activated by aversive odorants. Hence, a labeled line hypothesis is what has been proposed and represents the current understanding in the field. Even in these systems, neural responses evoked by odorants are dynamic and continue to evolve during and after the presentation of the stimulus. Whether and how the temporal dimension is exploited for encoding behaviorally relevant information is still not understood.

The locust olfactory system has been regarded atypical in that there are a large number of microglomeruli (~800 smaller glomeruli) in the antennal lobe, most projection neurons are multiglomerular, and local neurons fire only calcium spikelets and not full-blown sodium spikes. The patterns of spiking responses evoked by the odorants are complex and patterned over both neurons and time^{10-12,14}. How information regarding behavioral outcomes is encoded by these spatiotemporally patterned neural responses is the problem we sought to examine in this study. We hope that the reviewer would agree that this is a novel problem that has not been examined in other models of olfaction as well.

First, we found that individual neurons in the locust antennal lobe have diverse odor ‘tuning,’ and vary in how neural activity relates to the overall behavioral outcomes (replotted- Rebuttal Figure 3 below). Further, the combination of neurons activated during the odor presentation (i.e. ON response) is completely different from the combination of neurons activated after its termination (i.e. OFF response). Even though the spatiotemporal responses

evoked by all odors were distinct from one another to allow recognition, the response patterns evoked by ‘appetitive odors’ during ON and OFF periods were still confined to a small subset of all possible combination of neural activities. Therefore, these ensemble response patterns were on or near a low-dimensional ‘neural manifold.’ Similarly, the response patterns evoked by ‘non-appetitive’ odors were confined to a different subset of odor-evoked activity patterns and therefore resided on or near a different neural manifold (**Manuscript Fig. 7**).

Rebuttal Figure 10: Distributions showing the correlations of PN responses with innate behavioral preferences for all 22 odors. Distributions are identical to those reported in **Manuscript Fig. 2e**, but are overlaid for comparison purposes. (p -value = 0.0466, t-test).

Our results indicate that both ON and OFF neural responses could be used to predict the overall behavioral outcomes (**Manuscript Fig. 4**). As can be expected, which combination of neurons contributed to predicting the overall appetitiveness of the odorant differed significantly between the ON and the OFF neuron-behavior mapping models (**Manuscript Fig. 4e**). Stated differently, stimulus-evoked activity in two different subgroups of neurons in the antennal lobe correlate with and therefore are predictive of the same behavioral outcome. We had highlighted this novel result in our manuscript but this important point appears to have been completely missed by the reviewer.

Furthermore, the conditioning results we presented reveal several novel results that have not been previously reported. First, not all odors could be successfully reinforced with a gustatory reward. Odorants that evoked responses in one neural manifold (hexanol and isoamyl acetate) were good candidates for CS-US pairing, while the odorants that evoked responses in a different neural manifold (benzaldehyde and citral) were not. Overlapping the reward presentation with either the ON neural responses or the OFF neural responses (of hexanol and isoamyl acetate) led to effective learning and increased POR to the conditioning stimulus (**Manuscript Fig. 5**).

Intriguingly, our results also indicate a temporal window when no learning happened (**Rebuttal Fig. 11**). When the reward presentation was delayed to occur 4 s after the termination of the odorant, no learning occurred for any odorant. Most results on trace conditioning have reported successful conditioning even when the reward is presented tens of seconds after stimulus termination. However, our results indicate a time window when CS-US pairing cannot happen irrespective of what odorant was used for training! We have discussed and highlighted this novel result as well in the revised discussion section.

Rebuttal Figure 11: OFF-conditioning locusts with 4 s delay produces no learned responses. [Rebuttal Fig. 1 is replotted here for convenience]

We attempted to condition locusts to the offset of hexanol and benzaldehyde with a 4 second gap between odor presentation and food reward during the training phase (similar approach as results in **Manuscript Fig. 5, 6**). Trained locusts were then tested for PORs to hexanol, iaa, benzaldehyde, and citral. The mean PORs for each test condition are shown. As can be seen, locusts produced negligible learned behavioral responses to all odors during the test phase for both conditioning paradigms **Rebuttal Fig. 11**.

Therefore, our results indicate that both ON and OFF responses, though distinct for each odorant on our panel, could be used to predict the overall behavioral outcomes. Note that ON and OFF responses evoked by an odorant were the most distinct responses that were generated. In other words, the angular similarity between ON responses of an appetitive and a non-appetitive odorant was greater than that of the ON and OFF responses of the same odorant! Nevertheless, the odor-evoked spatiotemporal response patterns were not randomly scattered in the state-space. They were organized such that both ON and OFF responses of odorants with similar overall behavioral preferences resided on or near a neural manifold. Odorants that evoked diverging behavioral responses evoked neural responses with response variance being captured in distinct neural manifolds. This is the spatiotemporal coding logic for mapping neural response onto a 1-d behavioral response that we were referring to. This is now clarified in the manuscript.

In sum, our results incorporate the temporal dimension to the problem of valence coding. Further, we develop a single framework for linking neural responses with innate and acquired odor preferences. The discussion section has been revised to clarify the novelty and importance of our findings. We hope that the reviewer will now appreciate the contributions made in the revised manuscript.

Reviewer #3 (Remarks to the Author):

This manuscript by Chandak and Raman uses the locust olfactory system to examine the relationship between olfactory projection neuron (PN) responses and an appetitive behavior (palp opening). The authors measure palp opening in response to a panel of odorants and compare these to neural responses to the same stimuli. They find that neural responses to odorants that evoke strong palp opening and those that suppress palp opening are clustered, such that palp opening can be predicted as a linear function of PN responses. They further explore plasticity of the palp opening behavior by pairing odors that either evoke or suppress palp opening with a food reward at different latencies relative to the stimulus. Surprisingly they found that pairing food with either type of odorant increased palp opening but only in response to odors that innately drive this behavior. The authors interpret their results through the lens of neural manifolds.

Overall this is an intriguing dataset that convincingly links specific patterns of PN activity to palp opening behavior. However, I have concerns about the design and interpretation of the behavioral experiments as well as the overall interpretation of the results.

We thank the reviewer for the excellent summary of our contributions. We have addressed the remaining concerns below and in the revised manuscript. We hope that the reviewer will find the revised article suitable for publication.

Major concerns:

It appears that the palp opening behavior is analog and dynamic and can be measured quantitatively as shown in Figure 5, however the behavioral measurements shown in Figure 1 are binarized and only one measurement is made per odor per animal (!). I do not understand why the behavior cannot be measured repeatedly in the same animal (as the authors argue that it does not adapt) with the same precision shown in figure 5 to provide measures of behavioral probability, amplitude, and dynamics. As the authors wish to argue that odor OFF responses are also related to the behavior, the latency and dynamics of the behavior across odorants and animals appears to be key. (i.e. responses to odor OFF cannot be causing the behavior if they occur after the behavior and any correlation between these two would have to reflect some other correlation between OFF responses and prior neural activity).

We performed an additional experiment where we recorded the innate palp-opening responses (PORs) of untrained locusts for six repeated presentations of hexanol (1% v/v). The results of this experiment are shown in **Rebuttal Fig. 2, 3**. We were worried whether assaying POR responses to same odorant several times and to different odorants will result in potential confounds due to muscle fatigue (for example, just 3 repetitions of each odor would lengthen the behavioral experiments by over one hour for each locust). Hence, we restricted our dataset to a single presentation of each odorant in our panel. As we noted, the behavioral POR preferences for odorants were stable and converged when we randomly sampled from subsets of 18 locusts or more. Complementing this result, **Rebuttal Fig. 2** reveals that POR responses to a single odorant remain stable when the same odorant is repeatedly presented. These results should nicely corroborate the main points made in the manuscript.

We binarized the behavioral data for this set of experiments as the only point that we were trying to make is that some odorants innately evoked more POR responses than others,

and we wanted to compute this preference index for each of the twenty-two odorants used in the study.

As the reviewer suggested, we computed the latency of innate behavioral responses for the new set of untrained locusts (same dataset as **Rebuttal Fig. 2**) by painting the palps similar to conditioning experiments (**Manuscript Fig. 5**) to allow accurate quantification of response latencies (**Rebuttal Fig. 12** replotted below). The response latency was then measured as the time taken after odor onset for the palp separation to exceed 1 standard deviation above the average baseline palp separation. As can be seen, locusts initiate behaviors within 0.6 – 1 s for all presentations of hexanol – well below the 4 s when odor is terminated (i.e., well before the odor OFF period has begun).

The reviewer raises an important concern about behaviors initiating/occurring in the odor OFF period. However, we found that even when locusts were conditioned for the OFF period of hexanol, the PORs during the test phase initiated within the odor ON period (refer normalized POR traces in **Manuscript Fig. 5**), reaching over 50% of their peak responses within 2.5 s (odor pulse of 4 s, **Supplementary Fig. 4**).

At the neural level in the antennal lobe, the high-dimensional response vectors capturing the ON and OFF periods in our data were also found to be almost orthogonal for all odors (**Rebuttal Fig. 17** below). This orthogonality resulted in a strong relationship between odor OFF responses and innate behaviors. The exact functional role of these OFF responses is not yet fully known; however, previous works suggest that the OFF responses may be more relevant to the termination of behaviors rather than the initiation¹⁵.

a Repeated hexanol presentations to untrained locusts

b Response latency of untrained locusts

Rebuttal Figure 12: Palp-opening response dynamics in untrained locusts. [Rebuttal Fig. 3 replotted here for convenience]

a) The mean palp-opening responses for 20 untrained locusts, presented with six trials of hexanol. Error bars indicate s.e.m.

b) Response latency of palp-opening responses. In order to accurately quantify response latency, we tracked the palp-opening responses of locusts for the results shown in **Rebuttal Fig. 1**. We used a similar approach as reported in the conditioning experiments (**Manuscript Figs. 5-6**). The time taken for the palp separation to exceed 1 s.d. of baseline (no odor) separation was used to compute the response latency. The mean latency across all locusts ($n = 20$) for each of the six trials are shown here (error bars indicate s.e.m.). As can be seen, the locusts initiate PORs within 0.6 s to 1 s of odor onset for all trials.

A related point is that the quantification of learning shown in Figures 5 and 6 would be more convincing if the authors showed a direct comparison of mean behavioral dynamics in trained and untrained locusts. Here they are relying on the fact that they chose locusts that did not show an initial reaction to the appetitive odorants, however it seems difficult to completely rule out some kind of facilitation or arousal mechanism to account for the “learning” shown here.

This is an important point that is raised by this reviewer.

For all our conditioning experiments, we pre-screened locusts for all 4 odorants – hexanol, iaa, benzaldehyde, and citral. Only locusts that did not produce innate PORs to ANY of the 4 odorants (not just the appetitive odorants as the reviewer mentioned above) were used for the experiments. Hence, prior to conditioning, the locusts displayed similar/equal innate preferences to these four odorants (all 4 odors evoked no innate PORs).

In order to rule out the possibility of facilitation driving our observed behavioral results, we would like to point out two important controls within our dataset. First, we would like to discuss the results from our attempts to condition locusts for the OFF period. For this set of experiments, we presented the food reward 4 s gap after the odor termination. We used both hexanol (appetitive) and benzaldehyde (non-appetitive) as conditioned odors for these set of experiments. Note that two independent sets of locusts were used, one for hex-OFF (4s latency) training and the other for bzald-OFF (4s latency) training . If locusts were simply responding to appetitive odors through some kind of “facilitation or arousal mechanism”, then we would expect to see a large fraction of them produce PORs during the unrewarded test phase to the trained odorant. However, as can be seen in **Rebuttal Fig. 13** below, locusts trained by pairing the conditioned stimulus with a reward 4 s after its termination failed to produce any behavioral responses to the trained odorant or any of the other three tested odorants. This result clearly indicates that locusts that did not have any POR to the odorants tested before training did not have POR responses to the trained odorant even though it was repeatedly encountered. This control result should convincingly rule out the main concern raised by this reviewer.

Secondly, as we explained, we also used benzaldehyde and citral as conditioning stimulus in some of our experiments. If our conditioning experiments were simply facilitating behavioral responses, then we would expect these locusts to also produce PORs to benzaldehyde or citral because these odorants were repeatedly encountered during the training phase. However, as shown in **Rebuttal Fig. 14**, locusts produce little to no learned responses to these odorants. Hence, while the untrained locusts were selected for conditioning experiments to have similar initial preferences for all 4 odors (i.e., no response to any odor), the conditioning protocols produced learned responses only in the hexanol and isoamyl acetate – thereby selectively altering preferences.

Taken together, these two results convinced us that the behavioral responses we observed were not simply an artifact of facilitation or arousal mechanism, but an active result produced due to the odor – food reward overlap presented during the training phase.

Rebuttal Figure 13: OFF-conditioning locusts with 4 s delay produces no learned responses. [Rebuttal Fig. 1 replotted here for convenience] We attempted to condition locusts to the offset of hexanol and benzaldehyde with a 4 second gap between odor presentation and food reward during the training phase (similar approach as results in **Manuscript Fig. 5, 6**). Trained locusts were then tested for PORs to hexanol, iaa, benzaldehyde, and citral. The mean PORs for each test condition are shown. As can be seen, locusts produced negligible learned behavioral responses to all odors during the test phase for both conditioning paradigms. The fractions of locusts with significant responses are indicated in **Rebuttal Figure 14**.

Rebuttal Figure 14: Quantifying locust learned responses across all training paradigms.

a) Heatmap showing the fraction of locusts that produced significant PORs to the test odorant (x-axis) for ON-training with four different odorants (y-axis).

b) Similar plot as **panel a** but for locusts trained with hexanol using ON- and OFF-training (0.5 s, 2 s, 4 s gaps) paradigms.

c) Similar plot as **panel a** but for locusts trained with benzaldehyde using ON- and OFF-training (0.5 s, 2 s, 4 s gaps) paradigms.

The authors argue that pairing reward with odor offset results in different behavioral dynamics as they show in Figure 5d. However in Figure 6, it appears that IAA ON training also results in hexanol responses with a delayed peak, and benzaldehyde OFF training results in IAA responses with normal latency. These variations make it challenging to conclusively link neural dynamics to the dynamics of the behavior.

The reviewer is correct in observing the nuanced differences produced by different learning paradigms. In order to better understand these results, we looked at how the neural similarity

between the trained and tested odorant evolved as a function of time (**Rebuttal Fig. 15**, black curves). Next, we compared these results to the mean behavioral responses (red curves)

produced by the test odorant for locusts conditioned with the trained odorant. Finally, we computed the similarity between these two curves using a correlation metric (R-values shown on each plot) and found them to be significant in each case (p-values on each plot). This analysis should reveal how the behavioral response dynamics (temporal and magnitude) produced during the test phase is tightly linked to the neural response evoked by the test odorant (i.e., the similarity of the neural response pattern evoked by the test odorant with the response evoked by the training stimulus).

Rebuttal Figure 15: Mapping learned behavioral responses to PN responses.

a) We computed the relationship between the mean behavioral responses for locusts trained on Hexanol ON and tested for Hexanol (red curve, right axis) and the PN response correlation (black curve, left axis) between the trained condition (Hexanol ON) and test condition (Hexanol ON and OFF). As can be seen, the two curves have a significant correlation ($R^2 = 0.78$, p -value = $8.4 \cdot 10^{-18}$), indicating a strong pattern match between neural response temporal dynamics and observed behavioral results.

- b) Similar plot as in **panel a**, but for locusts trained for IAA ON and tested on Hexanol.
- c) Similar plot as in **panel a**, but for locusts trained for Benzaldehyde OFF and tested on IAA.
- d) Similar plot as in **panel a**, but for locusts trained for Benzaldehyde ON and tested on IAA.

Overall it does not seem that the pairing done here produces a learned odor-reward association, rather the food pairing seems to facilitate an innate behavioral response to a fixed set of odorants. Although the authors note this in their Discussion, it seems that the most parsimonious mechanism for such plasticity would be modulation of connections downstream of the antennal lobe onto motor or premotor centers. In this view, a relatively fixed pattern of ORN activities is linked to the POR, but the strength of this connection is variable. Repeated food exposure could potentiate this connection to elicit stronger behavior in response to the same innate odorants, regardless of which odor was paired with food. The one piece of data arguing against this interpretation is the delayed OFF pairing with hexanol, but this seems like a lot of weight to place on this result in the context of all the other findings.

We disagree with the reviewer's interpretation that the pairing performed here produced no learned odor-reward associations as we discussed above (**Rebuttal Figs. 13-15**). To reiterate,

- PORs to any of the test odorants did not increase/change when the reward was presented 4 s after the termination of the training odorant
- Conditioning with benzaldehyde or citral with any latency (during ON or OFF training paradigms) did not result in an increased POR to these training odorants

Finally the authors argue in Figure 7 that only odors that evoke activity in the "learning manifold" can be associated with food reward however this seems to be in conflict with their statement in Fig. 6 that locusts trained with citral and benzaldehyde showed palp opening to hexanol and isoamyl acetate. Again, the simplest interpretation is that no odor-food association is formed but rather that certain set of PN activities can evoke palp opening and this response can be potentiated by repeated food reward or potentially by food-odor pairing. A good control here might be to expose a set of locusts to repeated food reward with no odor and observe any change in the probability of response.

We would once again disagree with the reviewer's suggestion about there being a lack of any food reward-odor pairing in our observed behavioral results (**Rebuttal Figs. 13-15**). Locusts trained with citral and benzaldehyde did not generate any PORs during the testing phase. This is what we are referring to. The cross-learning/generalization that we observed between neural responses and behavioral PORs could once again be understood if we examine how the neural similarity between the trained and tested odorant evolved as a function of time (**Rebuttal Fig. 15**)

As we discussed above, presenting the food reward 4s after termination of the training odorant is a control experiment very similar to what the reviewer is proposing. The palp-opening response probability did not increase in this control experiment (**Rebuttal Fig. 16**) .

Rebuttal Figure 16: OFF-conditioning locusts with 4 s delay produces no learned responses. [Rebuttal Fig. 1 replotted here for convenience] In the top row, locusts received food 4s after termination of hexanol. This was repeated six times during the training phase. But no PORs were observed for hexanol or any of the other test odorants. In the bottom row, locusts received food reward 4s after termination of benzaldehyde (6 trials). This is very similar to the control experiment proposed by this reviewer.

Given these concerns I am not convinced by the authors' statements about a neural manifold for learning although I think they show convincingly that the POR is driven by a specific pattern of PN activity that can be described either using a regression model or by a manifold.

Given the clarifications and additional experiments added to convincingly rule out the facilitation hypothesis proposed, we hope the reviewer will now support publication of our manuscript.

Minor concerns:

— its a small point but I am not sure palp opening is equal to having positive valence. Data from several labs suggests that different appetitive behaviors can be differentially driven by different patterns of odor-evoked neural activity (e.g. Jung and Bhandawat 2015).

Not sure what the argument here is. In the context of POR behaviors, our classification of valence is correct – hence we use appetitiveness as the adjective. In other assays it could be measuring valence in a different context such as attraction to mates. This is similar to the proboscis-extension reflex (PERs) in bees, flies, and moths.

— the linear regression model shown in Figure 4 is closely related to the vector valence coding model proposed by Badel and Kazama 2016. Both studies argue that the strength of a specific behavior can be predicted as a linear combination of PN activity where each PN has a relative valence weight either promoting or suppressing the behavior.

The results from our linear regression approach to predict innate valence using PN responses are more nuanced. We have shown/report/observe that the same set of PNs can be assigned completely different sets of weights (refer **Manuscript Fig. 4** last panel) depending on whether the odor/stimulus ON or OFF period is used to predict behavior. This novel result of multiple mappings from the neural space to the behavioral space has now been highlighted in

results/discussion

— I am a bit confused how the OFF activity can predict the behavior if the behavior typically starts while the odor is still going (as shown in Figure 5). Is it possible that OFF activity is correlated with earlier patterns of activity such as suppression below baseline?

The reviewer brings up a very important point here. We find that at the PN level, the ON and OFF responses for an odorant are encoded by very distinct subsets of neurons (i.e., minimal overlap) (**Rebuttal Fig. 17** below). A potential behavioral role for OFF responses has been in the termination of behaviors¹⁵. Thus, while ON-responsive PNs correlated with valence by playing a role in initiating behaviors, the OFF-responsive PNs could be similarly correlated by terminating innate behaviors.

Rebuttal Figure 17: ON and OFF responses to odorants are encoded by distinct subsets of PNs

a) Angle between ON- and OFF- periods are shown for each odor. The mean activity across the respective odor period (and all 10 trials) was computed for each PN to obtain 89-dimensional vectors (89 PNs) for the ON- and OFF- periods. The angle between these vectors was computed for each odor and is shown as a polar scatter plot. Each dot corresponds to a single odorant – the odor index shown along the horizontal axis ranks odors from 1 to 22 based on innate appetitiveness (same color scheme as previous plots). The distance along this dimension is arbitrary. The angles are close to 90° for almost all odors, indicating that the ON- and OFF-vectors are almost orthogonal in this high-dimensional space.

b) Similar analysis as **Manuscript Fig. 4**, but using the odor ON- and OFF-periods for 3 odors – hexanol, isoamyl acetate, and 2-octanol are shown. The blue ON-trajectories evolve in a different direction than the cyan OFF-trajectories. Note that the blue and cyan trajectories have minimal overlap, indicating different subsets of PNs are activated during the ON and OFF periods.

— the authors propose building a model in which behavior is predicted from a relatively sparse set of PN activities but then reject this model out of hand. I think it would be interesting to explore this model quantitatively and see how many PNs are required to match the performance of the whole ensemble in predicting behavior. This might be informative about the types of

activity that encode the behavior.

“but then reject this model out of hand” – We are not sure we rejected it! We have expanded this regression analysis on our revised **Manuscript Figs. 7, 8**.

Rebuttal Figure 18: Effect of regularization on linear model.

We performed linear regression to predict conditioned POR responses under different training paradigms similar to **Manuscript Fig. 6**. In the Manuscript Figure, we set a sparsity constraint of $\alpha = 0.01$. Here in **Rebuttal Fig. 18**, we show results when the alpha value is varied for different values between [0-1] for each of the six training paradigms. In each plot, the x-axis denotes the number of PNs that were assigned a non-zero weight using this approach. The mean squared error of the predicted values for each regularized case is plotted along the y-axis (normalized by the maximum value for each subplot to range between [0,1]). As can be noted, increasing alpha reduced the number of non-zero weighted PNs and increased the error of the learned model.

Rebuttal Figure 19: Comparing different levels of regularization in linear model.

The observed behavioral results (black curves) and predictions from different linear models with varying alpha-constraints (red curves) are shown for 8 values of alpha for hexanol-ON trained locusts, tested on all four odors (similar to **Manuscript Fig. 7**). As can be seen, the predictions get worse as the model regularization is increased. We are presenting these results for hexanol-ON training, but very similar results were obtained for other training conditions and were hence omitted for redundancy.

— Figure 6b axis labels are extremely small and hard to read

The original Figure 6 has now been split into two separate figures.

Once again, we thank the reviewer for highly useful feedback that helped us improve our manuscript.

References

- 1 Min, S. H., Ai, M. R., Shin, S. A. & Suh, G. S. B. Dedicated olfactory neurons mediating attraction behavior to ammonia and amines in *Drosophila*. *P Natl Acad Sci USA* **110**, 1321-1329, doi:DOI 10.1073/pnas.1215680110 (2013).
- 2 Roussel, E., Carcaud, J., Combe, M., Giurfa, M. & Sandoz, J.-C. Olfactory Coding in the Honeybee Lateral Horn. *Current Biology* **24**, 561-567 (2014).
- 3 Semmelhack, J. L. & Wang, J. W. Select *Drosophila* glomeruli mediate innate olfactory attraction and aversion. *Nature* **459**, 218-223, doi:Doi 10.1038/Nature07983 (2009).
- 4 Stensmyr, M. C. *et al.* A Conserved Dedicated Olfactory Circuit for Detecting Harmful Microbes in *Drosophila*. *Cell* **151**, 1345-1357, doi:DOI 10.1016/j.cell.2012.09.046 (2012).
- 5 Strutz, A. *et al.* Decoding odor quality and intensity in the *Drosophila* brain. *eLife* **3**, e04147 (2014).
- 6 Suh, G. S. B. *et al.* A single population of olfactory sensory neurons mediates an innate avoidance behaviour in *Drosophila*. *Nature* **431**, 854-859, doi:Doi 10.1038/Nature02980 (2004).
- 7 Hansson, B. S. & Anton, S. Function and Morphology of the antennal lobe: New Developments. *Annual Review of Entomology* **45**, 203–231 (2000).
- 8 Anton, S. & Homberg, U. in *Insect Olfaction* (ed B. S. Hansson) Ch. 4, (Springer, Heidelberg, 1999).
- 9 Saha, D. *et al.* A spatiotemporal coding mechanism for background-invariant odor recognition. *Nat Neurosci* **16**, 1830-1839, doi:10.1038/nn.3570 <http://www.nature.com/neuro/journal/v16/n12/abs/nn.3570.html#supplementary-information> (2013).
- 10 Nizampatnam, S., Zhang, L., Chandak, R., Li, J. & Raman, B. Invariant odor recognition with ON– OFF neural ensembles. *P Natl Acad Sci USA* **119**, e2023340118 (2022).
- 11 Nizampatnam, S., Chandak, R., Saha, D. & Raman, B. Dynamic contrast enhancement and flexible odor codes. *BioRxiv*, doi:<https://doi.org/10.1101/188011> (2017).
- 12 Brown, S. L., Joseph, J. & Stopfer, M. Encoding a temporally structured stimulus with a temporally structured neural representation. *Nat Neurosci* **8**, 1568-1576, doi:http://www.nature.com/neuro/journal/v8/n11/supinfo/nn1559_S1.html (2005).
- 13 Broome, B. M., Jayaraman, V. & Laurent, G. Encoding and Decoding of Overlapping Odor Sequences. *Neuron* **51**, 467-482, doi:10.1016/j.neuron.2006.07.018.
- 14 Laurent, G., Wehr, M. & Davidowitz, H. Temporal representations of odors in an olfactory network. *J Neurosci* **16**, 3837-3847 (1996).
- 15 Saha, D. *et al.* Engaging and disengaging recurrent inhibition coincides with sensing and unsensing of a sensory stimulus. *Nature Communications* **8**, 15413 (2017).

REVIEWER COMMENTS

Reviewer #1 (Remarks to the Author):

Apologies for my belated response on this paper; I've been sick over the last few days. I want to thank the authors for their responses, which I think address most of my comments. Below follow a few comments related to my previous revision and the revised version of the manuscript, respectively.

FOLLOW UP ON MY (REVIEWER #1) COMMENTS FROM REVISION 1:

Comment 2) I apologise for my confusing comment. If PN responses do not reflect motor output —I'm not an expert on this field—, then my comment was indeed scientifically inappropriate.

Comment 9) I enjoyed reading your response, and I would encourage you to include some of those reflections in the Discussion, e.g., in the last section.

COMMENTS ON THE REVISED VERSION

1) Abstract: (1) In the sentence “notably, only odorant's that evoked neural response excursions in the appetitive manifold were conducive for learning” -> indicate type of learning paradigm; (2) include a description of the relationship between appetitive and non-appetitive odorants.

2) I found the last two paragraphs of the Introduction harder to parse than most of the paper. The last paragraph in particular, is a bit dense, and I believe the term state space —that may not be familiar to many biologists or neuroscientists— should be defined; the same for neural manifold. Of course this is a recommendation; it's not my paper!

3) Figure 4e(right): Indeed, comparing model weights is interesting but since this a many (neurons) to one mapping (POR), there could potentially be other models that perform (almost) equally well while having very different weights. I'd suggest the authors include a comparison of cross-condition performance too as they did in the rebuttal letter.

4) Rebuttal Figure 6 is very interesting. I'd include it, along with some summary across all conditions, in the paper.

5) I really like the new Figure 9, but I have a small comment about panels c and d: In addition to showing the average angle using a bar, the authors show the angle for each comparison (e.g., and single markers), and the s.d./s.e.m. of the distribution. Also, if this is the arc cosine of the dot, don't +alpha and -alpha have the same cosine? Or am I missing something?

6) Looking at Figure 9a I wondered whether the organisation of the "neural trajectories" (in PCA space) elicited by appetitive odours similarly arranged (that is, homologous or congruent) between On and Off epochs. For example, there seems to be some responses with larger radii than others for both the On and Off epochs, are these triggered by the same odourants? (Perhaps this can be observed in the dendrogram but I'm struggling to).

7) In the figures, include all datapoints in addition to the error bars, e.g., in Figure 5 (this is not only my personal preference, but a requirement from Nature Comms). Also, please increase the font size for all the axes labels and headings, they are extremely tiny.

8) In future versions, please include the figures within the text.

Reviewer #2 (Remarks to the Author):

I have read the responses of the authors to my comments, and the comments of other reviewers. I still don't see how the results presented in this manuscript goes beyond a descriptive analysis of multi neuronal odor responses in locust antennal lobe and the correlation of these responses with behaviors to a select set of odors. The authors results provide no mechanistic insights of what is special about ON or OFF responses, what underlies the observed phenomenon.

I do not see the major conceptual and mechanistic advancement that is provided by this study in our understanding of olfactory system computations.

I have no further comments

Reviewer #3 (Remarks to the Author):

In this revised manuscript the authors have modified how they discuss the role of OFF responses in driving the behavior. I think the data here are solid: the authors show convincingly that odors differ in their innate ability to drive palp opening and this is correlated with patterns of PN activation. This PN representation also seems to constrain learning. Specifically, only odors within a subspace of the PN coding range can drive palp opening, although a wider set of odorants can promote increased responses to the appetitive odorants. My main quibble is with the characterization of this learning as “associative”. I am convinced that a pairing of odor (within some time window) with reward is necessary to increase the behavior, but the fact that pairing with aversive odors drives increased responses to the appetitive odors suggests that this learning is not “associative” at least as I have understood that term. Rather it seems that a wide range of odors can facilitate the behavior while only a narrower subset can actually drive it. This has interesting circuit implications that I think they could spell out in more depth in the Discussion, where I think a number of points are currently made that don’t fit the data shown. In addition, there are a number of papers on related topics (encoding of odor valence) that I think should be cited and discussed.

Major comments:

1) I don’t buy the argument that the learning is “associative” because benzaldehyde pairing does not lead to benzaldehyde responses. Benzaldehyde pairing drives responses to hexanol and IAA (Fig. 6b-d) even though it is on the “non-learning” manifold. I think the authors need a new name for this kind of learning. How does learning with this kind of profile fit with existing models of the mushroom body and known learning mechanisms in the insect brain? I think this should be discussed.

2) Several papers on related topics should be discussed and cited:

Wu...Su et al. (PNAS, 2022) showed recently that peripheral odor representations in *Drosophila* encode behavior and behavioral valence more reliably than odor identity (relevant to 1st paragraph of p.9)

Badel...Kazama (Neuron, 2016) examined innate behavioral valence and link this to PN population responses in the antennal lobe (relevant to Discussion p. 10).

Chong...Rinberg (Science, 2020) examined how spatio-temporal patterns of activity in the olfactory bulb are linked to odor recognition using optogenetic manipulation of OB activity (relevant to Discussion p. 11)

Specific comments:

Introduction: "how information regarding the valence of a chemical cue is encoded in the olfactory system". This should probably have some citations as several studies (including some listed above) have examined this question before.

p. 7: "learning manifold" I don't think the data quite support this, as odors not on the "learning manifold" can drive an increase in responses to other odors (e.g. benzaldehyde). So odor is different parts of the coding space must have different effects on either evoking the behavior or causing a facilitation of the behavior in response to other odorants.

p. 9: "behavioral recognition of odorants". I'm not sure how much evidence is presented here for tight "recognition" of particular odorants. It seems that the locusts will respond to a wide range of appetitive odorants and that the facilitative pairing generalizes widely to other appetitive odorants.

p. 10: "associate learning". As noted above I don't think what is being described here is well-fit by the term "associative learning". It is interesting but doesn't fit traditional models where odor identity is first computed then specific identities are linked to a behavioral outcome through plasticity.

p.10: "Therefore, if initially broad responses to appetitive and non-appetitive odorants were refined as a result of learning...". Again, I don't see any evidence here for refinement of odor identity coding, since the behavioral responses seem to generalize quickly and readily to anything appetitive.

p. 10: "no associative learning occurred while delivering rewards during responses excursion in the "non-appetitive manifold". This is not true. Benzaldehyde pairing did not lead to benzaldehyde responses but it did cause responses to the appetitive odors, so pairing with odors in the "non-appetitive manifold" can lead to learning, it just needs a different name.

p. 11: "There is no concept of time or dynamics in this current interpretation." I don't think this is fair to the existing literature. Lots of work from Rinberg and colleagues on temporal patterns and identity coding in the olfactory system, as well as older work from the Laurent group etc.

Figure 5: Is there a way to do the normalization so you don't get spurious "responses" as in c right panel? This seems solveable.

Dear Editor,

We would like to take the opportunity to thank the three reviewers again who have helped us tremendously improve the manuscript. In this revised version, we have included a computational model to (i) provide mechanistic insights (to address reviewer 2 comments), and (ii) explain the paradoxical observation regarding our conditioning experiments (to address reviewer 3 comments). Below, we have summarized how we have addressed each and every concern raised by the three reviewers. With these additional revisions we believe that we have addressed all remaining issues raised by the three reviewers. We hope that you and the reviewers find the revised manuscript worthy of publication in Nature Communications.

Sincerely,
Barani Raman
(on behalf of all authors)

REVIEWER COMMENTS

Reviewer #1 (Remarks to the Author):

Apologies for my belated response on this paper; I've been sick over the last few days. I want to thank the authors for their responses, which I think address most of my comments. Below follow a few comments related to my previous revision and the revised version of the manuscript, respectively.

FOLLOW UP ON MY (REVIEWER #1) COMMENTS FROM REVISION 1:

Comment 2) I apologise for my confusing comment. If PN responses do not reflect motor output—I'm not an expert on this field—, then my comment was indeed scientifically inappropriate.

Comment 9) I enjoyed reading your response, and I would encourage you to include some of those reflections in the Discussion, e.g., in the last section.

We would like to thank Reviewer #1 again for their suggestions and comments on the manuscript. As recommended, we have included portions from our previous rebuttal responses into our main manuscript as part of the results and discussions sections.

COMMENTS ON THE REVISED VERSION

1) Abstract: (1) In the sentence “notably, only odorant's that evoked neural response excursions in the appetitive manifold were conducive for learning” -> indicate type of learning paradigm; (2) include a description of the relationship between appetitive and non-appetitive odorants.

We made the following changes to the abstract:

“Notably, only odorants that evoked neural response excursions in the appetitive odor manifold could be associated with gustatory reward.” To address point 1.

“innately appetitive odorants (**higher palp-opening responses**)” to clarify the second point.

2) I found the last two paragraphs of the Introduction harder to parse than most of the paper. The last paragraph in particular, is a bit dense, and I believe the term state space—that may not be familiar to many biologists or neuroscientists— should be defined; the same for neural manifold. Of course this is a recommendation; it's not my paper!

We re-read those paragraphs and agree with the reviewer. We have re-written these paragraphs to be simple and avoid unnecessary jargons and abstract sentences.

3) Figure 4e(right): Indeed, comparing model weights is interesting but since this a many (neurons) to one mapping (POR), there could potentially be other models that perform (almost) equally well while having very different weights. I'd suggest the authors include a comparison of cross-condition performance too as they did in the rebuttal letter.

The analysis that the reviewer is referring to has been added to our manuscript as **Supplementary Figure 2b, c**. We agree, this is an interesting and effective control that enhances our main modeling results (**Figure 4**).

4) Rebuttal Figure 6 is very interesting. I'd include it, along with some summary across all conditions, in the paper.

We have incorporated Rebuttal Figure 6 from the previous rebuttal as part of our Discussion section (**Supplementary Figure 8**). This figure reveals how the behavioral response dynamics (temporal and magnitude) produced during the test phase is tightly linked to the neural response evoked by the test odorant (i.e., the similarity between the neural response pattern evoked by the test odorant with the

response evoked by the training stimulus). It supports/complements the other results we have presented in **Figures 7 and 10**, which show how information is spatiotemporally distributed across PNs can predict learned POR responses and their dynamics.

5) I really like the new Figure 9, but I have a small comment about panels c and d: In addition to showing the average angle using a bar, the authors show the angle for each comparison (e.g., and single markers), and the s.d./s.e.m. of the distribution. Also, if this is the arc cosine of the dot, don't +alpha and -alpha have the same cosine? Or am I missing something?

We believe there might be some confusion regarding the data plotted in these panels and would like to apologize for the same.

For a particular odor 'A', we computed the angle between odor A and all other 21 odorants in the high-dimensional PN-space (angular distance between two 89-dimensional response vectors corresponding to neural responses of 89 PNs; averaged over time and trials). The reviewer correctly notes that this angle cannot be negative. This pairwise angle was computed for all odor pairs for both ON and OFF periods, and angular distance matrix is now included in **Supplementary Figure 5b, c** (22 x 22 matrices for ON and OFF period pairwise angles; see below). For visualization in the main figure (**Fig. 9c, d**), we collapsed the 21 ON and OFF angles obtained for each odor using the following two steps. First, for each odor, we computed the average angle between the odorant and appetitive odors, and the average angle between the odorant and non-appetitive odors (ensuring to omit any self-comparison of the odor along the main diagonal). Second, we took the difference between these two angles such that a positive angle would denote higher similarity with appetitive odorants, and a net negative angle would imply similarity with non-appetitive odorants.

In **Figure 9c, d** we have plotted 22 bars, one for each odor shown along the x-axis. The color of each bar indicates the innate POR probability for the odor (color scale shown on the right). The y-axis denotes the difference between two angles (average similarity with appetitive – average similarity with the unappetitive odorants) as described above. While each angle obtained in **Supplementary Figure 5b, c** is non-negative, the bars shown in **Figure 9c, d** can indeed be negative, as they are denoting the difference between two angles.

6) Looking at Figure 9a I wondered whether the organisation of the “neural trajectories” (in PCA space) elicited by appetitive odours similarly arranged (that is, homologous or congruent) between On and Off epochs. For example, there seems to be some responses with larger radii than others for both the On and Off epochs, are these triggered by the same odourants? (Perhaps this can be observed in the dendrogram but I'm struggling to).

The reviewer has raised an interesting question here, one that we did not think about. To answer this, we analyzed the odorant responses shown in **Fig. 9a**. For each odorant in the 3-D PCA space, we have 80 data points in the ON-period, and 80 data points in the OFF-period (corresponding to 80 time bins). We took the means of these sets of 80 points to obtain a mean ON and OFF response vectors. The length or norm of these vectors indicated the overall strength of these ON and OFF responses. In **Supplementary Figure 5a**, we show a scatter plot corresponding to these vector lengths, with the mean ON-period response strength shown along the x-axis, and the mean OFF-period response strength shown along the y-axis. The red best-fit line indicated has an R-squared value = 0.42, indicating a positive relationship between the two sets of data. Thus, the reviewer is correct in hypothesizing that the stronger responses (with larger radii) in ON and OFF epochs typically correspond to the same odor.

New Supplementary Figure 5: a) For the PCA plots shown in **Fig. 9a**, we computed the mean ON and OFF response strength (vector norm) in PCA-space for all 22 odors. Here, we have plotted the mean ON response strength for each odorant along the x-axis, and OFF-response strength along the y-axis

7) In the figures, include all datapoints in addition to the error bars, e.g., in Figure 5 (this is not only my personal preference, but a requirement from Nature Comms). Also, please increase the font size for all the axes labels and headings, they are extremely tiny.

We have re-organized all our main text Figures to be legible, and have also increased the font sizes across all sub-panels. Figures 1, 2, 5, 6 summarizes all the data we collected for this manuscript. In addition, we have also included Supplementary Figures to break down complex analyses and show individual datapoints when appropriate (e.g. Supplementary Fig. 5b, c).

New Supplementary Figure 5: b) Similarity matrices used for computing the results shown in **Fig. 9c, d**. Pairwise angles between odors in high-dimensional neural space PN ($n = 89$ PNs) during either the ON period (left) or OFF period (right) are shown. A smaller angle indicates two odors have more similar responses (darker color). As can be seen on the left plot, appetitive odors tend to have very similar neural

responses within the group (dark cluster on the top left), and likewise for non-appetitive odors (dark cluster on bottom right). This trend is less pronounced for OFF-responses (right plot).

8) In future versions, please include the figures within the text.

We have re-organized the Main Text document to now include all Figures within the results section close to where they are discussed.

We once again thank this Reviewer for helping us significantly help our manuscript.

Reviewer #2 (Remarks to the Author):

I have read the responses of the authors to my comments, and the comments of other reviewers. I still don't see how the results presented in this manuscript goes beyond a descriptive analysis of multi neuronal odor responses in locust antennal lobe and the correlation of these responses with behaviors to a select set of odors. The authors results provide no mechanistic insights of what is special about ON or OFF responses, what underlies the observed phenomenon.

I do not see the major conceptual and mechanistic advancement that is provided by this study in our understanding of olfactory system computations.

I have no further comments.

We thank the reviewer for taking the time to read the manuscript. In the revised version, we have now developed and included a computational model to reveal how a neural network with Hebbian connections can reproduce our associative conditioning results (new Manuscript Figure. 10 and a new result section that has been added to explain these results; see below). Hence, the manuscript provides a mechanistic insight regarding how the network downstream to the antennal lobe neurons should be organized and connections altered to reproduce the results we observed.

Reviewer #3 (Remarks to the Author):

In this revised manuscript the authors have modified how they discuss the role of OFF responses in driving the behavior. I think the data here are solid: the authors show convincingly that odors differ in their innate ability to drive palp opening and this is correlated with patterns of PN activation. This PN representation also seems to constrain learning. Specifically, only odors within a subspace of the PN coding range can drive palp opening, although a wider set of odorants can promote increased responses to the appetitive odorants.

We thank the reviewer for their comments that helped us significantly strengthen our manuscript.

My main quibble is with the characterization of this learning as “associative”. I am convinced that a pairing of odor (within some time window) with reward is necessary to increase the behavior, but the fact that pairing with aversive odors drives increased responses to the appetitive odors suggests that this learning is not “associative” at least as I have understood that term. Rather it seems that a wide range of odors can facilitate the behavior while only a narrower subset can actually drive it. This has interesting

circuit implications that I think they could spell out in more depth in the Discussion, where I think a number of points are currently made that don't fit the data shown. In addition, there are a number of papers on related topics (encoding of odor valence) that I think should be cited and discussed.

The reviewer makes several important points and raises concerns that we believe we have addressed in this revised manuscript (see below). The point on 'circuit implications' is a remarkable comment, one that inspired us to develop a model to explain our results. We have cleaned up some of the speculative components in the discussion and used the new modeling work to describe the remaining observations that appear paradoxical.

A Hebbian neural network for sensory to behavior mapping

Finally, to gain mechanistic insights regarding how conditioning odorants with reward increased PORs for only some odorants (hex and iaa) but not others (bza and cit), we developed a computational model (**Fig. 10a**). In this model, the input neuron responses (obtained directly from the antennal lobe projection neuron responses we recorded experimentally) feed-forward onto two downstream neurons that had opposing functions: a 'Decoding Neuron 1 (DN1)' that drives appetitive response and a 'Decoding Neuron 2 (DN2)' that inhibits that same response (i.e., an 'anti-neuron'). Both downstream neurons received input from the entire input ensemble. However, weights from one set of input neurons (encoding neural ensemble 1) onto the appetitive Decoding Neuron 1 alone were Hebbian plastic in this model. The rest of network connections remained unaltered after initialization (see **Methods** for details).

Such 'neuron – anti-neuron' pairs have been utilized for predicting overall motor outputs^{31–33}, and are highly consistent with the emerging view from other insect models that have shown mushroom body output neurons form segregated channels to drive opposing behaviors^{34,35}. Finally, the motor output neuron that drives behavior in the model merely takes the difference in overall activity of the 'neuron-anti-neuron pair' (i.e. DN1– DN2) to determine the final behavioral output: successful POR only if appetitive Decoding Neuron 1's activity was stronger than the suppressive Decoding Neuron 2's response. In order to replicate the results from our conditioning experiments, the model would require two criteria to be met: a) reinforceable odorants (hex and iaa) should strongly activate encoding neural ensemble 1 that makes plastic connections with Decoding Neuron 1, b) Non-reinforceable odorants (bza and cit) should evoke strong neural activity in ensemble 2, and at least some weak activity in the encoding neural ensemble 1 (**Fig. 10b**).

Consistent with our hypothesis, we found that the neural responses we recorded for the four odorants used in the conditioning experiments did meet the above expectations regarding how the appetitive and non-appetitive odorants activated the neural ensemble (**Fig. 10c**). As can be noted, some neurons (at the top of PN activity vectors) were activated more by hexanol and isoamyl acetate while responding less to benzaldehyde and citral. On the other hand, only a smaller subset of neurons that were strongly activated by benzaldehyde (near the bottom of PN activity vectors) also responded to hexanol and isoamyl acetate. Therefore, the antennal lobe activity that drives the responses in downstream neurons is consistent with the schematized inputs shown in **Fig. 10b**.

Next, to simulate behavioral conditioning, we updated the weights between the encoding input ensembles and Decoding Neuron 1 alone using a simple Hebbian update rule (see **Methods**). Note that the response threshold for Decoding Neuron 1 was set high to prevent false positives PORs before learning. On the other hand, the response threshold for Decoding Neuron 2 was set low as the overall response strength for non-appetitive odorants were weaker and to allow robust suppression of PORs to all odorants. Our results indicate that irrespective of whether the hexanol or benzaldehyde ensemble responses were used for updating network weights, it always resulted in an increased input to the Decoding Neuron 1 (**Fig. 10d** vs. **e**). Therefore, Decoding Neuron 1 had a transient output after onset for hexanol and isoamyl acetate irrespective of the odor used for reward pairing (**Fig. 10d, e arrowheads**). The stronger response of Decoding Neuron 2, that was only primarily activated by benzaldehyde and citral ensured that there was no POR output to these odorants even after Hebbian modification of network

weights. Thus, this simple neural network with a ‘neuron–anti-neuron pair’ and selective Hebbian connections was sufficient to replicate results from our conditioning experiments.

New Figure 10: A simple neural network with Hebbian plasticity recreates our conditioning experiment results.

a) Schematic of the network used to examine the effects of associative learning on behavioral PORs. The input neurons (corresponding to PNs) are divided into two non-overlapping groups. While all neurons connect to both the downstream decoding neurons, only the connections between encoding ensemble 1 and the decoding neuron 1 are plastic and are altered during associative conditioning using a simple Hebbian rule (see **Methods**).

b) A key model assumption: Appetitive odorants (hex and iaa) that elicited PORs after pairing with food reward should activate the encoding ensemble 1 more. Non-appetitive odorants (bza and cit) that did not elicit PORs after

conditioning are expected to activate PNs in the encoding ensemble 2 more. However, the non-appetitive odors should also activate a few neurons in the encoding ensemble 1.

c) Binary categorization of PN responses in our experimental dataset ($n = 89$ PNs) as responsive or non-responsive to a given odorant. The PNs are ordered such that those activated by hexanol are at the top and those activated by bza are at the bottom. PNs with peak odor-evoked activity greater than mean + 6.5 standard deviations of pre-stimulus activity were considered responsive. Same ordering was used to show PN response category for all four odorants used in our conditioning experiments. Note that the neurons in the top half were activated more by hex and iaa, and many of the neurons in the bottom half responded to bza and citral. Also, both bza and cit activated a few neurons that were responsive to hex and iaa. Hence the neural dataset we collected was consistent with the model assumptions described in **panel b**.

d) Left panel: The activity of the decoding neuron 1 (DN1) and decoding neurons 2 (DN2) in the model along with the expected POR responses generated before any Hebbian alteration of network weights are shown for all four odorants. Note that before learning the DN1 is not activated by any of the odorants and therefore no POR responses were generated. *Right panel:* Similar plots but now showing the DN1 and DN2 along with predicted POR after Hebbian learning. Only the PN activities during *hexanol* presentations were used for altering the network connections.

e) Similar plots as in **panel d** but showing model output

Major comments:

1) I don't buy the argument that the learning is "associative" because benzaldehyde pairing does not lead to benzaldehyde responses. Benzaldehyde pairing drives responses to hexanol and IAA (Fig. 6b-d) even though it is on the "non-learning" manifold. I think the authors need a new name for this kind of learning. How does learning with this kind of profile fit with existing models of the mushroom body and known learning mechanisms in the insect brain? I think this should be discussed.

The reviewer again raises an important point. We fully understand the concern here as this was something that we deeply thought about as well. The concern essentially boils down to, why are behavioral responses to hexanol and IAA increasing after learning with benzaldehyde, especially when there is still no behavioral response to benzaldehyde. Is this still associative learning?

First, we found that there have been reports in other model organisms (particularly honey bees [1]) where associating one odorant with reward generated a stronger response to a different untrained odorant (see Rebuttal Fig. 1). Such asymmetric response generalizations after odor-reward classical conditioning [1, 2] are consistent with our results. However, our result is a bit extreme in that there was no behavioral response to the trained odorant but stimulus-reward conditioning increased response to other odorants. As this reviewer hinted, this indeed has potential implications regarding what aspect of the stimulus-evoked responses are reinforced due to this conditioning.

Rebuttal Figure 1: Generalization of proboscis extension reflex (PER) in honeybees after stimulus-reward. PER to the conditioned odorant (diagonal) and the response to a panel of other odorants (non-diagonal elements) are summarized. Note that training with 1-heptanol resulted in stronger response to 2-octanol, whereas training with 2-octanol resulted only in strong response for that trained odorant and only a weaker response to 1-heptanol. Reproduced as is from [1].

To explain why this is the case, we developed a simple neural network model to transform the PN spiking activities we recorded onto the conditioned behavioral responses we observed (see new Manuscript Figure 10).

Is the learning still “associative”? In the model, reward is still presented while benzaldehyde was presented. Connections from a subset of neurons that responded to be benzaldehyde were strengthened. These neurons were also strongly activated by appetitive odorants. The rest of benzaldehyde responses that suppressed the behavioral response were left unmodified. This was sufficient to generate the behavioral results we observed. As can be noted, selected neural response features of the CS (benzaldehyde) evoked neural response that co-occurred with the reward was emphasized as a result of learning. Hence, this should still technically be associative learning.

2) Several papers on related topics should be discussed and cited:

Wu...Su et al. (PNAS, 2022) showed recently that peripheral odor representations in *Drosophila* encode behavior and behavioral valence more reliably than odor identity (relevant to 1st paragraph of p.9)

Badel...Kazama (Neuron, 2016) examined innate behavioral valence and link this to PN population responses in the antennal lobe (relevant to Discussion p. 10).

Chong...Rinberg (Science, 2020) examined how spatio-temporal patterns of activity in the olfactory bulb are linked to odor recognition using optogenetic manipulation of OB activity (relevant to Discussion p. 11)

All excellent papers. We have now added citations to them in our revised manuscript.

Specific comments:

Introduction: “how information regarding the valence of a chemical cue is encoded in the olfactory system”. This should probably have some citations as several studies (including some listed above) have examined this question before.

Done.

p. 7: “learning manifold” I don’t think the data quite support this, as odors not on the “learning manifold” can drive an increase in responses to other odors (e.g. benzaldehyde). So odor is different parts of the coding space must have different effects on either evoking the behavior or causing a facilitation of the behavior in response to other odorants.

The reviewer is correct. We have now relabeled this as ‘Manifold of odorants that were associated with food reward.’

p. 9: “behavioral recognition of odorants”. I’m not sure how much evidence is presented here for tight “recognition” of particular odorants. It seems that the locusts will respond to a wide range of appetitive odorants and that the facilitative pairing generalizes widely to other appetitive odorants.

It is reasonable to expect that the behavioral response following odor-reward conditioning will be limited to that odorant/stimulus that was used in the training trials. But as we note above (see Rebuttal Figure 1 for similar results in honey bees), there is quite a bit of generalization that has been reported in other model organisms as well [1, 3]. While this generalization is usually explained based on similarity of chemical features (similar functional groups, carbon chain length etc.), our results indicate that neural response similarity during both stimulus presence (ON response) and after its termination (OFF response) can allow better prediction of the behavioral results. Since the behavioral responses are only observed for the trained odorant and other stimuli that evoke similar neural responses, behavioral output (PER or POR or salivation) have been regarded as a valid indicator of behavioral recognition [1 – 4].

Rebuttal Figure 2: The probability of PORs for locusts trained using hex as conditioning stimulus is shown as a bar plot. PORs to the trained odorant (shown in red) and to a diverse odor panel (nontrained odors shown in gray) are shown. Reproduced as is from [4]. Note that locusts trained with hexanol only respond to three out of the twelve odorants used during the test phase.

p. 10: “associate learning”. As noted above I don’t think what is being described here is well-fit by the term “associative learning”. It is interesting but doesn’t fit traditional models where odor identity is first computed then specific identities are linked to a behavioral outcome through plasticity.

See above for clarification.

p.10: “Therefore, if initially broad responses to appetitive and non-appetitive odorants were refined as a result of learning...”. Again, I don’t see any evidence here for refinement of odor identity coding, since the behavioral responses seem to generalize quickly and readily to anything appetitive.

Agreed. This is speculative. We have removed this and discussed results from our modeling work.

p. 10: “no associative learning occurred while delivering rewards during responses excursion in the “non-appetitive manifold”. This is not true. Benzaldehyde pairing did not lead to benzaldehyde responses but it did cause responses to the appetitive odors, so pairing with odors in the “non-appetitive manifold” can lead to learning, it just needs a different name.

This has been revised as follows:

“delivering rewards while the odor-driven neural activities were in the ‘appetitive manifold’ resulted in successful conditioning, whereas delivering rewards during responses excursion in the ‘non-appetitive manifold’ did not result in the non-appetitive odorants being associated with the food reward”

p. 11: “There is no concept of time or dynamics in this current interpretation.” I don’t think this is fair to the existing literature. Lots of work from Rinberg and colleagues on temporal patterns and identity coding in the olfactory system, as well as older work from the Laurent group etc.

The importance of time as a coding dimension is well established in olfaction. Our point only concerns with innate responses. How neural activity patterned over time shapes innate behavioral preferences is not fully understood. That is all our point was. We have clarified this in the discussion section.

“Although the importance of time as a coding dimension has been well established in olfaction^{37,56–60}, the importance of time-varying neural activity for shaping innate behavioral preferences is not fully understood.” [Work from Laurent, Rinberg and others have been cited to clarify our point]

Figure 5: Is there a way to do the normalization so you don't get spurious “responses” as in c right panel? This seems solveable.

We thank the reviewer for this suggestion and have re-plotted the data such that the spurious yellow responses visible in previous versions of this Figure are minimized. Please note that this was merely a visualization modification and no re-analysis of data was performed.

For each locust, we are now normalizing the POR responses to go from 0 to 1, where 1 corresponds to the maximum palp separation produced by that locust across all test odorants. This ensures that for test odors such as citral and benzaldehyde, where locusts produced no significant responses, the normalization was done using stronger responses evoked by appetitive odors (hex and iaa) as the denominator, which resulted in minimal yellow bands appearing post normalization.

References:

[1] Guerrieri F, Schubert M, Sandoz JC, Giurfa M (2005) Perceptual and Neural Olfactory Similarity in Honeybees. *PLOS Biology* 3(4): e60. <https://doi.org/10.1371/journal.pbio.0030060>

[2] Menzel R, Greggers U, Hammer M (1993) Functional organisation of appetitive learning and memory in a generalist pollinator, the honey bee. In: Papaj D, Lewis AC, editors. *Insect learning: Ecological and evolutionary perspectives*. New York: Chapman and Hall. pp. 79–125.

[3] Nick Bos, Patrizia d'Ettorre, Fernando J. Guerrieri; Chemical structure of odorants and perceptual similarity in ants. *J Exp Biol* 1 September 2013; 216 (17): 3314–3320. doi: <https://doi.org/10.1242/jeb.087007>

[4] Nizampatnam S, Zhang L, Chandak R, Li J, Raman B. Invariant odor recognition with ON-OFF neural ensembles. *Proc Natl Acad Sci U S A*. 2022 Jan 11;119(2):e2023340118. doi: 10.1073/pnas.2023340118. PMID: 34996867; PMCID: PMC8764697.

REVIEWERS' COMMENTS

Reviewer #1 (Remarks to the Author):

The authors have addressed all my comments satisfactorily.

Reviewer #3 (Remarks to the Author):

The authors have addressed my concerns. The new circuit model clarifies the hypothesized relationship between AL neural activity and behavior. I commend the authors on a careful and thoughtful study!

Minor points:

p. 15 OFF-conditioning paradigm. Is this similar to the "Trace conditioning" described by Galili and Tanimoto (J neurosci 2011)? Might be interesting to reference.

p. 31, paragraph 2: several typos in this paragraph: "both bee references and "with one exception" not grammatical

Response to the Reviewers:

Reviewer #1 (Remarks to the Author):

The authors have addressed all my comments satisfactorily.

We thank the reviewer for taking the time to evaluate our manuscript and for helping us improve it significantly.

Reviewer #3 (Remarks to the Author):

The authors have addressed my concerns. The new circuit model clarifies the hypothesized relationship between AL neural activity and behavior. I commend the authors on a careful and thoughtful study!

We thank the reviewer for raising several important concerns that helped clarify several key results and for inspiring us to develop the computational model.

Minor points:

p. 15 OFF-conditioning paradigm. Is this similar to the "Trace conditioning" described by Galili and Tanimoto (J neurosci 2011)? Might be interesting to reference.

Yes, this is indeed relevant to our study as the authors have employed a similar conditioning protocol but with longer delays between odor termination and presentation of electric shock. We have added this reference to the manuscript.

p. 31, paragraph 2: several typos in this paragraph: "both bee references and "with one exception" not grammatical

We apologize for the oversight. These internal edits have been removed and grammatical errors have been corrected.